# Structural basis of antimicrobial membrane coat assembly by human GBP1

Tanja Kuhm[1], Clémence Taisne [1], Cecilia de Agrela Pinto [1], Luca Gross[2], Evdokia A. Giannopoulou [1], Stefan T. Huber[1], Els Pardon [3,4], Jan Steyaert [3,4], Sander J. Tans[1,2] & Arjen J. Jakobi [1]✉

Guanylate-binding proteins (GBPs) are interferon-inducible guanosine triphosphate hydrolases (GTPases) mediating host defense against intracellular pathogens. Their antimicrobial activity hinges on their ability to self-associate and coat pathogen-associated compartments or cytosolic bacteria. Coat formation depends on GTPase activity but how nucleotide binding and hydrolysis prime coat formation remains unclear. Here, we report the cryo-electron microscopy structure of the full-length human GBP1 dimer in its guanine nucleotide-bound state and describe the molecular ultrastructure of the GBP1 coat on liposomes and bacterial lipopolysaccharide membranes. Conformational changes of the middle and GTPase effector domains expose the isoprenylated C terminus for membrane association. The α-helical middle domains form a parallel, crossover arrangement essential for coat formation and position the extended effector domain for intercalation into the lipopolysaccharide layer of gram-negative membranes. Nucleotide binding and hydrolysis create oligomeric scaffolds with contractile abilities that promote membrane extrusion and fragmentation. Our data offer a structural and mechanistic framework for understanding GBP1 effector functions in intracellular immunity.

Robust mechanisms for recognizing and eliminating microbial pathogens are essential for maintaining the integrity of mammalian organisms. The innate and adaptive immune systems cooperate to form rapid responses to eliminate extracellular pathogens. However, many clinically relevant microbes have developed strategies to survive and replicate inside host cells[1]. As a response, mammalian cells have evolved molecular machinery that elicits effector mechanisms to defend against intracellular microbes at the level of individual cells[2]. These include pathogen elimination by autophagy, effector immune activation by interferon (IFN) cytokines and the activation of inflammasome complexes[3–7]. To subvert cytosolic surveillance, some intracellular pathogens co-opt the host cell endomembrane system to sequester themselves in pathogen-associated vacuoles[1,8]. Other pathogens disrupt this compartment to replicate in the cytosol[9–11]. In both cases, cell-autonomous immunity acts as the last line of defense against such pathogens.

One potent effector system in cell-autonomous immunity releases type I and type II IFN cytokines to induce the expression of IFN-stimulated genes. Among the most strongly induced genes is a conserved superfamily of dynamin-like guanosine triphosphatases (GTPases) including the family of guanylate-binding proteins (GBPs)[12,13]. Over the past decade, GBPs have been recognized as key players in mediating host defense against intracellular bacteria, parasites and viruses[14–17] by forming sensory platforms[4,7], affecting vacuolar integrity[6,18] or engaging directly with the membrane of cytosolic gram-negative bacteria and parasites[16,18–21].

[1]Department of Bionanoscience, Kavli Insitute of Nanoscience, Delft University of Technology, Delft, The Netherlands. [2]AMOLF, Amsterdam, The Netherlands. [3]VIB-VUB Center for Structural Biology, Brussels, Belgium. [4]Structural Biology Brussels, Vrije Universiteit Brussel, Brussels, Belgium. ✉e-mail: a.jakobi@tudelft.nl

The human genome encodes seven GBP paralogs sharing similarities with other members of the dynamin-like GTPase superfamily that undergo guanine nucleotide-dependent oligomerization and mediate membrane fusion or fission in diverse biological processes[22]. GBPs have a high intrinsic GTPase activity for hydrolysis of guanosine-5′-triphosphate (GTP) to guanosine-5′-diphosphate (GDP) without the requirement for auxiliary GTPase activating proteins or guanine nucleotide exchange factors[23–25]. The enzymology of GBPs is unique among the dynamin superfamily in that some GBPs can also bind GDP with high affinity to produce guanosine-5′-monophosphate (GMP)[26,27], which can affect bacterial growth and inflammatory signaling[28]. In the absence of infection, GBPs localize to the cytosol or sparsely associate with endogenous membranes[29]. Upon IFN induction, GBPs rapidly assemble into supramolecular membrane-associated coats on pathogen-containing vacuoles (PCVs)[18] and encapsulate cytosolic gram-negative bacteria. Corecruitment of other effectors and release of lipopolysaccharide (LPS), a glycosylated lipid component of the outer membrane of gram-negative bacteria, activate the noncanonical inflammasome pathway leading to caspase 4-dependent cleavage of gasdermin D and pyroptosis[6,20,21,30]. GBP recruitment to membranes relies on post-translational modifications of a CaaX isoprenylation motif. Three human GBPs contain CaaX motifs that lead to covalent attachment of a 20-carbon geranylgeranyl (GBP2 and GBP5) or 15-carbon farnesyl (GBP1) moiety mediating membrane association in vivo[29,31]. In its nucleotide-free state, the farnesyl moiety of GBP1 is buried in a hydrophobic pocket and requires nucleotide binding for release[32]. Subsequent conformational changes promote engagement with lipid membranes or self-oligomerization into micellar structures in the absence of lipids[33,34]. All reported antimicrobial functions of GBPs are critically dependent on GBP1 isoprenylation, rendering mechanistic insight into the conformational changes that facilitate physical engagement with membranes important for understanding their role in cytosolic host defense. In the absence of high-resolution structural data on full-length GBP1 in its activated state and on native-state structures of membrane-associated GBP assemblies, important mechanistic questions related to their mode of action remain unclear.

Here, we determined the cryo-electron microscopy (cryo-EM) structure of the full-length nucleotide-bound dimer of human GBP1. This structure reveals large-scale conformational changes of the α-helical middle and GTPase effector domains that stabilize an outstretched conformation suitable for membrane association. In vitro biochemical analysis and electron tomography of membrane-assembled GBP1 suggest a critical role of this conformation in GBP coat formation on endogenous and bacterial membranes. Importantly, we show that membrane-assembled GBP1 oligomers possess GTPase-dependent membrane-remodeling capacity that may explain observations reporting GBP-dependent modulation of membrane integrity and LPS release. Our data provide a structural framework for further studies aimed at unraveling the molecular mechanism of antimicrobial and antiviral activities of GBPs.

## Results

### A nanobody stabilizing an outstretched conformation of GBP1
GBP1 is a multidomain protein consisting of an N-terminal large GTPase (LG) domain and a C-terminal α-helical region (C-terminal helical domain, CTHD), divided into a middle domain (MD; α7–α11) and an elongated C-terminal GTPase effector domain (GED; α12–α13) (Fig. 1a)[35]. In the absence of guanine nucleotides, the GED folds onto the MD and interacts with the LG domain and MD to maintain GBP1 in the resting state. Quantitative Förster resonance energy transfer experiments suggested that nucleotide binding and hydrolysis cause major rearrangements, liberating the latch between α12 and the LG domain[36]. This extended conformation releases the C-terminal C15-farnesyl moiety required for membrane association. To map these conformational changes on a structural level, we used cryo-EM to determine

the structure of human GBP1 bound to GDP·AlF₃, a compound mimicking the GTP hydrolysis transition state[37]. Unlike the isolated LG domain that readily dimerizes under several guanine nucleotide conditions[37], full-length GBP1 forms stable dimers only in the presence of GDP·AlF₃ (Fig. 1b, Extended Data Fig. 1a and Supplementary Table 1). Two-dimensional (2D) class averages of the GDP·AlF₃-stabilized GBP1 dimer showed one predominant class comprising 92% of particles (Fig. 1c,d), while three-dimensional (3D) reconstructions showed no visible density for the stalks (MD and GED), suggesting that they are either highly flexible or engage in air–water interface interactions (Supplementary Fig. 1 and Supplementary Table 2). To stabilize the outstretched conformation for structural studies, we raised camelid antibodies (nanobodies) specific for human GBP1 and selected GBP1 binders through phage display and ELISA[38]. We tested different complementarity-determining region (CDR) clusters and selected nanobody 74 (Nb74) that bound both GBP1 monomers and dimers in an apparent 1:1 molar ratio (Fig. 1e and Extended Data Fig. 1b–c). Cryo-EM micrographs of the GDP·AlF₃-stabilized GBP1 dimer with Nb74 showed better preservation of the α-helical stalk, with Nb74 selectively binding the extended α-helical region (Fig. 1f,g).

### Nucleotide binding induces MD crossover in GBP1 dimers
We next determined the 3D structure of Nb74-bound GBP1 dimers in complex with GDP·AlF₃, yielding a pseudo-$C2$ symmetric 3D reconstruction at a nominal resolution of 3.7 Å (Fig. 1h,i, Table 1 and Extended Data Fig. 1d). The stalk region showed residual flexibility, supported by local resolution analysis and flexible refinement (Extended Data Fig. 1d, Extended Data Fig. 2a,b and Supplementary Video 1). GBP1 associates through the LG domains and additional contacts between the MDs, which extend in parallel from the LG dimer in a crossover arrangement mediated by the Gly[307]–Val[316] linker region connecting the LG domain and MD. Nb74 binds the MD at the junction formed by helices α7–α8 and α10–α11 (Fig. 1h). The GED (residues 481–592) is likely flexible and not visible in our structure. The LG dimer interface is stabilized by a large contact surface across the A-face of the GTPase domain (Extended Data Fig. 2c,d), consistent with crystal structures of the nucleotide-bound LG domain dimer[37]. Nucleotide binding induced partial unraveling of the C-terminal part of helix α6 (Ile[304]–Ser[306]) and the N-terminal end of helix α7 (Cys[310]–Val[316]) to form the crossover interface, in which the MD swings out to form contacts with the respective pairing monomer (Fig. 2a,b). This interface is primarily formed by a hydrophobic patch between the LG domain of one monomer and an aliphatic stretch of residues at the C-terminal end of the linker region and helix α7 (Ala[315]–Ile[322]) in the other monomer (Fig. 2c,d). While the MDs undergo a large spatial transformation relative to the resting state, the overall conformation of helices α7–α11 remains essentially unchanged with a root-mean-square deviation (r.m.s.d) of 2.03 Å relative to the nucleotide-free state. The parallel arrangement of the MD is stabilized by an electrostatic zipper along α9–α11 of the protruding stalks (Extended Data Fig. 2e–g). Toward the C-terminal end of the MD, the α11 helices come into proximity and form an additional contact site (Extended Data Fig. 2h–k). While the EM density at this location was not of sufficient quality to identify individual interactions, it appears to provide additional stabilization to the pseudosymmetric parallel MD arrangement. This interpretation was supported by the 3D flexible refinement (Fig. 2e and Supplementary Video 1), which showed that both MDs undergo concerted motion relative to the LG domains. Close inspection of the EM map revealed additional weak density interspersed between helices α3 and α3′ in the LG domain and protruding beyond the apical end of the MD. A model-independent implementation of local density sharpening[39] allowed tracing the α3–α3′ loop (residues 156–167) (Extended Data Fig. 2o). In addition, we observed tubular density protruding from the α11 helices, which likely corresponds to the N-terminal end of the flexible α12 helices of the GEDs (Extended Data Fig. 2p).

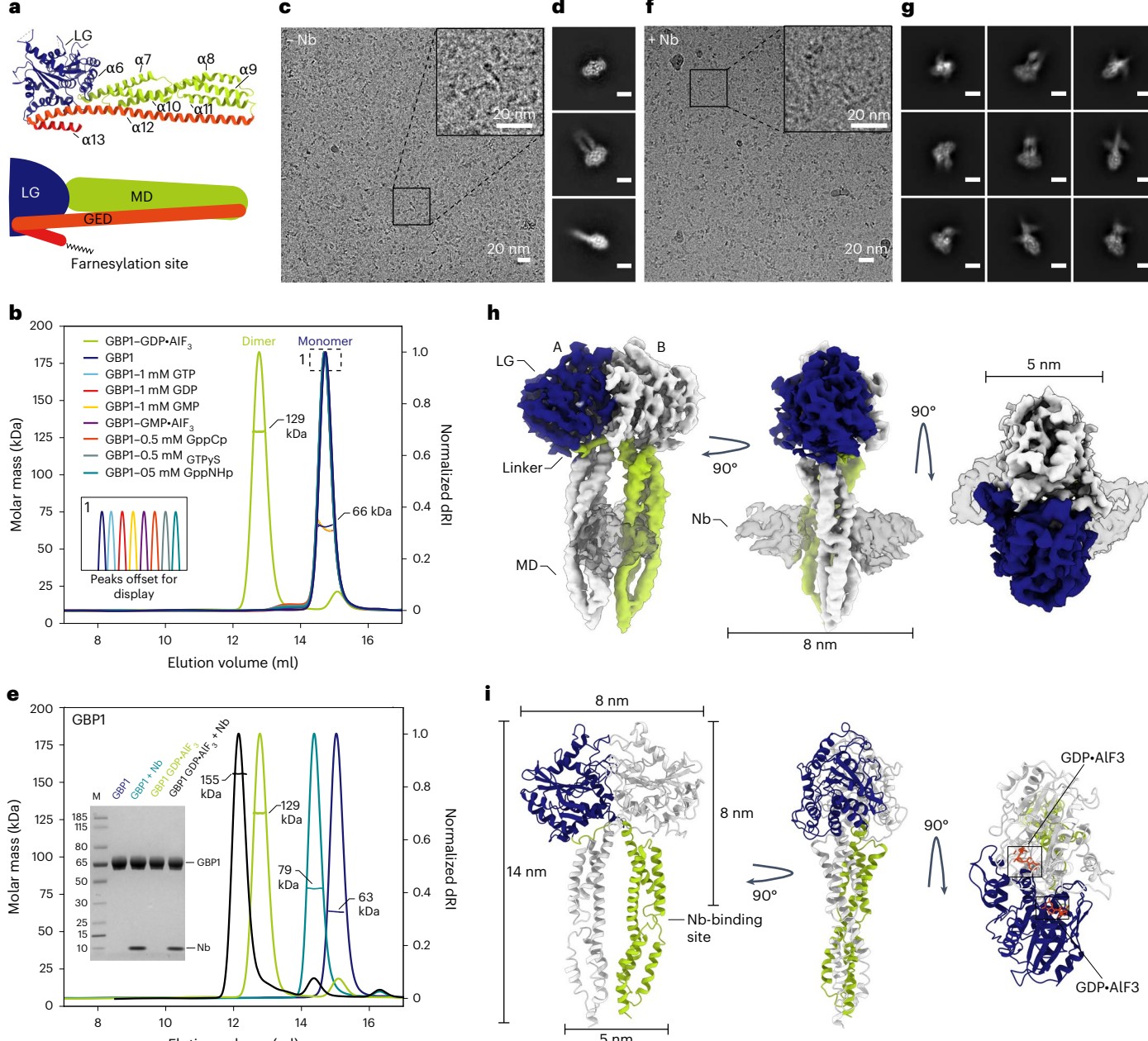

**Fig. 1 | Cryo-EM structure of the GDP·AlF₃-stabilized GBP1 dimer. a**, Atomic model of monomeric GBP1 (PDB 1DG3) in cartoon representation alongside a schematic representation displaying the domain architecture. LG domain, blue; MD, green, GED, orange and red. Individual α-helices in the MD and GED are numbered sequentially. **b**, SEC–MALS analysis of GBP1 with different nucleotides. GBP1 appears monomeric on SEC–MALS in the presence of GTP, GDP, GMP, GMP·AlF₃, GppCp, GTPyS or GppNHp, while a dimer peak emerges in the presence of GDP·AlF₃. The experimentally determined molecular weight is plotted across the chromatographic peak and is reported in kDa. **c,d**, Representative cryo-EM micrograph (**c**) and 2D class averages (**d**) of GBP1–GDP·AlF₃. The scale bar in **d** is 5 nm. **e**, SEC–MALS analysis of recombinant GBP1:Nb74 complex in the presence and absence of GDP·AlF₃. An SDS–PAGE analysis of peak fractions is also shown (molecular marker in leftmost lane, in kDa). **f,g**, Cryo-EM micrograph (**f**) and 2D class averages (**g**) of GBP1–GDP·AlF₃–Nb74 (scale bar, 5 nm). **h**, The 3D cryo-EM density map of the GDP·AlF₃-stabilized GBP1 dimer bound to Nb74. **i**, Refined atomic model of the GBP1 dimer as derived from fitting into the cryo-EM density in **h**. The nucleotide-binding sites located at the LG dimer interface are highlighted in orange.

## Farnesylated GBP1 (GBP1_F) forms soluble micelles in absence of lipids

Our structure was obtained with recombinant protein devoid of the farnesyl modification normally required for association with membranes[40]. In this structure, the MDs point away in parallel from the LG domains, compatible with a model in which nucleotide binding primes both farnesyl anchors for membrane insertion. To test this hypothesis, we coexpressed GBP1 together with the human farnesyl transferase machinery to natively purify GBP1_F. Unexpectedly,

size-exclusion chromatography (SEC) of GBP1_F with GDP·AlF₃ did not yield GBP1 dimers (Fig. 1b) but often showed an additional peak corresponding to higher-molecular-weight species (Extended Data Fig. 3a). Negative-stain imaging of this fraction revealed circular particles 58 nm in diameter resembling flowers with discernible petals formed by a dense perimeter and spoke-like protrusions extending toward the particle center (Fig. 3a,b). These structures are consistent with previous observations[33] and were present in higher occurrences without size-exclusion separation before the imaging experiment.

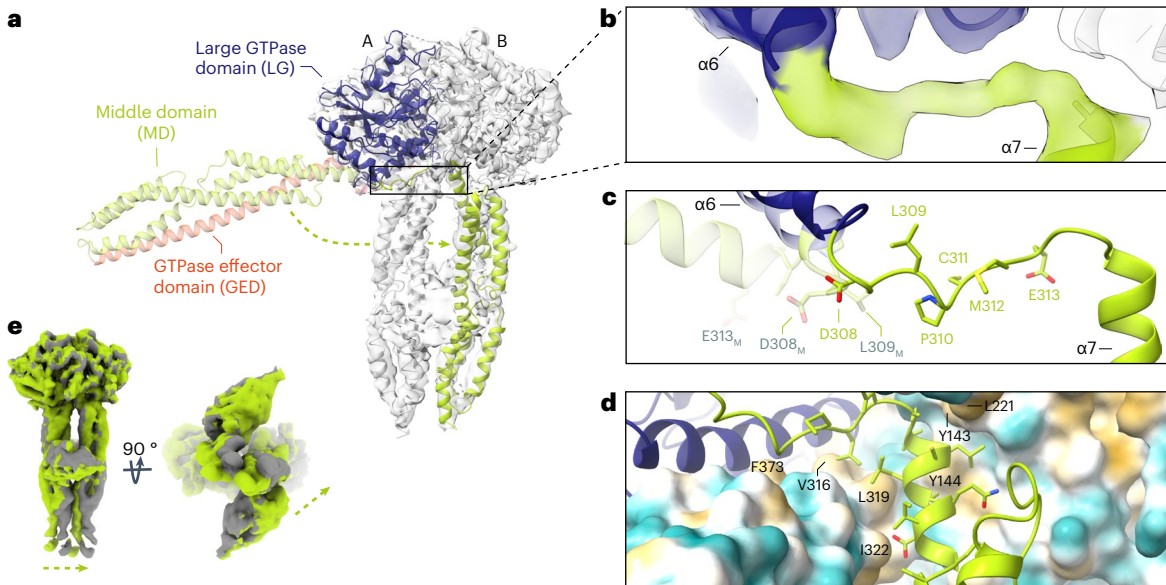

**Fig. 2 | Structural details of the GDP·AlF₃-stabilized GBP1 dimer. a**, The full-length nucleotide-free GBP1 monomer (pale colors; PDB 1DG3) is superposed onto one of the monomers in the GDP·AlF₃-stabilized GBP1 dimer (bright colors) by least-squares superposition of their LG domains. Dimer formation involves a large conformational rearrangement in which the α-helical MDs swing across each other to form a parallel arrangement protruding from the LG domains. **b**, Close-up view of the crossover region (in cartoon representation, superposed on the EM density) formed by a linker region Gly³⁰⁷–Val³¹⁶ originating from partially unraveled helices α6 and α7. **c**, Residues 307–316, which are part of a linker connecting α6 and α7, need to flip by 180° when changing from the monomer into the dimer conformation. Relevant residues in the nucleotide-free conformation are labeled in gray. **d**, The dimer conformation is stabilized by a hydrophobic pocket formed by residues located on α3, α4′ and α7, interacting with an aliphatic stretch encompassing residues Ala³¹⁵–Ile³²² on helix α7 of the second monomer. **e**, Flexibility of the α-helical MD visualized through flexible refinement. End points of the density morph along one exemplary latent space dimension are displayed (green and gray densities). The arrow indicates the direction of movement.

To test whether these structures require transition-state GBP1, we also prepared samples in the presence of GTP and observed equivalent particles (Fig. 3b) albeit at lower occurrence and requiring higher GBP1$_F$ concentrations. To gain more insight into their molecular architecture, we prepared cryo-EM samples of GBP1$_F$ in the presence of GDP·AlF₃ (Extended Data Fig. 3b) and performed 2D class averaging (Fig. 3a,c). Additional electron cryo-tomograms showed that the particles are spherical micelles and not discs (Extended Data Fig. 3c, Supplementary Table 2 and Supplementary Video 2). The GBP1 assemblies in tomograms and 2D averages appear highly ordered. The increased detail in cryo-EM micrographs of individual particles allowed discerning spherical densities (4.5 nm ± 0.7 nm in diameter, $n$ = 50) at the particle periphery connected to spokes that extend radially toward the center, which we assigned to the LG domains and α-helical stalks, respectively. We next quantified the dimensions for comparison with our high-resolution model of the GBP1 dimer. If two oppositely oriented GBP1 dimers assemble through interactions at the C terminus of their respective MDs, the resulting assembly would span 28 nm (Fig. 1i). Instead, we found the rim-to-rim diameter of the assemblies to be 58.1 ± 1.2 nm ($n$ = 33), suggesting that additional structural elements are required to make up the remaining distance. To map the location of the MD within the flower-like assembly, we incubated GBP1$_F$ with GDP·AlF₃ in the presence of Nb74. Negative-stain images of this sample showed particles with additional spherical density at approximately 8 nm of radial distance to the LG domain (Fig. 3d–f), consistent with the position of Nb74 in the cryo-EM structure of the Nb74-bound GBP1 dimer (Fig. 1h,i). The overall particle radius of 29 nm is consistent with GBP1 containing a fully unlatched α12 helix (compare Fig. 1a,i), suggesting that the remaining density toward the particle center comprises the GED. The center in both negative-stain images (5.6 ± 0.8 nm, $n$ = 27) and cryo-EM micrographs (6.3 ± 1.1 nm, $n$ = 22) displayed higher contrast than the peripheral LG domain and MD and GED spokes of the petal. Because the diameter of the particles is incompatible with a fully extended conformation of the entire GED, we hypothesize that this density corresponds to a cluster of α13 helices of the GED and the exposed farnesyl anchors (Fig. 3g).

## GBP1$_F$ forms membrane coats and scaffolds tubular protrusions

Fluorescently labeled GBP1$_F$ (GBP1$_F$-Q577C–Alexa Fluor 647) uniformly stained brain polar lipid extract (BPLE)-derived giant unilamellar vesicles (GUVs) (Fig. 4a). To determine at the structural level if the conformation observed in the lipid-free GBP1$_F$ micelles is relevant for membrane association, we mixed BPLE-derived small unilamellar vesicles (SUVs) with GBP1$_F$–GDP·AlF₃ or GBP1$_F$ GTP for transmission EM (TEM) analysis. Cryo-EM micrographs of these samples showed SUVs densely covered with a proteinaceous coat of 29.6 ± 3.1 nm ($n$ = 47) in radial extension (Fig. 4b–d), consistent with the dimensions of the extended GBP1$_F$ conformation observed in the lipid-free GBP1 micelles. Strikingly, we observed either fully coated or uncoated SUVs (Fig. 4b,c), suggesting cooperativity in membrane association. On a subset of SUVs, we observed extended tubular protrusions of 59.8 ± 2.4 nm ($n$ = 37) in diameter scaffolded by GBP1 in an arrangement reminiscent of that on spherical liposomes. Cryo-EM micrographs of such structures in unsupported ice revealed these protrusions to be highly flexible (Fig. 4e), precluding 2D averaging. The formation of protrusions was highly concentration dependent, transitioning from uniformly coated SUVs to scaffolded tubule extrusion beyond a certain threshold concentration (Fig. 4f and Extended Data Fig. 4a,b). The 2D class averages and associated power spectra of negatively stained protrusions showed repetitive features consistent with overall dimensions of laterally associated GBP1 molecules (Extended Data Fig. 4c and Fig. 1i). The micrographs did not allow us to uniquely discriminate whether the protrusions contained a membrane or were formed by excess GBP1 through the aggregation of exposed farnesyl anchors like GBP1 micelles. To test these possibilities, we also performed concentration series experiments with GBP1$_F$ in the

**Table 1 | Cryo-EM data collection, refinement and validation statistics**

| | GBP1–GDP·AlF$_3$–Nb74 |
|---|---|
| | **(EMD-16794), (PDB 8CQB)** |
| **Data collection and processing** | |
| Magnification | ×105,000 |
| Voltage (kV) | 300 |
| Electron exposure (e⁻ per Å²) | 60 |
| Defocus range (μm) | −0.6 to −2.2 |
| Pixel size (Å) | 0.834 |
| Symmetry imposed | *C1* |
| Initial particle images (no.) | 432,341 |
| Final particle images (no.) | 181,161 |
| Map resolution (Å) | 3.7 |
| FSC threshold | 0.143 |
| Map resolution range (Å) | 3.0–6.0 |
| **Refinement** | |
| Initial model used (PDB code) | 1F5N and 2B92 |
| Model resolution (Å) | 3.9 (4.1 unmasked) |
| FSC threshold | 0.5 |
| Model resolution range (Å) | 3.1–5.9 |
| Map sharpening *B* factor (Å²) | −158 |
| Model compositions | |
| Nonhydrogen atoms | 7,688 |
| Protein residues | 956 |
| Ligand | 4 |
| *B* factors (Å²) | |
| Protein | 75.1 |
| Ligand | 52.6 |
| R.m.s.d. | |
| Bond lengths (Å) | 0.005 |
| Bond angles (°) | 0.993 |
| **Validation** | |
| MolProbity score | 0.94 |
| Clashscore | 5.28 |
| Poor rotamers (%) | 1.29 |
| Ramachandran plot | |
| Favored (%) | 93.49 |
| Allowed (%) | 6.1 |
| Disallowed (%) | 0.42 |

FSC, Fourier shell correlation.

absence of lipids. Under these conditions, we did not observe tubular structures, suggesting that filamentation of GBP1 involves extrusion of membrane material (Extended Data Fig. 4d).

## GTP hydrolysis promotes membrane fragmentation

We next tested whether GBP1-scaffolded protrusions persist in conditions of GTP turnover. Importantly, for conditions containing GTP, we exclusively observed short tubular membrane stubs scaffolded by a GBP1 coat, while coated SUVs were absent in micrographs, suggesting that GTP hydrolysis drives GBP1-dependent scission or fragmentation of liposomes (Fig. 4g–i and Extended Data Fig. 5a). Consistent

with the higher GBP1 concentrations required for the formation of micellar assemblies in the absence of lipids, we observed weaker binding to GUVs for equimolar levels of GBP$_F$ in the presence of GTP compared to GDP·AlF$_3$ (Fig. 4a and Extended Data Fig. 5b), providing additional support for threshold-dependent activity. To probe the consequences of GTP-dependent GBP1 coat formation in real time, we made use of a dual-trap optical tweezer assay coupled to confocal microscopy. Two 2-μm silica beads were held in optical traps at 6 μm of trap separation within a laminar flow cell operated at constant pressure. One bead coated with a bilayer membrane containing rhodamine 6G-labeled lipids served as a membrane donor, whereas the second uncoated 'catch' bead served to sequester lipid material released from the donor bead (Fig. 4j). We monitored lipid transfer from the donor bead to the catch bead using fluorescence imaging (Fig. 4k and Supplementary Video 3). In the absence of GBP1$_F$, fluorescence in the interbead space and on the catch bead remained at a constant baseline level (Fig. 4l). We then dispensed GBP1$_F$ into the flow channel near the donor bead and in the presence of GTP. If GTP-dependent membrane scaffolding by GBP1$_F$ results in membrane scission, membrane fragments released from the donor bead would be sequestered by the catch bead under continuous flow. Indeed, we found lipid fluorescence in the interbead space and on the catch bead to increase ~2-fold and ~7-fold, respectively, approximately 30 s after the addition of GBP1$_F$ (*n* = 18) (Fig. 4l). This suggests that lipid material is released from the donor bead in a GBP1$_F$-dependent manner. Control experiments with GDP, GDP·AlF$_3$ or a GTPase activity-deficient mutant of GBP1 (GBP1$_F$-R48A) or in the absence of GBP1 showed that lipid release is strictly dependent on the GTPase activity of GBP1 (Extended Data Fig. 5c–f). Altogether, our data indicate that GBP1 scaffolding can promote severing of bilayer membranes and lipid release dependent on GTPase activity.

In tomographic reconstructions of GBP1-coated SUVs, we observed a continuous GBP1 coat stabilized by the lateral association of GBP1 subunits (Fig. 5a,b, Supplementary Table 3 and Supplementary Videos 2 and 4). The partially regular appearance of the coat in individual *z* slices was indicative of short-range order and appeared to be mediated primarily through contacts of adjacent LG domains. To test this hypothesis, we also acquired cryo-EM micrographs of GBP1-coated SUVs in the presence of Nb74, which, through its interactions with the MD domain, may sterically affect the lateral association of GBP1 dimers in the coat (Fig. 1h,i). Indeed, for these conditions, we frequently observed partially coated SUVs with signs of a structurally disordered coat (Fig. 5c), suggesting that perturbation of the lateral association affects coat stability. To investigate whether Nb74 also affects GBP1 coat formation on gram-negative bacteria, we imaged mCerulean3-expressing *Salmonella* Typhimurium after incubation with GBP1$_F$ and recombinant green fluorescent protein (GFP)–Nb74. Nb74 did not completely abrogate coat formation but appeared to be confined to sharply delimited puncta on the coated bacteria. At and surrounding these puncta, GBP1 coat density was noticeably weakened, consistent with our cryo-ET observations showing partially disrupted coats on SUVs (Fig. 5d and Extended Data Fig. 6a,b). We next asked how Nb74 binding affects GBP1 coat formation in bacteria-infected cells and generated clustered regularly interspaced short palindromic repeats (CRISPR)–Cas9-engineered HeLa *GBP1* knockout (KO) cells stably expressing mCherry–GBP1 under a Tet-inducible promoter (HeLa *ΔGBP1* + Tet-mCherry-*GBP1*). We then infected *ΔGBP1* + Tet-mCherry-*GBP1* cells transiently expressing GFP–Nb74 with *S.* Typhimurium and imaged GBP1 coat formation at 2 h after infection using confocal microscopy. Consistent with the in vitro observations on bacteria alone, we stained cytosolic bacteria nonuniformly with Nb74 preferentially accumulated within distinct patches (Fig. 5e and Extended Data Fig. 6c). However, local confinement of Nb74 was less prominent in cells compared to the in vitro assays, suggesting that GBP1 coats on cytosolic bacteria cells are more permissive to GFP–Nb74 integration under the conditions tested.

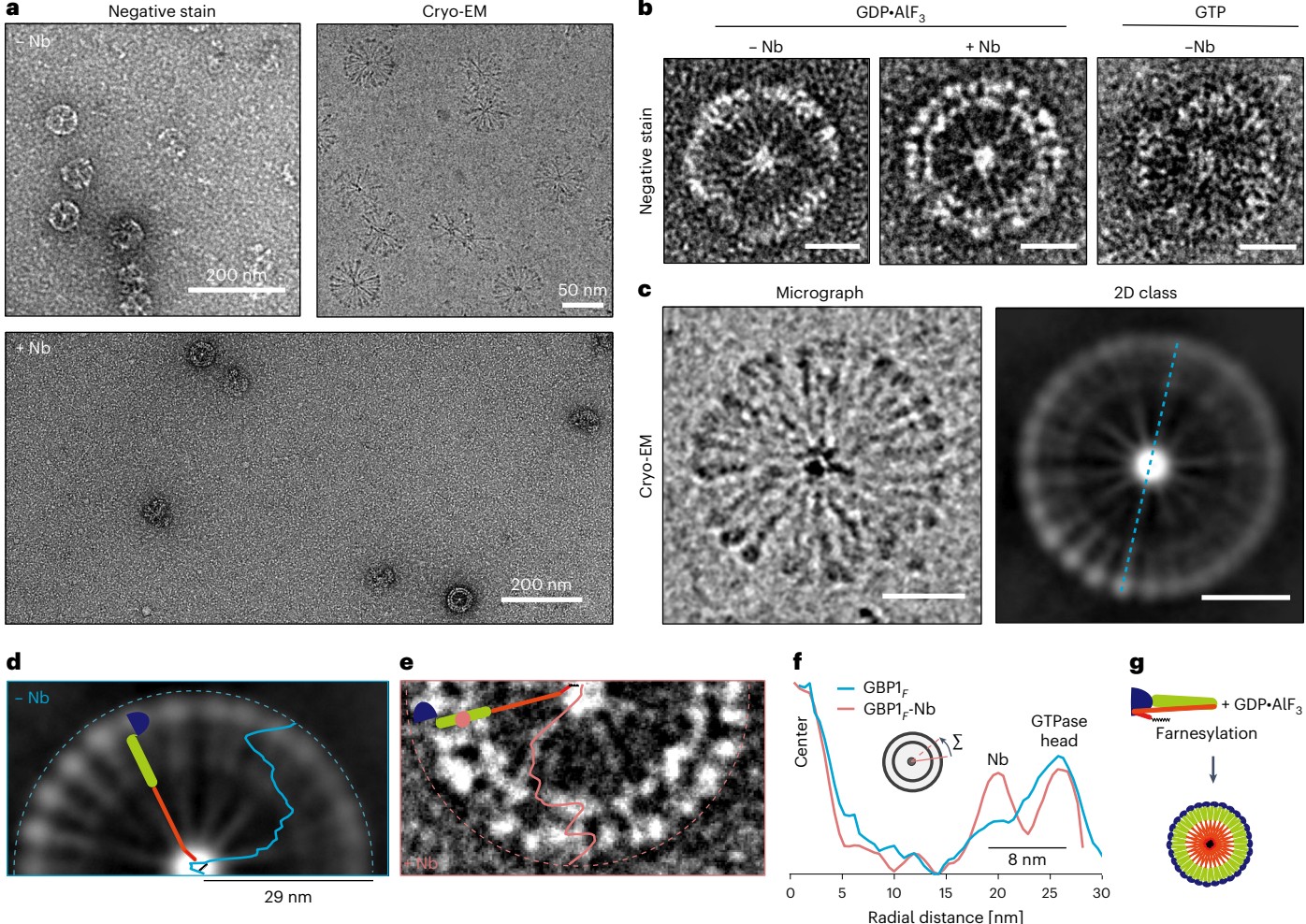

**Fig. 3 | Micellar self-assembly by GBP1$_F$ –GDP·AlF$_3$. a**, Negative-stain images (top left) and cryo-EM images (top right) of GBP1$_F$ –GDP·AlF$_3$ and negative-stain images of GBP1$_F$ –GDP·AlF$_3$ with Nb74 (bottom) showing the abundant formation of flower-like GBP1 micelles. **b**, Close-up view of individual GBP1$_F$ –GDP·AlF$_3$ micelles formed by GBP1$_F$ alone or in the presence of Nb74 (GBP1$_F$:Nb74). A GBP1$_F$ micelle formed in the presence of GTP is also shown (scale bar, 20 nm). **c**, Left, close-up view of a cryo-EM micrograph with a GBP1$_F$ –GDP·AlF$_3$ flower showing discernible repetitive elements. Right, cryo-EM 2D class average of GBP1 micelles highlighting spherical densities at the perimeter and spokes pointing toward the particle center (scale bar, 20 nm). **d,e**, Radially averaged intensity profiles of GBP1 micelles plotted on top of a cryo-EM 2D class average of GBP1$_F$ –GDP·AlF$_3$ micelles (**d**) or on top of a negative-stain image of a GBP1$_F$ –GDP·AlF$_3$ micelle formed with Nb74 (**e**). A second ring of globular density is visible at 8 nm of distance from the perimeter. **f**, Radially averaged intensity profiles from **d,e** plotted for comparison. **g**, Schematic representation of GBP1$_F$ –GDP·AlF$_3$ micelle formation.

## The MD crossover is critical for membrane association

We next asked whether the crossover arrangement of the nucleotide-activated GBP1 dimer is required for membrane association. To test this hypothesis, we sought to identify mutants that disrupt the interfaces stabilizing the extended conformation but retain the ability to form dimers through the LG domains. First, we analyzed sequence conservation in the α6–α7 region forming the loop structure in the crossover conformation, the MD–LG interface, the electrostatic zipper motif and the C-terminal contact site of the MD and designed point mutants to weaken conserved motifs and interactions (Fig. 6a,b and Extended Data Fig. 7). The main interface in the GBP1 dimer is formed between the LG domains of both monomers, contributing 2,138 Å$^2$ (62%) of the total buried surface area of the dimer interface as inferred from our structure. We, therefore, expected the different variants to retain their ability to form LG dimers and enzymatic activity (Fig. 6c and Extended Data Fig. 8) but to reduce membrane association by destabilizing the parallel arrangement of the MDs. To test this hypothesis, we mixed BPLE SUVs with GBP1$_F$ variants activated by GDP·AlF$_3$ and performed cosedimentation assays followed by quantitative SDS–PAGE analysis of pellet and supernatant fractions. Of the four

variants tested, two encompassing substitutions in the crossover region showed 33% (D308S; $P = 0.011$) and 44% (D308S;L309A;P310A; $P = 0.0014$) reductions in the membrane-bound fraction compared to the control (Fig. 6d,e). These bulk observations were supported by negative-stain imaging of SUVs incubated with the two GBP1 variants (Extended Data Fig. 8), showing substantially decreased coat formation but no complete disruption. For variants affecting the LG–MD (Y143A) and MD–MD (K466D) interfaces, we found no significant effect (Fig. 6e), suggesting that the crossover arrangement is the primary determinant for membrane association of GBP1.

## Dimers are the essential unit of GBP coats on LPS membranes

In addition to targeting intracellular membranes, GBP1 has been reported to directly associate with LPS, a glycosylated lipid component of the outer membrane in gram-negative bacteria. LPS consists of a lipid A moiety mediating the integration in the membrane leaflet, a core region of nonrepetitive oligosaccharides and the O-antigen consisting of an extended and branched chain of repetitive oligosaccharides. The LPS composition can vary greatly across bacterial strains. To determine whether GBP1 coat formation is dependent on the specific

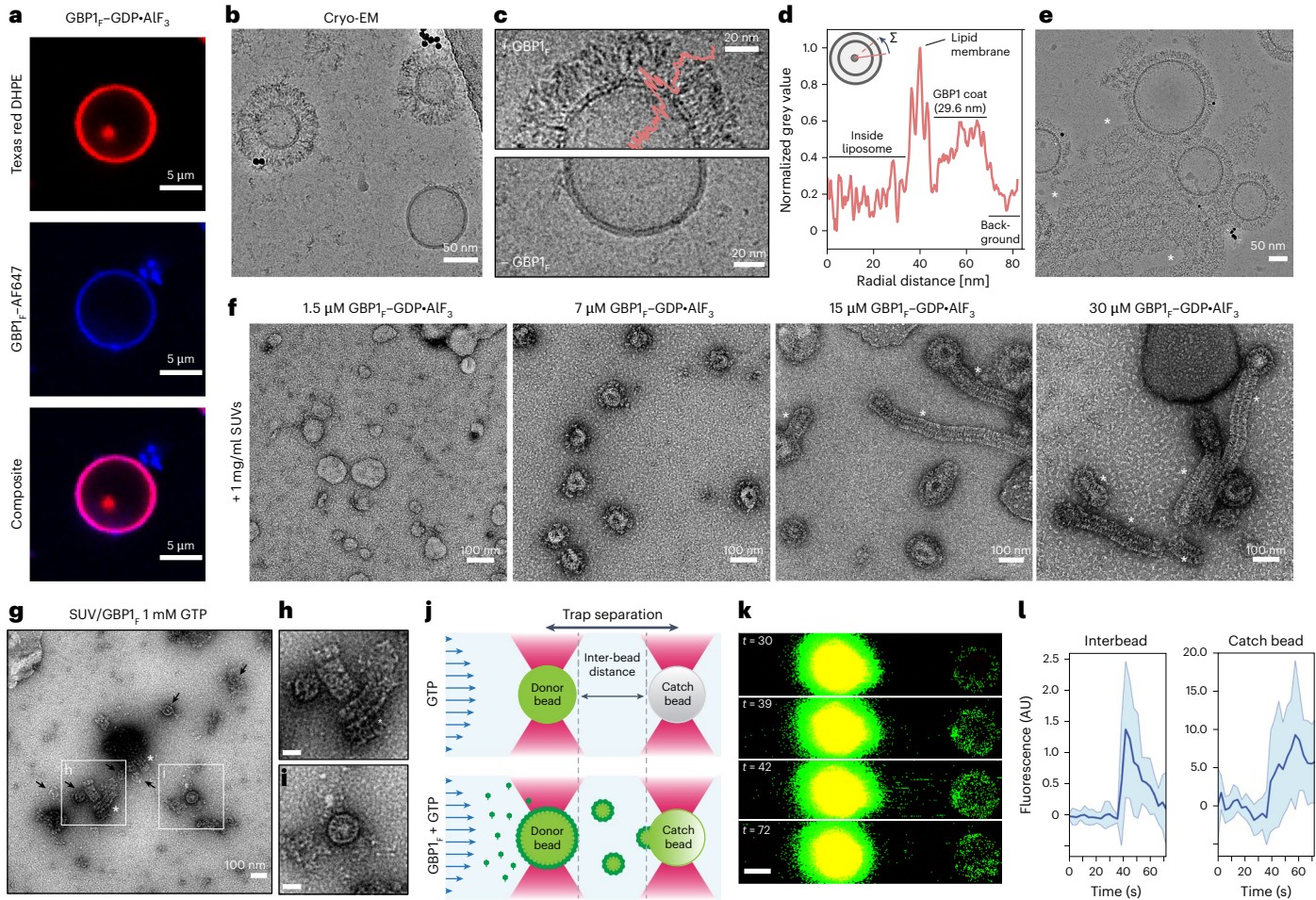

**Fig. 4 | Coat formation and membrane-remodeling capability of GBP1.**
**a**, Confocal fluorescence imaging of Alexa Fluor 647-labeled GBP1$_F$ binding to Texas red DHPE-labeled BPLE GUVs in the presence of GDP·AlF$_3$. **b**, Cryo-EM micrograph of GBP1$_F$–GDP·AlF$_3$ binding to BPLE liposomes. **c**, Close-up view of BPLE SUVs with or without GBP1$_F$–GDP·AlF$_3$ coat. **d**, Radially averaged intensity profile across a GBP1$_F$-coated SUV. **e**, Cryo-EM micrographs of tubular protrusions (asterisks) formed on GBP1-coated SUVs. **f**, GBP1 coat formation is concentration dependent. A visible coat starts to form at GBP1$_F$ concentrations of 7 µM. At a concentration of 15 µM and higher, GBP1-coated tubular protrusions (asterisk) become visible and are the dominant structures at concentrations exceeding 30 µM. **g–i**, In the presence of GTP, GBP1$_F$ remodels SUVs into spherical micelles (arrows) and short filaments (asterisks). Scale bar in **h** and **i**, 50 nm. **j**, Dual-trap membrane fragmentation assay. A donor bead coated with

rhodamine 6G-labeled membrane is held in place by an optical trap in a flow cell operating at constant pressure. A second, uncoated catch bead is held in an additional optical trap at 6 µm of distance from the donor bead. Bottom, GBP1$_F$-dependent membrane scission or fragmentation would result in membrane transfer from the bead-supported donor membrane to the catch bead. The schematic transfer of membrane vesicles is shown for illustrative purposes only and the precise structure of membrane fragments and the GBP1$_F$ scaffold is unclear. **k**, Representative confocal fluorescence images illustrating GBP1$_F$-dependent lipid transfer from the donor bead (DB) to the catch bead (CB). Scale bar, 1 µm. **l**, Integrated fluorescence intensity time traces of the interbead space and the catch bead in the presence of GBP1$_F$ and GTP. Solid lines represent mean fluorescence intensities of 18 experiments and shaded areas represent 95% confidence intervals.

oligosaccharide structure of LPS, we incubated nucleotide-activated GBP1$_F$ with three different LPS chemotypes from bacterial pathogens differing in the presence of inner and outer core sugars and O-antigen components; *S.* Typhimurium LPS containing extended O-antigen (LPS-ST), smooth LPS from *Escherichia coli* O111:B4 (LPS-EB) and deep rough LPS from *S. enterica* sv. Minnesota R595 (LPS-SM) consisting of only the lipid A core (Fig. 7a and Supplementary Table 4). LPS containing the outer core and O-Ag forms elongated bilamellar micelles, whereas deep rough LPS displays a semivesicular morphology. For all three cases, we observed a dense GBP1 coat on remodeled LPS micelles, extending ~28 nm from the center and sandwiching a parallel layer of continuous density with a ~5–7-nm cross-section, compatible with the estimated thickness of a micellar bilayer and fuzzy contributions of oligosaccharide residues (Fig. 7a). These dimensions agreed with the GBP1 coat on brain polar lipid SUVs, suggesting that the overall mode of assembly is similar. As GBP1 formed equivalent coats on all LPS forms

tested, we conclude that the primary association with LPS is mediated by insertion of the C-terminal farnesyl anchor in the lipid layer but our data preclude quantitative conclusions for potential preference for certain types of LPS over others. Analogous to micellar assemblies and GBP1 coats on lipid SUVs, the coat on LPS micelles showed GBP1 molecules assembling together (Supplementary Fig. 2). To analyze the assembly mode within the coat, we performed 2D class averaging of individual rims of GBP1-coated LPS micelles (Fig. 7b). The 2D class averages of the GBP1 coat revealed low-resolution densities compatible with our high-resolution GBP1 dimer structure viewed in projection, suggesting that dimers form the repeating unit in the mature GBP1 coat. In some of the classes, we also observed rodlike density extending from the MD toward the LPS lipid surface, consistent with an extended GED bridging the distance to the membrane (Fig. 7b). Together, our results support a model in which nucleotide binding by GBP1 unlatches the C-terminal all-α-helical MD and GED from the LG domain, leading to a

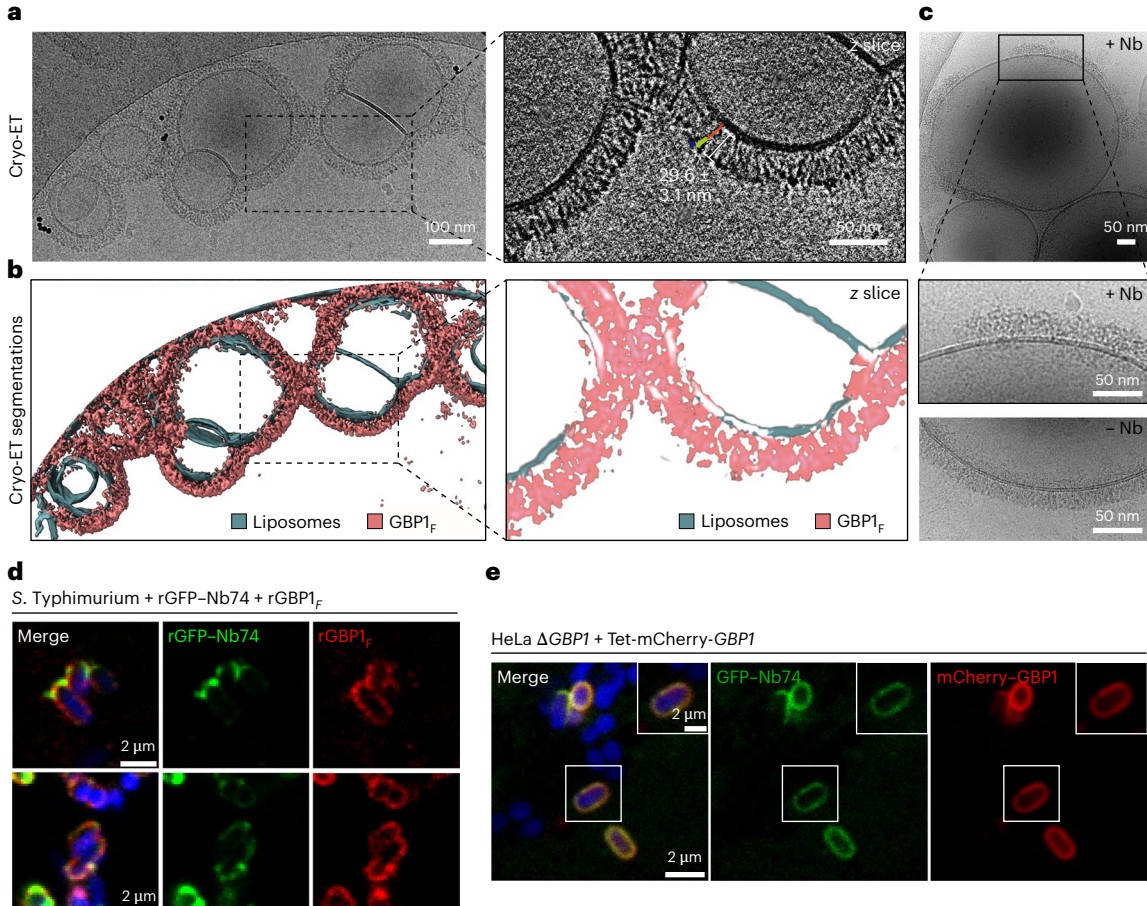

**Fig. 5 | Electron cryo-tomography and confocal imaging of the GBP1 coat.**
**a**, Electron cryotomogram of GBP1-coated liposomes. Projected z stack of reconstructed tomogram and close-up view of an individual z slice showing discernible repetitive subunits. **b**, Segmented tomogram from **a**. GBP1, pink; membranes, green. **c**, Cryo-EM micrograph of GBP1$_F$-coated liposomes in the presence of Nb74. Bottom, the close-up view shows a comparison to GBP1$_F$-coated liposomes in the absence of Nb74. The presence of Nb74 results in only partially coated liposomes with a higher degree of structural disorder. **d**, Confocal microscopy images of immobilized S. Typhimurium incubated with GBP1$_F$ in the presence of recombinant GFP–Nb74. **e**, Confocal microscopy of HeLa $\Delta$GBP1 Tet-mCherry-GBP1 cells expressing GFP–Nb74 at 2 h after infection with mCerulean3-expressing S. Typhimurium. Both immobilized bacteria and cytosolic bacteria in infected cells showed a nonuniform distribution of GFP–Nb74 across the GBP1 coat.

swing-like conformational transition of the MD that reassociates with the LG of the adjacent monomer and forms a parallel arrangement of extended GEDs for association with membranes (Fig. 7c). Interestingly, the dimensions of an extended GED are compatible with the lateral dimensions of extended bacterial LPS O-antigen (LPS-ST, 10.3 ± 3 nm, n = 23; LPS-EB, 13.1 ± 2.1 nm, n = 11; measured in negative-stain EM), suggesting that these may represent a functional adaption to allow intercalation between the dense O-antigen and core oligosaccharide on LPS-containing membranes.

**Two polybasic motifs differentially affect LPS binding**
GBP1 variants of two distinct polybasic motifs have been reported to affect coat formation on intracellular S. Typhimurium, *Franciscella novicida* and *Shigella flexneri*[16,20,41]. To test whether these are functionally linked to formation of the outstretched crossover conformation of GBP1, we first expressed recombinant GBP1$_F$[K61−63A] and GBP1$_F$[R584−586A] and tested their ability to dimerize in the presence of GDP·AlF$_3$. The K61−K63 motif is located in the α1−β2 loop near the LG dimerization interface (Extended Data Fig. 9a) and stabilizes the β6−α5 loop involved in nucleotide coordination. Accordingly, the GBP1$_F$[K61−63A] mutant showed a markedly reduced dimer fraction relative to wild-type (WT) GBP1. In contrast, the R584−R586 motif is located near the end of α13, directly adjacent to the farnesylation site. For this mutant, dimerization propensity was

unaffected (Extended Data Fig. 9b,c). We next tested whether GBP1$_F$[K61−63A] and GBP1$_F$[R584−586A] affected the ability of coat formation and membrane remodeling. We incubated nucleotide-activated GBP1$_F$[K61−63A] and GBP1$_F$[R584−586A] with BPLE SUVs or micelles or LPS-EB for TEM imaging. Unlike WT GBP1$_F$, which efficiently coated and remodeled SUVs and LPS, GBP1$_F$[K61−63A] only sparsely decorated LPS and SUVs and did not assemble into the characteristic micellar assemblies observed for WT GBP1$_F$ in the absence of lipids (Extended Data Fig. 9d). GBP1$_F$[R584−586A] entirely failed to associate with liposomes or LPS. Instead, GBP1$_F$[R584−586A] polymerized into elongated assemblies even in the absence of lipids (Extended Data Fig. 9d). For WT GBP1$_F$, filamentous structures formed only with lipids (Fig. 4 and Extended Data Fig. 4a,d), suggesting that the positively charged α13 requires negatively charged lipids for filamentous packing. Charge neutralization in GBP1$_F$[R584−586A] eliminated this requirement, promoted constitutive polymerization and reduced membrane association. Collectively, these findings explain the altered phenotypes and loss of cytosolic bacteria localization seen in previous studies[16,20,41].

**MD crossover is required for antibacterial coat formation**
Having established the importance of the outstretched crossover arrangement of GBP1 for binding to LPS membranes in vitro using GDP·AlF$_3$-stabilized dimers, we next asked whether it is required for antimicrobial coat formation in infected cells using the previously

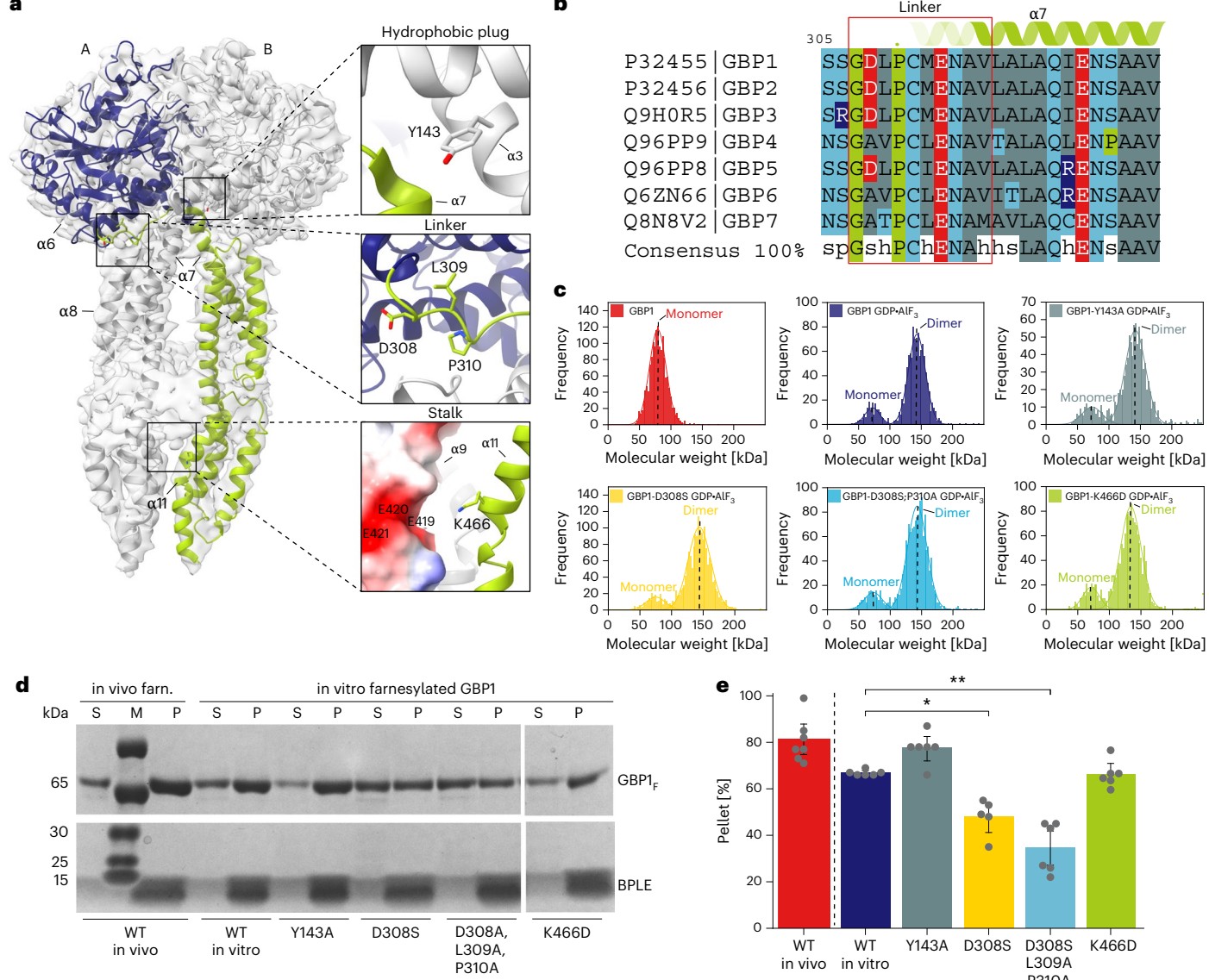

**Fig. 6 | Effect of GBP1 variants on membrane association. a**, Schematic overview of the GBP1 dimer highlighting individual point substitution sites of tested variants. **b**, Multiple-sequence alignment of all human GBP paralogs zooming in on the linker region (307–316) between α6 and α7. The overall linker region is highly conserved; D308 is conserved across all GBPs containing prenylation motifs (+GBP3). **c**, Mass photometry spectra of nucleotide-free GBP1, GBP1–GDP·AlF3 and GDP·AlF3-stabilized GBP1 variants. The determined molecular weights are indicated and show that the point substitutions did not

affect the ability of the GBP1 variants to dimerize. **d**, Representative SDS–PAGE analysis of cosedimentation assay of GBP1$_F$ variants with BPLE liposomes. S, supernatant; P, pellet. **e**, Quantitative analysis of the cosedimentation assay. The sum of densitometric intensities of protein in the pellet and supernatant fractions was used to determine the relative percentage of GBP1 in each fraction. The mean intensity and s.d. are displayed. Statistical significance was determined using a two-sided Welch's $t$-test with Bonferroni correction (*$P$ = 0.0106 and **$P$ = 0.00138; $n$ = 6).

characterized GBP1 variants (Fig. 6). We used CRISPR–Cas9-engineered HeLa *GBP1* KO cells stably expressing mCherry–GBP1 variants under a Tet-inducible promoter (HeLa *ΔGBP1* + Tet-mCherry-*GBP1*) cells for infection with mCerulean3-expressing *S*. Typhimurium and quantified coat density on cytosolic bacteria at 2 h after infection (Fig. 7d). Consistent with the in vitro cosedimentation assays, we observed the most significant effect for the crossover mutant D308;L309;P310A, which resulted in a 56% reduction in coat density relative to WT GBP1. Other mutants (K466D and Y143A) showed weaker effects with 35% and 40% reductions in GBP1 coat density, respectively (Fig. 7e). The D308S mutant alone showed no significant effect. To verify that observed differences in coat formation are not in part the result of differential doxycycline (Dox)-induced expression levels, we quantified relative expression levels of WT and variant GBP1 and found no significant

differences (Fig. 7f and Extended Data Fig. 8). Together, the data from in vitro and in cellulo experiments suggest that the outstretched MD crossover conformation of nucleotide-activated GBP1 dimers is important for promoting efficient coat formation on target membranes.

## Discussion

GBPs have recently emerged as important effector molecules in cell-autonomous immunity against intracellular bacteria and GBP1 forms the central organizing unit of this cellular response. The main antimicrobial function of GBP1 has been ascribed to its ability to coat the membrane of pathogen-containing compartments or the outer membrane of gram-negative cytosolic bacteria, where it appears to form a multivalent signaling platform activating the noncanonical inflammasome[16,17,20,21]. Coat formation is dependent on nucleotide

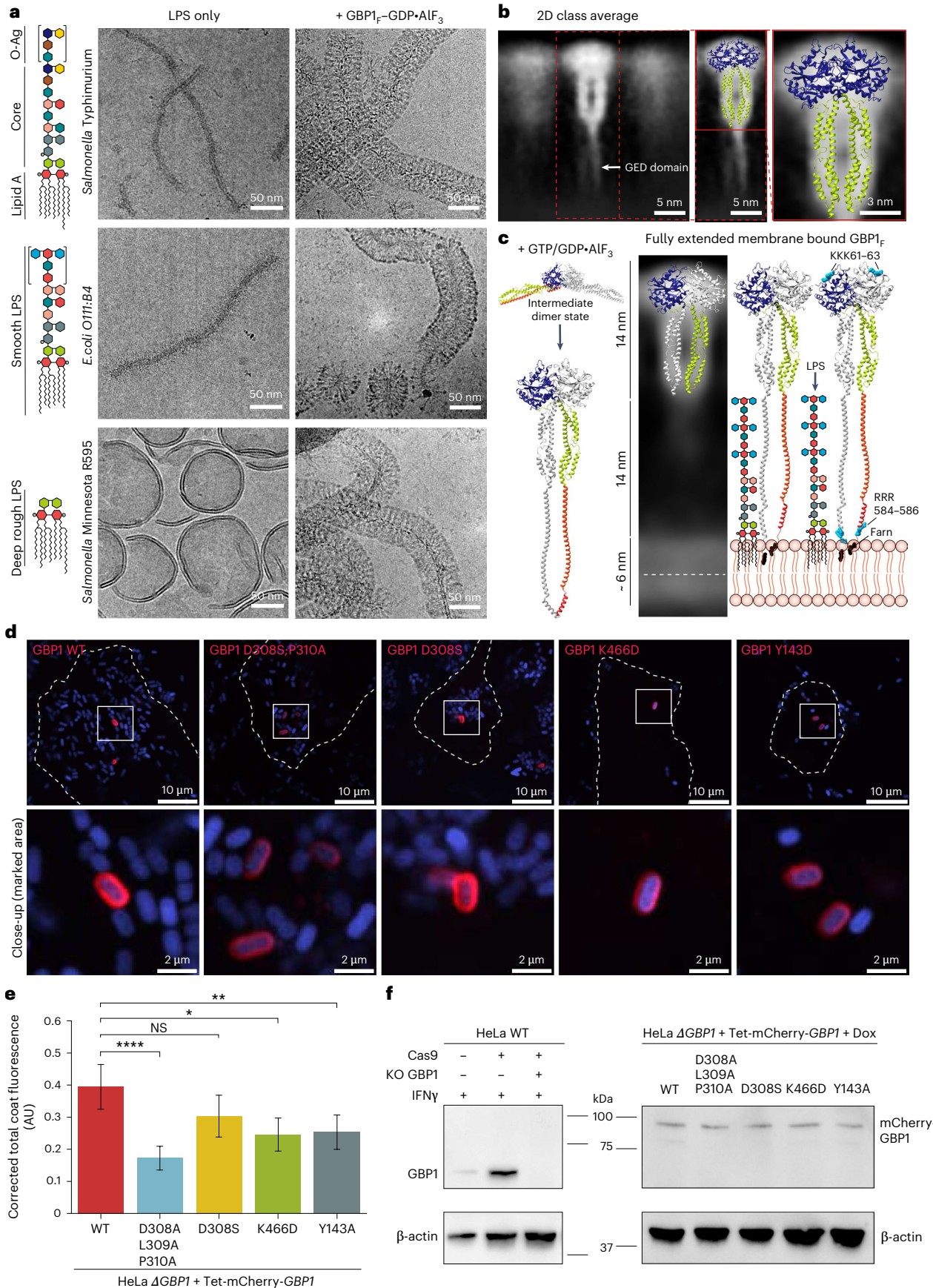

**Fig. 7 | GBP1_F coat formation on pathogen-derived LPS and effect of MD crossover destabilization on antibacterial GBP1 coat formation in _S._ Typhimurium-infected epithelial cells. a**, Schematic representation of complex O-antigen-containing LPS-ST, LPS-EB and LPS-SM (green, 2-keto-3-deoxyoctonic acid; gray, L-glycerol-D-manno-heptose; blue-green, galactose; pink, glucose; red, 2-amino-2-deoxyglucose; light blue, colitose; brown, rhamnose; yellow, abequose; dark blue, mannose). Cryo-EM micrographs of the three types of LPS in the absence (left column) and presence (right column) of GBP1_F−GDP·AlF_3. **b**, Selected 2D class average of GBP1_F bound to LPS-EB. Left, the extended GEDs projecting toward the membrane surface are visible in the average (white arrow). The atomic model of the GDP·AlF_3-stabilized GBP1 dimer is superposed onto the projected density. **c**, Schematic of nucleotide-dependent activation of GBP1 for membrane binding. Hypothetical encounter complex for initial dimerization (based on PDB 1DG3 and PDB 2B92), formation of the crossover conformation of GBP1 dimers upon nucleotide binding with extended

GED and one-dimensional model of the GBP1 coat on membranes. The radial extension of O-antigen-containing LPS is shown for comparison. **d**, HeLa Δ*GBP1* Tet-mCherry-*GBP1* expressing WT or variant mCherry−GBP1 was infected with mCerulean3-expressing _S._ Typhimurium for 2 h. Top, representative confocal images of GBP1-coated bacteria. Bottom, close-up views of marked areas. Blue, mCerulean3-expressing _S._ Typhimurium; red: mCherry−GBP1 variants. **e**, Mean and s.d. of the normalized CTCF. The statistical significance of differences relative to WT GBP1 was determined using a two-sided Welch's _t_-test with Bonferroni correction ($n = 58$–61; GBP1 D308A;L309A;P310A, ****$P = 0.000012$; GBP1 D308S, $P = 0.11$ (nonsignificant, NS); GBP1 K466D, *$P = 0.043$; GBP1 Y143A, **$P = 0.022$). **f**, Immunoblot validation of the CRISPR-engineered HeLa Δ*GBP1* and HeLa Δ*GBP1* Tet-mCherry-*GBP1* genotypes. The Dox-induced expression of GBP1 variants showed comparable expression levels for all variants. β-actin was used as a loading control.

binding and self-assembly of GBP1. While the functional consequences for GBP1 in intracellular immunity have been firmly established, the mechanistic underpinnings of these functions remain currently unclear.

Our cryo-EM data show that the GBP1 coat consists of ordered arrays of GBP1 dimers with their α-helical MDs and GEDs protruding in parallel toward the membrane surface. The molecular envelope of the repeating unit is consistent with our high-resolution cryo-EM structure of the full-length GDP·AlF_3-stabilized GBP1 dimer, displaying a crossover arrangement of the MD and extended GEDs. We found membrane association of GBP1 to be critically dependent on the ability to form crossover dimers both in vitro and infected cells. This crossover conformation is consistent with a recent crystallographic structure of a truncated GBP5 dimer[42] and resembles that of atlastins[43], which are related but functionally different members of the dynamin superfamily. Interestingly, GBPs and atlastins appear to share a set of key structural features stabilizing this conformation: a conserved linker region that mediates the MD crossover, an extended hydrophobic interaction region that latches the MD onto the LG domain of the opposing monomer and a series of weak interactions holding together the C-terminal end of the MD. While atlastins and other dynamin-like proteins associate with membranes through specialized domains, transmembrane anchors or amphipathic helices, GBPs are unique in the requirement of isoprenylation for membrane binding. Another distinguishing feature of GBPs is the extended α-helical effector domain. While the LG and MD of the GBP1 dimer appear rigid, the GED exhibits substantial flexibility. Our data provide important clues for these specializations. Assembling a dense coat on gram-negative bacterial membranes with extended LPS oligosaccharides requires elongated, flexible elements to intercalate between the O-antigen of complex LPS cores. Intriguingly, the dimensions of the extended GED are consistent with the estimated length of LPS chains with an extended O-antigen[44], suggesting that the isoprenylated GEDs can act as flexible anchors that allow breaching the LPS permeability barrier to assemble a dense coat stabilized through interactions between neighboring dimers.

GBP1 coat formation occurs cooperatively, dependent on a critical threshold concentration. Cooperativity is a hallmark of processes that require a sharp transition in their biological response. The antimicrobial function of GBPs is induced through activation of the IFN pathway that upregulates the basal transcription levels of GBP1 up to three orders of magnitude[12,45]. Thresholded self-assembly may, thus, prevent GBP1 coat formation on endogenous membranes under homeostatic conditions, activating only during infection.

Several recent studies have linked GBP1 coat formation to the activation of the noncanonical inflammasome pathway, involving the recruitment of caspase 4 to the GBP coat and induction of inflammatory cell death (pyroptosis)[16,20,21]. Pyroptosis is dependent on the cleavage of gasdermin D by caspase 4, which in turn is activated by binding to the lipid A component of LPS[30,46]. Caspase 4-dependent pyroptosis is abrogated in the absence of GBP1, suggesting that caspase 4 cannot

bind lipid A on bacterial outer membranes by itself. How does GBP1 facilitate access to lipid A components? Our data show that high GBP1 concentrations lead to tubulation of lipid membranes and LPS micelles, indicating that GBP1 has membrane-remodeling activity. The tip of membrane tubules forms a region of maximum curvature, which could facilitate access to the membrane-embedded acyl chains of lipid A otherwise shielded by the dense oligosaccharide chains of LPS and, therefore, inaccessible to the ligand-binding caspase activation and recruitment domain of caspase 4.

Different from recent tomography data suggesting that the GBP1 coat consists of monomers[47], our cryo-EM data show GBP1 dimers as the functional unit, which is consistent with previous biochemical studies[34]. Interestingly, LPS-dependent and GBP1-dependent retrieval of caspase 4 in cellular pulldowns requires GDP-AlFx (ref. 21), supporting the functional relevance of the dimeric crossover conformation during membrane association. Local remodeling of bacterial membranes by GBP1 oligomers may, therefore, provide platforms for caspase 4 recruitment and activation and reconciles observations displaying discontinuous GBP1-dependent recruitment of caspase 4 on cytosol-invasive gram-negative bacteria[21]. Our structural data were obtained with GBP1 in an activated but nonhydrolyzing state. GTP hydrolysis likely causes further structural changes. Unlike for transition-state stabilized GBP1, we did not observe coated liposomes in the presence of GTP. Instead, we observed fragmented GBP1-decorated membranes (short filaments) or flower-like assemblies resembling but distinctly different from the micellar structure observed for GBP1_F−GDP·AlF_3 in the absence of lipids. While our present data preclude quantitative conclusions, our observations suggest that GBP1 can fragment membranes. How the GBP1 coat and structural changes during GTP hydrolysis relate to this property will be important questions for further studies. Importantly, the concentration required for membrane remodeling in the presence of GTP was increased at least eightfold compared to that for GBP1−GDP·AlF_3. Because the amount of activated GBP1 in the presence of GTP will always be lower than in the presence of GDP·AlF_3 at equimolar GBP1 concentrations, this observation is consistent with a threshold-dependent response of GBP1 activity.

In summary, our data establish nucleotide-dependent GBP1 dimers as essential for GBP coat formation and demonstrate that GBP1 uses GTP hydrolysis to remodel and fragment membranes. Further studies are needed to resolve how the GBP1 coat is stabilized, how GBP1 recruits nonprenylated GBPs and how these interactions affect coat functionality.

## Online content

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

## Methods

### Plasmid and lentiviral vector construction

**GBP1.** Codon-optimized synthetic DNA encoding human GBP1 (UniProt accession P32455) was cloned into the NcoI–NotI-linearized pETM14 vector containing an N-terminal 6xHis-tag and 3C cleavage site, yielding pETM14-GBP1.

**GBP1 variants.** Expression vectors containing GBP1 variants were generated from pETM14-GBP1 by QuikChange mutagenesis using appropriate oligos (Supplementary Table 4). Mutations were confirmed by DNA sequencing (Macrogen Europe).

**pCDFDuet-FNTA-FNTB.** A coexpression vector for farnesyl transferase (FTase) was constructed using the pCDFDuet1 (Novagen) vector backbone. FNTA inserts were PCR-amplified with AJLO-023 and AJLO-024 from pANT7-FNTA-cGST (DNASU HsCD00630808). To allow subcloning into MCS1 of pCDFDuet1, BsmBI sites compatible with NcoI and NotI overhangs were inserted at the 5′ and 3′ ends of FNTA. The BsmBI-digested FNTA fragment was cloned into NcoI–NotI-digested pCDFDuet1, yielding pCDFDuet-FNTA. For cloning of FNTB into MCS2 of pCDFDuet-FNTA, FNTB was PCR-amplified from pANT7-FNTB-cGST (DNASU HsCD00077919) using AJLO-25 and AJLO-026 (Supplementary Table 4) to create a 5′-NdeI site and a 3′-BsmBI site compatible with a XhoI overhang. An internal NdeI site in pANT7-FNTB-cGST was removed by QuikChange mutagenesis with AJLO027 and AJLO-028. The NdeI–BsmBI-digested FNTB fragment was cloned into NdeI–XhoI-digested pCDF-DuetFNTA, yielding the FTase coexpression vector pCDFDuet-FNTA-FNTB.

**pCDFDuet-His6-FNTA-FNTB.** The pJET1.2 constructs containing FNTA or FNTB were obtained by amplification of AJLD0007 or AJLD0022 (Supplementary Table 5) using AJLO-023–AJLO-026 (Supplementary Table 4) following the manufacturer's recommendations. The pCDFDuet-His6FNTA-FNTB (AJLD0063) vector was obtained by Gibson assembly. The DNA fragments originated from AJLD0052, AJLD0053 and AJLV0038 using primers AJLO-076–AJLO-078, AJLO-083 and AJLO-090–AJLO-093, respectively (Supplementary Table 4). Successful cloning was confirmed at all stages by DNA sequencing (Macrogen Europe).

**Lentiviral vectors.** GBP1 complementary DNA (cDNA) was amplified from pANT7GBP1-cGST (DNASU plasmid repository, clone HsCD00077778) and cloned into an intermediate vector to produce GBP1 D308S, K466D, Y143A and D308A;L309A;P310A mutants. Lentiviral plasmids (pLV) were constructed by eZyvec (Polyplus) using modular assembly and contained expression cassettes for the expression of the Tet-On 3G protein driven by the pCMV promoter, the expression of the mCherry–GBP1 mutants driven by the Tet-responsive pTREG promoter that binds the Tet-On 3G transactivator protein in the presence of Dox and the puromycin selection marker under the SV40 promoter (Supplementary Table 5). Lentiviruses were produced by the GIGA Viral Vectors platform (University of Liège). Briefly, Lenti-X 293T cells (Clontech, 632180) were cotransfected with pSPAX2 (Addgene), a VSV-G-encoding vector and the pLV plasmids containing the GBP1 variants. Viral supernatants were collected at 96 h after transfection, filtrated (0.2 μm) and concentrated 100× with lentivirus concentration solution (SanBio, TR30026). Virus titers were quantified by qPCR (ABM, LV900).

### Cell lines and cell culture

**HeLa Cells.** HeLa cells were obtained from the German Collection of Microorganisms and Cell Culture (DSMZ; ACC57). HeLa cells and stable cell lines were grown in RPMI-1640 medium (Gibco) supplemented with 10% FBS. Cells were cultured at 37 °C under 5% $CO_2$.

**Generation of KO cell lines.** KO of *GBP1* in Hela cells was performed using CRISPR–Cas9 by the Giga-Genome Editing platform of Liège University (Belgium). Approximately $5 \times 10^5$ cells were resuspended in Neon electroporation buffer R and electroporated using the 100-μl Neon transfection system kit, with two pulses at 1,005 mV and 35-ms width. The ribonucleoprotein complex consisting of three single guide RNAs (sgRNAs) targeting exons 1, 3 and 5 (RNAsg1 (GBP1 exon 1), GAACACTAATGGGCGACTGA; RNAsg3 (GBP1 exon 3), TCCTATGCTATTGTACACGA; RNAsg5 (GBP1 exon 5), TTGATCGGCCCGTTCACCGC; all from Synthego) and *Streptococcus pyogenes*. HiFi Cas9 Nuclease V3 (Integrated DNA Technologies) was freshly prepared immediately before transfection by mixing 15 μg of recombinant HiFi Cas9 Nuclease protein with 1 μg of each sgRNA guide in a total volume of 20 μl followed by incubation for 15 min at room temperature. Transfected cells were incubated for 48 h in prewarmed DMEM supplemented with 10% FBS before subcloning. Single-cell KO clones for GBP1 were validated by western blotting and used for the remainder of the study. The selected clone was tested negative for *Mycoplasma* using DNA staining with DAPI.

**Establishment of stable cell lines.** Monolayers of the HeLa KO GBP1 cells were transduced with the different lentiviruses described above using 5 μg ml⁻¹ Polybrene (Santa Cruz) and HEPES (10 mM, pH 7.5) in RPMI-1640 medium supplemented with 10% FBS for 24 h. Cells were selected using 2 μg ml⁻¹ puromycin and the protein expression was induced using 10 μg ml⁻¹ Dox for 24 h. Cells were tested negative for *Mycoplasma* using DNA staining with DAPI.

**Transient expression of Nb74.** Confluent HeLa cells were transfected with 1 μg of pcDXC3GMSG3-Nb74 and 1.5 μg of dummy DNA mixed in OptiMEM and PEI MAX for 4 h, then washed with RPMI-1640 supplemented with FBS 10% and incubated for 48 h before infection. For mock-transfected cells, DNA was replaced by OptiMEM.

**Bacterial strains.** *S. enterica* serovar Typhimurium was obtained from the DSMZ (DSM 19587). Initial cultures were resuspended in Luria–Bertani (LB) medium, plated on LB medium supplemented with agar (LBA) and 0.001% Congo red (RC) and grown overnight at 37 °C. Subcultures were prepared from one red colony and grown overnight at 30 °C. Bacteria were washed in 25 ml of ice-cold double-distilled water (ddH₂O), then 10 ml of ice-cold ddH₂O and 5 ml of ice-cold ddH₂O and resuspended in 250 μl of ice-cold glycerol (10% v/v). To create a fluorescent strain, *S.* Typhimurium was transformed with pFPV25.1mCerulean3 (a gift from M. Corvert; Addgene, 124904) using electroporation pulses of 2,500 V for 6 s. Bacteria were regenerated in LB before selection on LBA and 0.001% RC supplemented with ampicillin. Red colonies were used to make glycerol stocks or for the preparation of precultures for the infection.

**Infections.** *S.* Typhimurium was plated on LBA with 0.001% CR supplemented with ampicillin and grown overnight at 37 °C. Subcultures were made from a fresh red colony and grown overnight in LB medium supplemented with the appropriate antibiotics at 30 °C. For infections, bacteria were resuspended in Hanks' balanced salt solution (HBSS) to an optical density at 600 nm ($OD_{600}$) of ~2.5. Confluent HeLa cells in 24-well plates were washed three times with warm RPMI to remove Dox and were infected with 10 μl of the resuspended bacteria at 37 °C for 2 h. For mock-infected cells, HBSS buffer was added instead of the bacteria.

**Western blot analysis.** HeLa cells were lysed by scraping in Laemmli buffer (BioRad), 5% 2-mercaptoethanol and protease inhibitor cocktail (cOmplete, Roche) and incubated at 100 °C for 10 min. Samples were separated on 4–12% Bis-Tris PAGE gels (SurePage, Genscript) and transferred on PVDF membranes using the Trans-Blot Turbo Transfer system (BioRad). Membranes were blocked in PBS containing 0.1% Tween-20 supplemented with 5% BSA. Blots were incubated in primary antibody overnight at 4 °C followed by incubation in secondary antibody for 1 h at room temperature. Visualization was performed using the detection

reagent (SuperSignal West Pico PLUs Chemiluminescent Substrate, Thermo Fisher Scientific) in the ChemiDoc imaging system (BioRad). Primary antibodies used were monoclonal anti-hGBP1 (sc53857), monoclonal anti-Myc tag (MA1-980, Invitrogen) and monoclonal anti-β-actin (MA1-140, Invitrogen). Secondary antibodies used were polyclonal anti-rat IgG (112035-003 from Jackson ImmunoResearch) and polyclonal anti-mouse IgG (7076 from Cell Signaling Technology) both conjugated to horseradish peroxidase.

### Protein expression and purification

**GBP1 and GBP1 variants.** Proteins were expressed in *E. coli* BL21(DE3) (Supplementary Table 6) using autoinduction in lactose-containing medium[48]. Precultures were grown in LB medium overnight at 37 °C. For protein expression, ZYP5052 medium was inoculated at 1:50 (v/v) with preculture and cells were grown at 37 °C and 180 r.p.m. for 3–4 h before lowering the temperature to 20 °C for 15–20 h. Cells were harvested by centrifugation at 4 °C and 4,000 r.p.m. and the cell pellet was resuspended in lysis buffer (50 mM HEPES pH 7.8, 500 mM NaCl and 0.1% Triton X-100) on ice. The cells were disrupted by three successive freeze–thaw cycles. To digest genomic DNA, 1–10 µg ml$^{-1}$ DNAseI was added and incubated on a rotating wheel for 1–2 h at 4 °C. To separate cell debris, the lysate was centrifuged at 20,000$g$ for 40 min at 4 °C. The supernatant was applied to TALON (Takara) affinity resin. The bound fraction was washed with 20 column volumes (cv) of wash buffer (50 mM sodium phosphate, 300 mM NaCl and 10 mM imidazole, pH 7.4) and eluted in the same buffer containing 150 mM imidazole. The eluent was dialyzed into 3C cleavage buffer (50 mM HEPES pH 7.4, 150 mM NaCl and 0.5 mM DTT) and incubated with 1:100 (mol/mol) 3C protease overnight at 4 °C. Following cleavage, the proteins were further purified by SEC using a GE Superdex200 Increase 10/300 GL column (GE Healthcare) in running buffer (50 mM HEPES pH 7.4, 150 mM NaCl and 0.5 mM DTT).

**Farnesyl transferase.** His-FNTA-FNTB was expressed as described before for GBP1. After harvesting, the cell pellet was placed on ice and resuspended in lysis buffer (50 mM HEPES pH 7.8, 150 mM NaCl and 0.1% Triton X-100). A reduced salt concentration of 150 mM was necessary to avoid disassembly of the FNTA-FNTB. After separating the cell debris, the supernatant was applied to Ni-NTA (GE Healthcare) affinity resin. The bound fraction was washed with 20 cv of wash buffer (50 mM HEPES pH 7.8, 150 mM NaCl, 0.5 mM DTT and 1,030 mM imidazole) and eluted in the same buffer containing 250 mM imidazole. The proteins were further purified by SEC using a Superdex200 Increase 10/300 GL column (GE Healthcare) in running buffer (50 mM HEPES pH 7.4, 150 mM NaCl and 0.5 mM DTT).

**In vivo farnesylation of GBP1.** Cotranslational farnesylation of GBP1 was performed essentially as described previously[40]. *E. coli* BL21(DE3) cells were cotransformed with pETM14-GBP1 and pCDF-Duet1-FNTA-FNTB plasmids. The expression and initial purification of GBP1 were performed as described for pETM14-GBP1. To separate nonfarnesylated and GBP1$_F$, an additional hydrophobic interaction chromatography (HIC) step was performed.

Briefly, NH$_4$SO$_4$ was added to the protein solution in 3C cleavage buffer to a final concentration of 1 M. The solution was bound to a HiTrap Butyl HP column (GE Healthcare), washed with 30 cv of high-salt buffer (1.5 M NH$_4$SO$_4$, 50 mM Tris-HCl pH 8, 2 mM MgCl$_2$ and 2 mM DTT) before elution over 20 cv with a linear gradient into low-salt buffer (50 mM Tris-HCl pH 8, 2 mM MgCl$_2$ and 2 mM DTT). Fractions containing GBP1$_F$ were pooled and further purified by SEC on a Superdex200 Increase 10/300 GL column (GE Healthcare) in running buffer (50 mM HEPES pH 7.4, 150 mM NaCl and 0.5 mM DTT).

**In vitro farnesylation of GBP1.** In vitro prenylation of GBP1 was adapted from a previous study[49]. In brief, 5 µM purified GBP1 was incubated with 5 µM FTase for farnesylation and supplemented with 25 µM farnesyl

pyrophosphate (Cayman) in prenylation buffer (50 mM HEPES pH 7.2, 50 mM NaCl, 5 mM DTT, 5 mM MgCl$_2$ and 20 µM GDP). The reaction mixtures were incubated for 60 min at room temperature and dialyzed overnight at 4 °C into running buffer.

**Preparation of GDP·AlF3-stabilized GBP1 dimers.** Briefly, 15 µM GBP1 was incubated with 200 µM GDP, 10 mM NaF, 300 µM AlCl$_3$, 5 mM MgCl$_2$ and 1 mM DTT for 10 min at room temperature.

### Nanobody generation, selection and purification

**Nanobody generation.** Llamas were immunized with purified monomeric GBP1, GBP1$_F$ or GDP·AlF$_3$-stabilized dimeric GBP1. From each llama, a blood sample was taken and the peripheral blood lymphocytes were isolated followed by the purification of RNA and synthesis of cDNA. Nanobody coding sequences were then PCR-amplified and cloned into a phage display library, creating libraries with >10$^8$ independent clones.

**Nanobody selection.** For phage display selections, farnesylated, monomeric or GDP·AlF$_3$-stabilized dimeric GBP1 was solid-phase-coated in 50 mM HEPES pH 7.4, 150 mM NaCl and 0.5 mM DTT and selections were performed in the same buffer. To detect the presence of GBP1-specific nanobodies, the His-tag was detected by an anti-His monoclonal antibody followed by the addition of an anti-mouse antibody conjugated to alkaline phosphatase. As a substrate for alkaline phosphatase conjugates, 2 mg ml$^{-1}$ of 4-nitrophenyl phosphate disodium salt hexahydrate was used. In total, 78, 26 and 33 clones were found positive on the dimeric GBP1–GDP·AlF$_3$, GBP1$_F$ and monomeric GBP1, respectively. We selected nanobodies from different families and performed an SEC coupled to multiangle light scattering (MALS) analysis to investigate the binding behavior. Nb74 was chosen because it bound to GBP1 in a 1:1 ratio without breaking the GDP·AlF$_3$-stabilized GBP1 dimer apart.

**Expression and purification.** Nb74 and GFP–Nb74 were expressed in *E. coli* WK6 (su−) and BL21(DE3) cells, respectively. Precultures were grown overnight in LB medium containing 100 µg ml$^{-1}$ ampicillin (Nb74) or 30 µg ml$^{-1}$ kanamycin (GFP–Nb74), 2% glucose and 1 mM MgCl$_2$. TB medium (2.4% yeast extract, 2% tryptone, 0.4% glycerol, 17 mM KH$_2$PO$_4$ and 72 mM K$_2$HPO$_4$), supplemented with 100 µg ml$^{-1}$ ampicillin, 0.1% glucose and 2 mM MgCl$_2$, was inoculated with a 1:50 (v/v) dilution of the preculture and cells were grown at 37 °C at 190 r.p.m. Protein expression was induced with 1 mM IPTG at an OD$_{600nm}$ between 0.7 and 1.2, before lowering the temperature to 25 °C for 18 h of expression. Cells were harvested by centrifugation at 4 °C and 4,000$g$ for 20 min. For lysis by osmotic shock, a pellet of a 1-L culture (with OD$_{600}$ = 25) was resuspended with 10 ml of TES buffer (0.2 M Tris pH 8, 0.5 mM EDTA and 0.5 M sucrose) for 2 h on a rotating wheel. Next, 30 ml of TES/4 buffer (TES buffer, diluted four times in H$_2$O) was added and left on a rotating wheel for 1 h.

The resuspended cell lysate was centrifuged for 30 min at 8,000$g$ and the supernatant was kept. Approximately 1 ml of Ni-NTA agarose (Qiagen) was used for purification of the lysate resulting from 1 L of culture. Pre-equilibrated Ni-NTA agarose beads, in 50 mM sodium phosphate and 1 M NaCl (pH 7), were added to the supernatant and left to incubate on a rotating wheel for 1 h at room temperature. Following incubation, the beads were washed with 20 ml of 50 mM sodium phosphate, 1 M NaCl and 10 mM imidazole (pH 7) and protein was eluted with 2.5 ml of 50 mM sodium phosphate, 0.15 M NaCl and 0.3 M imidazole (pH 7). For Nb74, the elution fractions were dialyzed (Spectra/Por 3, 3.5-kDa cutoff) for 3 days against 50 mM HEPES and 0.15 M NaCl (pH 7.5) and subsequently concentrated (Amicon, 3-kDa cutoff) to 150–500 µM before storage at −80 °C. For GFP–Nb74, the elution fractions were pooled and concentrated (Amicon, 10-kDa cutoff). The concentrated eluate was further purified by SEC on a Superdex200 column, pre-equilibrated in 50 mM HEPES pH 7.5 and 150 mM NaCl.

After analysis by SDS–PAGE, the desired fractions were pooled and concentrated before storage at 4 °C.

## Biophysical analysis

**SEC–MALS.** The oligomerization states of GBP1 at 15 µM in the presence and absence of GTP and nucleotide analogs were estimated using analytical SEC–MALS. Purified protein samples were resolved on a Superdex 200 Increase 10/300 GL column (GE Healthcare) connected to a high-performance liquid chromatography (HPLC) unit (1260 Infinity II, Agilent) running in series with an online ultraviolet (UV) detector (1260 Infinity II VWD, Agilent), an eight-angle static light scattering detector (DAWN HELEOS 8+, Wyatt Technology) and a refractometer (Optilab T-rEX, Wyatt Technology).

For SEC–MALS measurements, proteins were diluted to a final concentration of 15 µM in SEC buffer (50 mM HEPES pH 7.4, 150 mM NaCl and 0.5 mM TCEP or DTT) with or without the GTP transition-state mimic or with 1 mM GTP, GDP or GMP and 0.5 mM GppCp, GTPγS or GppNHp and incubated for 5–10 min at room temperature before injection. On the basis of the measured Rayleigh scattering at different angles and the established differential refractive index increment of value of 0.185 ml g$^{-1}$ for proteins in solution with respect to the change in protein concentration (dn/dc), weight-averaged molar masses for each species were calculated using ASTRA software (Wyatt Technology, version 7.3.1).

**Mass photometry.** WT GBP1 and GBP1 variants were purified as described before. The data were collected on a Refeyn OneMP instrument using the AcquireMP software (versions 2.3 and 2.4). Silicon gaskets (Culture Well Reusable gaskets, Grace Biolabs) were adhered to clean cover slips (High Precision cover slips, no. 1.5, 24 × 50 mm, Marienfeld). For measurements, samples were diluted in 50 mM HEPES pH 7.5 and 150 mM NaCl to a final concentration of 15 nM. Data were acquired and analyzed using DiscoverMP (versions 2.3 and 2.4), using the smallest acquisition window and default settings.

**GTPase activity assay.** To determine the GTPase activity of GBP1, the GTPase-Glo assay (Promega) was used[50] with the protocol for intrinsic GTPase activity. Briefly, 5 µl of 5 µM GBP1 (WT or variants) in running buffer was added per well to a 384-well plate. Then, 5 µl of 2× GTP solution containing 10 µM GTP and 1 mM DTT was added to the same well. The reaction was incubated at room temperature for 60 min. Next, 10 µl of reconstituted GTPase-Glo reagent was added to the reaction and incubated for 30 min at room temperature while shaking. Finally, 20 µl of detection reagent was added and, after another incubation step of 10 min, the luminescence was measured using a microplate reader (Synergy H1, BioTek). BSA was used as a negative control and measurements were performed in triplicate.

## Liposome preparation

**SUV preparation.** First, 1 mg of BPLE (Avanti Polar Lipids) and 1,2-dioleoyl-*sn*-glycero-3-phosphocholine (DOPC, Avanti Polar Lipids), purchased as chloroform solutions, were each dried under a gentle N$_2$ stream. The resulting lipid film was further dried in a desiccator connected to a vacuum pump for 1 h. To hydrate the lipid film, 1 ml of 50 mM HEPES pH 7.5 and 150 mM NaCl was used. SUVs were prepared with an Avanti Mini Extruder (Avanti Polar Lipids) using hydrophilic polycarbonate membranes with a pore size of 0.1 µm. The solution of swollen lipid was filled into one of the syringes and monodisperse emulsions of SUVs were produced by passing this mixture through the membrane at least 11 times. The SUVs were stored at 4 °C until further use.

**GUV preparation.** Per experimental condition, 30 µl of 10 mg ml$^{-1}$ BPLE (Avanti Polar Lipids) was added to 10 µl of 0.1 mg ml$^{-1}$ Texas red 1,2-dihexadecanoyl-*sn*-glycero-3-phosphoethanolamine, triethylammonium (Texas red DHPE, Invitrogen) and 1% (v/v) DSPE–PEG

(2000)–biotin (Sigma-Aldrich). Then, 30 µl of this solution was carefully aspirated and spread onto a polyvinyl alcohol (PVA)-coated glass cover slide (5% PVA was prepared in water, dried on a 22 × 22 mm cover slide for 30 min at 50 °C), before an additional 30 min in a desiccator connected to a vacuum pump. To the dried lipid film, 250 µl of inside buffer (50 mM HEPES pH 7.5, 150 mM NaCl and 50 mM sucrose) were added and lipids were allowed to swell in the dark for 15 min with gentle shaking. The GUVs were collected and freshly used.

## Confocal microscopy

**Preparation of the imaging chamber.** Glass cover slips (22 × 40 mm) were attached, with UV resin, to a homemade predrilled piece of plexiglass to form the imaging chambers. The chambers were flushed with 2 mg ml$^{-1}$ BSA–biotin, containing 3 mol of biotin per mol of BSA (BioVision). After the removal of biotin, the chambers were washed with buffer (50 mM HEPES pH 7.5 and 150 mM NaCl) and incubated a further 5 min with 1 mg ml$^{-1}$ avidin (Thermo Fisher Scientific) before addition of the GUVs for imaging.

**Maleimide labeling of GBP1$_F$-Q577C.** Alexa Fluor 647 (Thermo Fisher Scientific) dissolved in DMSO to a final concentration of 10 mM was added dropwise to the protein until a 20× molar excess was achieved. Before addition of the fluorophore, the protein was reduced for 5 min with 0.5 mM TCEP. After addition, the sample was incubated 2 h at room temperature. Separation of the labeled protein from excess dye was performed according to the manufacturer using a desalting column (5 ml, HiTrap Desalting, Cytiva) in 50 mM HEPES pH 7.4, 150 mM NaCl and 0.5 mM DTT.

**GBP1$_F$–GDP·AlF$_3$.** The GTP transition-state mimic was prepared as described before. Briefly, 20 µl of Texas red DHPE-labeled GUVs were mixed with 5 µl of protein solution consisting of 18.5 µM GBP1$_F$–GDP·AlF$_3$ and 1.5 µM GBP1$_F$-Q577C–GDP·AlF$_3$, labeled with Alexa Fluor 647–C2-maleimide, resulting in a final protein concentration of 4 µM. The mixture was incubated at 30 °C for 30 min before imaging.

**GTP-activated GBP1.** First, 20 µl of GUVs were mixed with 5 µl of protein solution consisting of 18.5 µM GBP1$_F$ and 1.5 µM GBP1$_F$-Q577C, labeled with Alexa Fluor 647–maleimide. Then, 5 µl of 10 mM GTP was added to the well directly before imaging.

**Cell fixation and immunofluorescence staining.** HeLa cells were grown on glass cover slips before infection. Following infection, cells were washed three times in PBS and fixed in 4% paraformaldehyde (PFA) in PBS for 15 min at 37 °C. Cells were successively washed three times in PBS and twice in ddH$_2$O. Cover slips were directly mounted on microscopic slides with mounting medium Glycergel (C0563, Dako) and polymerized overnight at 4 °C. *S.* Typhimurium incubated with GBP1$_F$–GDP·AlF$_3$·GFP–Nb74 were fixed with 4% PFA on polylysine-coated cover slips for 15 min. Bacteria were washed three times in protein buffer (50 mM HEPES pH 7.4 and 150 mM NaCl) and blocked for 30 min with GBP1 buffer + 5% BSA. Indirect staining was performed using the monoclonal primary antibody anti-hGBP1 (sc53857) followed by the polyclonal secondary antibody anti-rat IgG conjugated to Alexa Fluor 555 (A-21434, Invitrogen). Fixed cells were washed in buffer followed by two washing steps in ddH$_2$O. Cover slips were mounted on microscopic slides with the mounting medium Glycergel (Dako) and polymerized overnight at 4 °C.

**In vitro analysis of GBP1 coat formation on *S.* Typhimurium.** A fresh culture of mCerulean3-expressing *S.* Typhimurium (OD$_{600}$ = 0.6) was supplemented with 7 µM GBP1$_F$–GDP·AlF$_3$ and an equimolar concentration of GFP–Nb74 and incubated for 30 min at 30 °C before immunofluorescence staining and imaging.

**Fluorescence intensity quantification and statistical analysis.** Confocal images were acquired using the same laser settings to prevent bias by fluctuations in fluorescence intensity the across different samples. Image analysis was performed in Fiji. The same threshold was applied to both channels (405 nm for mCerulean3-expressing *S.* Typhimurium and 561 nm for mCherry–GBP1). GBP1 coats were segmented by subtracting the 405-nm channel from the 561-nm channel and integrated coat intensities were calculated including correction for overlapping coat segments. Using the integrated intensities, the corrected total coat fluorescence (CTCF) was calculated as the integrated density − (area of selected coat × mean fluorescence of background readings). CTCF data for all GBP1 variants were normalized relative to WT GBP1. Statistical analysis was performed using Welch's unpaired *t*-test.

**Dual-trap bead-supported membrane transfer assay**
**Bead-supported bilayer preparation.** Lipid bilayer-coated silica beads were prepared by mixing lipids in chloroform in the desired molar ratios: 84.69 mol% DOPC (850375, Avanti), 15 mol% DOPS (840035, Avanti), 0.15 mol% 18:1 Liss Rhodamine PE (810150, Avanti) and 0.16 mol% Biotin lipids DSPE–PEG (2000)–biotin (880129, Avanti).

Lipids were dried to a thin film on the walls of a flask. After removal of residual chloroform, the flask was wrapped in aluminum foil and placed in a desiccator overnight. Lipids were resuspended in 1 ml of deionized $H_2O$ (d$H_2O$) to a final lipid concentration of 1 mg ml$^{-1}$, then resuspended for 30 min in a 37 °C water bath and subsequently subjected to three freeze–thaw cycles. The lipid solution was extruded 21× using an Avestin-LF-1 extruder with a 100-nm membrane. Next, 10 μl of the lipid solution was added to 89.5 μl of $H_2O$ containing NaCl and 0.5 μl of a 5% solid solution of 2-μm silica microparticles (Sigma-Aldrich) to a final concentration of 0.1 mM lipids and 1 mM NaCl, followed by incubation on a lab rotator at room temperature for 45 min. Finally, 30 μl of the solution was diluted into 270 μl of GBP buffer (50 mM HEPES pH 7.5, 150 mM NaCl, 5 mM MgCl$_2$ and 0.5 mM DTT). For the catching bead, 10 μl of a 200 mM NaCl solution was mixed with 1 μl of NTV-DNA 3.5 kDa and 2 μl of anti-DIG beads (QDIGP-20-2 ProSciTech), then incubated for 30 min on a lab rotator at 4 °C and finally diluted in 290 μl of GBP buffer.

**Optical trapping and confocal microscopy.** Optical trapping experiments were performed in a LUMICKS C-Trap. One bead containing the bead-supported bilayer and one uncoated bead were successively trapped in one of the LUMICKS C-Trap lasers. The bead pairs were then moved far downstream close to the upper wall of the flow chamber to ensure straight, laminar flow upon switching of the running solution. The bead pairs were flushed at least 30 s with GBP1 buffer + 1 mM GTP at 0.1 bar before dispensing sample containing GBP1 buffer + 100 μM GBP1 + 1 mM GTP into the flow chamber. A 532-nm laser operated at 8 mW was used to excite 18:1 Liss Rhodamine PE and confocal fluorescence images were acquired in a 14.15 × 3.35 μm window at 50-nm pixel size to measure Liss Rhodamine PE lipid fluorescence. We set the pixel dwell time to 0.1 ms, the line rate to 100 ms, an interframe waiting time of 10 ms and an overshoot time of 8 ms and used three preconditioning lines. These settings collectively resulted in a frame rate of 0.334 s$^{-1}$. The red channel (638 nm) was used to visualize the beads and as an internal control for potential dirt particles in the flow channel. Before adding GBP1$_F$, the beads were flushed for at least 30 s with GBP1 buffer supplemented with 1 mM GTP. The time of solute arrival at the first bead was calibrated with a fluorescent dye in separate experiments and estimated to 9 s. To determine fluorescence intensity time traces, the *z* axis profile of the interbead space and the catch bead was selected in ImageJ and the total relative fluorescence of each frame was processed in Origin. For baseline correction, the average fluorescence across 30 s before the addition of GBP1$_F$ was subtracted from each dataset, resulting in a baseline of 0 AU (arbitrary units). All curves were averaged with the 'average multiple curves' option in Origin and the resulting average time traces with the 95% confidence interval were plotted using

Python's matplotlib library. Control experiments were normalized to a baseline value of 1 AU, calculated as the mean fluorescence intensity of three frames before incubation. We then calculated the ratio relative to this baseline within frames 5–15 after incubation onset. The statistical significance of differences in interbead fluorescence was determined using the nonparametric Mann–Whitney *U*-test.

**Liposome cosedimentation assays.** WT GBP1 and GBP1 variants were in vitro farnesylated as described above. Experiments were also performed with in vivo GBP1$_F$ for comparison. The GTP transition-state mimic was prepared as described before. WT GBP1 or GBP1 variants were diluted to 2 μM and mixed with 1 mg ml$^{-1}$ BPLE SUV liposomes in SEC buffer to a final volume of 100 μl. Samples were incubated for 60 min at room temperature, followed by ultracentrifugation (Beckman Coulter Optima L-90K, rotor: 42.2 Ti) at 222,654$g$ for 20 min at 4 °C. The pellet and supernatant fractions were separated as quickly and gently as possible before analysis, whereby 15 μl of the fractions were loaded onto a 4–12% SurePAGE Bis-Tris gel for separation by SDS–PAGE.

The lanes of interest were identified and the bands were automatically detected using the Geldoc Image Lab software (version 6.1.0.07). After automatic background subtraction, the sum of intensities of the protein present in the pellet and supernatant fractions was used to determine the relative percentage of GBP1 in each fraction. The pelletation assay was performed five to seven times to compute the fractional average intensities and s.d.

**Negative-stain EM.** First, 3.5 μl of protein or lipid solution was applied onto a freshly glow-discharged carbon-coated copper mesh grid (Quantifoil). After 1 min, grids were washed twice with 12 μl of buffer (50 mM HEPES pH 7.4, 150 mM NaCl and 0.5 mM DTT) followed by staining with 3.5 μl of 2% (w/v) uranyl acetate at room temperature. At each step, excess sample, wash solution and stain were blotted with filter paper and grids were air-dried for 15 min. Grids were imaged on a JEM1400Plus TEM instrument (JEOL) operated at 120 kV and recorded on a bottom-mounted TVIPS F416 complementary metal–oxide–superconductor camera.

To observe micelle formation, 0.5–2 mg ml$^{-1}$ GBP1$_F$–GDP·AlF$_3$ or 8 mg ml$^{-1}$ GBP1$_F$ with 1 mM GTP was applied onto a carbon grid. The incubation time was 10 min at room temperature and 2 min at room temperature for GBP1$_F$–GDP·AlF$_3$ and GTP, respectively. To analyze the membrane binding of GBP1$_F$, 1 mg ml$^{-1}$ GBP1$_F$ was added together with all other components to form the GTP transition-state mimic (see above). SUVs were added in a 1:10 dilution (10 mg ml$^{-1}$, $d$ = 100 nm) and incubated for 10 min at room temperature. For filament formation to occur, samples needed to incubate overnight at 4 °C and 30 min at 30 °C or the concentration was increased to 2 mg ml$^{-1}$ following an incubation step of 10 min at room temperature.

**Single-particle imaging**
**GBP1–GDP·AlF$_3$ dataset.** A total of 3.0 μl of 0.7 mg ml$^{-1}$ GBP1–GDP·AlF$_3$ was applied to glow-discharged Quantifoil grids (QF-1.2/1.3, 300-mesh holey carbon on copper) on a Leica GP2 vitrification robot at 99% humidity and a temperature of 22 °C. The sample was blotted for 4 s from the carbon side of the grid and immediately flash-cooled in liquid ethane. Micrographs were acquired on a FEI Titan Krios (Thermo Fisher Scientific) operated at 300 kV. Images were recorded on a K2 Summit direct electron detector (Gatan) with a pixel size of 1.09 Å. Image acquisition was performed with EPU Software (Thermo Fisher Scientific) and micrographs were collected at an underfocus varying between −3.5 μm and −0.5 μm. We collected a total of 48 frames accumulating to a total exposure of 60 e$^-$ per Å$^2$. In total, 1,193 micrographs were acquired. Data acquisition parameters are summarized in Supplementary Table 2.

**GBP1–GDP·AlF$_3$–Nb74 dataset.** GBP1 was incubated in a 1:1 molar ratio with Nb74 for 70 min at room temperature, before adding GDP·AlF$_3$ and incubating for 10 min at room temperature. A total of

3.0 µl of 0.7 mg ml$^{-1}$ GBP1–GDP·AlF$_3$ bound to Nb74 was applied to glow-discharged Quantifoil grids (QF-1.2/1.3, 300-mesh holey carbon on copper) on a GP2 vitrification robot at 99% humidity and 22 °C. The sample was blotted for 4 s from the carbon side of the grid and immediately flash-cooled in liquid ethane. Micrographs were acquired on a FEI Titan Krios (Thermo Fisher Scientific) operated at 300 kV. Images were recorded on a K3 Summit direct electron detector (Gatan) at a magnification of ×105,000, corresponding to a pixel size of 0.834 Å at the specimen level. Image acquisition was performed with EPU 2.8.1 Software (Thermo Fisher Scientific) and micrographs were collected at an underfocus varying between −2.2 µm and −0.6 µm. We collected a total of 50 frames accumulating to a total electron exposure of 60 e$^-$ per Å$^2$. In total, 5,214 micrographs were acquired.

**GBP1$_F$–GDP·AlF$_3$.** A total of 3.0 µl of 1 mg ml$^{-1}$ GBP1$_F$–GDP·AlF$_3$ was applied to glow-discharged Quantifoil grids (QF-1.2/1.3, 300-mesh holey carbon on copper) on a Leica GP2 vitrification robot at 97% humidity and 20 °C. The sample was blotted for 4 s from the carbon side of the grid and immediately flash-cooled in liquid ethane. Grids were imaged on a JEM 3200FSC TEM instrument (JEOL) operated at 300 kV. Images were recorded on a K2 Summit direct electron detector (Gatan). Two different datasets were acquired: one at a magnification of ×30,000, corresponding to a pixel size of 1.22 Å at the specimen level, and the other at a magnification of ×15,000, corresponding to a pixel size of 2.449 Å. Image acquisition was performed with SerialEM[51] and micrographs were collected at an underfocus varying between −3 µm and −1 µm. We collected a total of 60 frames accumulating to a total electron exposure of 48.37 e$^-$ per Å$^2$ (for the dataset at ×30,000) or to a total electron exposure of 12.92 e$^-$ per Å$^2$ (for the dataset at ×15,000). In total, 395 (dataset at ×30,000) or 606 (dataset at ×15,000) micrographs were acquired.

**GBP1$_F$–GDP·AlF$_3$ with LPS-EB, LPS-SM or LPS-ST.** First, 1 mg ml$^{-1}$ GBP1$_F$–GDP·AlF$_3$ was mixed with 0.22 mg ml$^{-1}$ LPS-EB (InvivoGen), LPS-SM (InvivoGen) or LPS-ST (Enzo) and incubated for 30 min at 30 °C. Then, 3.0 µl of the mixture was applied on to glow-discharged Quantifoil grids (QF-1.2/1.3, 200-mesh holey carbon on copper) on a Leica GP2 vitrification robot at 98% humidity and 22 °C. The sample was blotted for 4 s from the carbon side of the grid and immediately flash-cooled in liquid ethane. Grids were imaged on a JEM 3200FSC TEM instrument (JEOL) operated at 300 kV. Images were recorded on a K2 Summit direct electron detector (Gatan) using automated image acquisition in SerialEM[51]. Data collection statistics for each dataset are summarized in Supplementary Table 4.

**Single-particle image processing**
**GBP1–GDP·AlF$_3$–Nb74.** The GBP1–GDP·AlF$_3$–Nb74 dataset was processed using cryoSPARC (version 3.3.2)[52]. The in-built patch-motion correction[53] routine in cryoSPARC was used to correct for stage drift and beam-induced specimen movement over the acquired frames. A total of 5,208 micrographs were selected for further processing and patched contrast transfer function (CTF) determination[54] was performed in cryoSPARC. Using a blob-based particle picker, 2,171,521 particles were extracted and cleaned through multiple rounds of 2D classification, each consisting of 50–100 classes. Classes only containing the LG domain of the protein were actively sorted out as they did not yield a full 3D reconstruction (Extended Data Fig. 1d). Selected 2D classes comprising 119,071 particles were used to train a Topaz model[55], which was then used to extract a total of 6,539,167 particles. Following particle extraction, four iterative rounds of 2D classification were performed and 2D class averages were selected displaying secondary-structure features. A total of 500,186 particles were used to perform ab initio reconstruction to generate five different models. Three classes were selected for heterogeneous refinements without imposing symmetry or imposing C2 symmetry. A single class with 187,161 particles was selected and used

for nonuniform refinement[56] either without imposing symmetry or with imposed C2 symmetry (Extended Data Fig. 1d). Per-particle defocus and global CTF refinement improved the resolution to 3.7 Å. Local resolution was estimated in cryoSPARC[57] and visualized in ChimeraX[58]. Map sharpening was performed in cryoSPARC by applying the overall B factor estimated from Guinier plots. For flexible refinement, the final particle stack of the cryo-EM density was clipped to 256 pixels, Fourier-cropped to 96 pixels (pixel size: 2.2240 Å) and used as input for cryoSPARC (version 4.1.1) 3D flexible refinement[59] with four latent dimensions. Morphs of the density along the four dimensions of the latent space were generated to display different modes of flexibility and were displayed in ChimeraX[58].

**GBP1–GDP·AlF$_3$.** In total, 1,193 videos of GBP1–GDP·AlF$_3$ were processed in cryoSPARC (version 3.1)[52]. Patch-motion correction and patched CTF estimation were followed by manual particle picking. Those manual picks were used to train a Topaz model[55], from which 240,487 particles were extracted. After multiple rounds of 2D classification, particles assigned to classes displaying secondary structure were used as an input to perform ab initio reconstruction to generate five different models (67,197 particles). Three classes were used for heterogeneous refinement imposing C2 symmetry. A final nonuniform refinement[56] consisting of 35,715 particles resulted in a 4.9-Å-resolution structure that only covered the LG domain of GBP1.

**GBP1–GDP$_F$·AlF$_3$.** Images of GBP1–GDP$_f$·AlF$_3$ micelles were processed in cryoSPARC (version 3.1.0)[52]. Patch-motion correction and patched CTF estimation were followed by manual particle picking and 2D classification.

**GBP1–GDP$_F$·AlF$_3$ with LPS-EB.** The dataset was processed using cryoSPARC (version 3.3.2)[52]. After patch-motion correction and patched CTF estimation, the particle segments were generated from traced filaments using the cryoSPARC filament tracer. After multiple rounds of 2D classification, classes displaying clear molecular features were used as templates for the cryoSPARC template picker. Extracted particles were again subjected to multiple rounds of 2D classification.

**Atomic model building.** Atomic models of the GTPase Domain of human GBP1–GDP·AlF$_3$ (Protein Data Bank (PDB) 2B92)[37] and the C-terminal stalk (aa 320–483) of GBP1 (PDB 1DG3)[35] were rigid-body fitted into the cryo-EM density.

Manual model building was performed in Coot (0.9.5)[60] followed by real-space refinement against one of the half maps in PHENIX (1.13)[61]. The second half map was used as a test map for assessment of overfitting. Ligand geometry and restraints for GDP·AlF$_3$ were generated using the electronic ligand builder and optimization workbench (eLBOW)[62] implemented in PHENIX. Secondary-structure restraints were used throughout the refinement. A locally sharpened and filtered map was generated using the hybrid version of LocScale[39], which integrates reference-based sharpening for modeled regions[63] with generalized scattering properties of biological macromolecules for unmodeled regions approximated by pseudoatoms[39]. The atomic displacement factors of the combined model were refined using ten Refmac[64] iterations as implemented in servalcat[65] with the keywords 'refi bonly'. Before refinement, all the atomic displacement factors were set to 40 Å$^2$.

**Tomography**
**GBP1$_F$–GDP·AlF$_3$ with liposomes.** For the dataset of GBP1$_F$–GDP·AlF$_3$ together with liposomes, a total of 3.5 µl of 1 mg ml$^{-1}$ GBP1$_F$–GDP·AlF$_3$ containing freshly extruded liposomes (1 mg ml$^{-1}$ BPLE, Avanti; d = 100 nm) and 10-nm gold fiducials (1:5 (v/v)) was applied to glow-discharged Quantifoil grids (QF-1.2/1.3, 200-mesh holey carbon on copper, Quantifoil) on a Leica GP2 vitrification robot (Leica) at 98% humidity and 20 °C. The sample was blotted for 4 s from the carbon

side of the grid and immediately flash-cooled in liquid ethane. Grids were imaged on a JEM 3200FSC TEM instrument (JEOL) operated at 300 kV. Images were recorded on a K2 Summit direct electron detector (Gatan) at a magnification of ×12,000, corresponding to a pixel size of 3.075 Å at the specimen level. Image acquisition was performed with SerialEM[51] and micrographs were collected at a nominal defocus of −5 μm or −4 μm. Bidirectional tilt series were acquired from 0° to −60° and from 0° to 60° with a 2° increment. We collected tilt series of 61 micrographs each consisting of ten frames and a total electron exposure of 93.94 e⁻ per Å² for tomogram 33 and tomogram 39 (1.54 e⁻ per Å² per tilt increment). For tomogram 50, we collected a tilt series of 61 micrographs consisting of 20 frames and a total electron exposure of 100.04 e⁻ per Å² (1.54 e⁻ per Å² per tilt increment). Micrographs were motion-corrected with MotionCor2 (ref. 53) and dose-weighted according to their accumulated electron exposure[66]. CTF correction was performed using ctfphaseflip from the IMOD package[67] and the tilt series was aligned using patch tracking and reconstructed using weighted backprojection as implemented in Etomo from the IMOD package. Segmentation of lipid membranes and protein coat was performed with tomoseg as part of the EMAN2 package[68] on reconstructed tomograms binned by a factor of 2. Segmented tomograms were visualized with ChimeraX[58].

### Bioinformatic analysis
**Multiple-sequence alignment.** The sequences of hGBP1–hGBP7 (UniProt P32455, P32456, Q9H0R5, Q96PP9, Q96PP8, Q6ZN66 and Q8N8V2) were used as input for Clustal Omega[69]. The resulting sequence alignment was displayed and consensus sequences were computed in MView[70].

**Sequence conservation.** The sequence of hGBP1 (UniProt P32455) was used as input for the ConSurf Server[71] to search the UniRef90 database with HMMer[72,73] using one iteration. The resulting sequence alignment was displayed and consensus sequences were computed in MView[70]. The conservation was mapped onto the atomic model using ChimeraX[58].

### Reporting summary
Further information on research design is available in the Nature Portfolio Reporting Summary linked to this article.

## Data availability
The refined atomic model of the pseudosymmetric GBP1 dimer was deposited to the PDB under accession code 8CQB. The primary cryo-EM density and the LocScale map of the pseudosymmetric GBP1 dimer are available from the EM Data Bank (EMDB) under accession code EMD-16794. Tomogram reconstructions were deposited to the EMDB under accession codes EMD-16813, EMD-16814 and EMD-16815. Raw micrographs were deposited to the EM Public Image Archive (EMPIAR) under accession code EMPIAR-11459. Raw tilt series are available on Zenodo (https://doi.org/10.5281/zenodo.7740464)[74]. Source data are provided with this paper.

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

## Acknowledgements

We thank R. Kieffer and J. Capoulade for help with fluorescence imaging, M. Avellaneda and F. Wruck for setting up the initial optical trapping experiments and W. Evers for cryo-EM data collection. We acknowledge Instruct ERIC (PID7267), part of the European Strategy Forum on Research Infrastructures, and the Research Foundation Flanders for their support and use of resources and A. Lundqvist for technical assistance during nanobody discovery. We thank the Plateforme GIGA Viral Vectors and the Platefrome GIGA Genome Editing (Université de Liège) for help with the generation of lentiviruses and CRISPR-engineered cell lines. Cryo-EM data collection benefited from access to the Netherlands Center for Electron Nanoscopy with financial support from the Dutch Roadmap Grant NEMI (NWO.GWI.184.034.014). This work was supported by the European Research Council (ERC-StG-852880 to A.J.), the Dutch Research Council (NWO.STU.018-2.007 to A.J.) and the Kavli Institute of Nanoscience Delft.

## Author contributions

T.K., C.P. and E.G. purified proteins. T.K. performed biophysical experiments and prepared cryo-EM samples. T.K. and A.J. collected cryo-EM data. T.K., S.H. and A.J. processed cryo-EM data. T.K. and A.J. analyzed cryo-EM data and built the atomic model. C.T. generated KO cell lines, performed infection, cellular and bacterial assays and fluorescence microscopy and quantified image data. C.P., E.G. and T.K. performed mutagenesis and cosedimentation assays. T.K. and C.T. performed cryo-EM of the LPS-bound GBP1 coat. L.G. performed and L.G. and S.J. analyzed optical trapping experiments. E.P. and J.S. supported nanobody generation. T.K. and A.J. wrote the manuscript. All authors commented on the final draft. A.J. conceptualized and supervised the study.

## Competing interests

The authors declare no competing interests.

## Additional information

**Extended data** is available for this paper at https://doi.org/10.1038/s41594-024-01400-9.

**Correspondence and requests for materials** should be addressed to Arjen J. Jakobi.

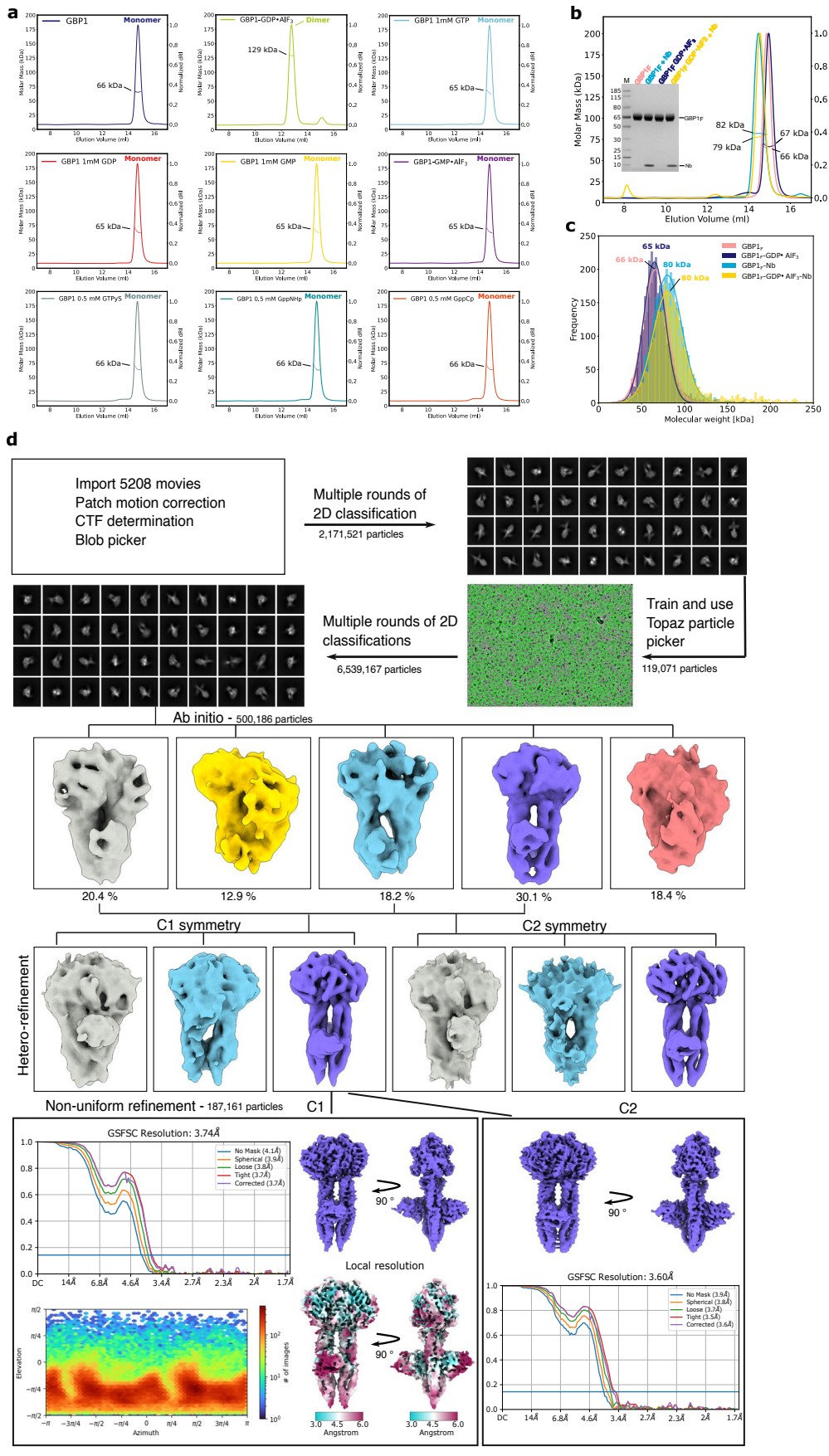

**Extended Data Fig. 1 | See next page for caption.**

**Extended Data Fig. 1 | Biophysical characterisation and structure determination of GBP1-Nb74.** (**a**) Individual SEC-MALS experiments for GBP1 with different guanine nucleotides. In the presence of GTP, GDP, GMP, GMP·AlF$_3$, GTPγS, Guanosine-5′-[(β,γ)-imido]triphosphate (GppNHp) and Guanosine-5′-[(β,γ)-methyleno]triphosphate (GppCp) GBP1 appears monomeric, while a GBP1 dimer peak emerges in the presence of GDP·AlF$_3$. The experimentally determined molecular weight is plotted across the chromatographic peak and is reported in kDa. (**b**) SEC-MALS experiments showing that farnesylated GBP1 (GBP1$_F$) appears primarily monomeric in the absence and in the presence of GDP·AlF$_3$ ($M_W$ = 66 kDa and 67 kDa). Nb74 binds GBP1$_F$ with 1:1 stochiometry both in the absence and in the presence of GDP·AlF$_3$. For conditions containing GDP·AlF$_3$, we frequently observed an additional peak close to the void volume of the SEC column corresponding to higher molecular weight species. The inset shows SDS-PAGE analysis of the SEC-MALS input. (**c**) Mass photometry analysis confirming that GBP1$_F$ is monomeric in the presence of GDP·AlF$_3$ and the 1:1 stochiometry of Nb74 binding to GBP1$_F$. Note that rare events corresponding to large GBP1$_F$ assemblies such as those observed by SEC-MALS may not be detected in the chosen field-of-view for the experiments shown. (**d**) Image processing and structure determination of GBP1-GDP·AlF$_3$-Nb74. The processing workflow is displayed for both the C1 reconstruction and for the reconstruction with C2 symmetry imposed.

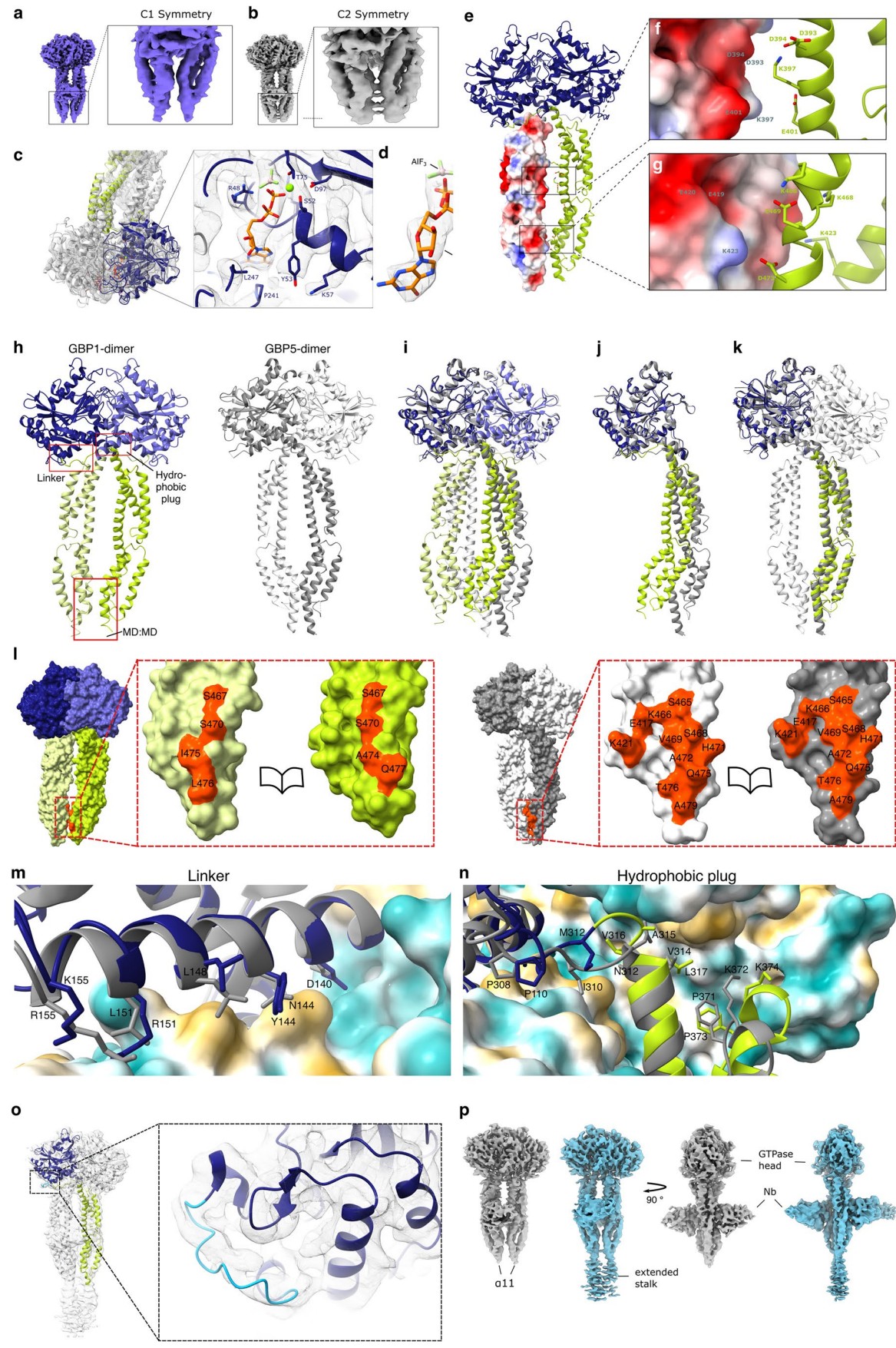

**Extended Data Fig. 2 | See next page for caption.**

**Extended Data Fig. 2 | Structural details of the GBP1-GDP·AlF₃ dimer.**
(**a**) Comparison of GBP1-GDP·AlF₃-Nb74 reconstructions without (C1) or
(**b**) with C2 symmetry imposed. Imposing strict C2 symmetry results in artefacts
at the C-terminal end of the MD. (**c**) Location of the GDP·AlF₃ ligand at the dimer
interface and close-up of the catalytic site with the ligand in stick representation
superposed onto the cryo-EM density. The catalytic arginine R48 and residues
proximal to the ligand are highlighted. (**d**) GDP·AlF₃ with corresponding density.
(**e**) Charged residues at the interface between the MD are displayed for one of the
monomers. The electrostatic potential is mapped to the surface representation
of the other monomer. The resolution of the EM density map in this region
precluded unambiguous modelling of side chains. Preferential rotamers are
shown without reference to potential interactions. (**f**) and (**g**) Close-up of
the MD interface. Residues with opposing charges locate to either side of the
interface, potentially stabilising the parallel arrangement of the MDs. (**h**) Atomic
coordinate models of the GBP1 (left) and GBP5 dimer (right; residues 1–487, PDB:

7E5A). (**i**) Overlay of both atomic models. (**j**) Comparison of monomer subunits.
The main differences are in the orientation of the MD relative to the LG domain,
with the MD in the GBP1 dimer displaying a larger twist relative to the long axis
(RMSD over all Cα atoms: 4.46 Å). (**k**) Separate alignment of LG domain (residues
1–306) and MD (residues 317–483) allows improvement of the fit, suggesting the
main determinant of the differential twists is the cross-over linker. (**l**) Close-up of
the dimer interfaces at the base of the MD. Likely interacting residues from PISA
analysis are shown. Map quality in this area precludes modelling of side chain
conformations. (**m**) Close-up of superposed cross-over linkers for GBP1 (dark
blue) and GBP5 (grey). (**n**) Close-up of the hydrophobic plug of GBP1 (dark blue
and light green) and GBP5 (grey). Putative residues stabilising the cross-over
arrangement are highlighted. (**o**) Locally sharpened density maps (LocScale2[39])
reveal additional density protruding from helix α11 consistent with a flexible GED.
(**p**) The LocScale map also allowed tracing the α3-α3′ loop (residues
156 to 167; light blue).

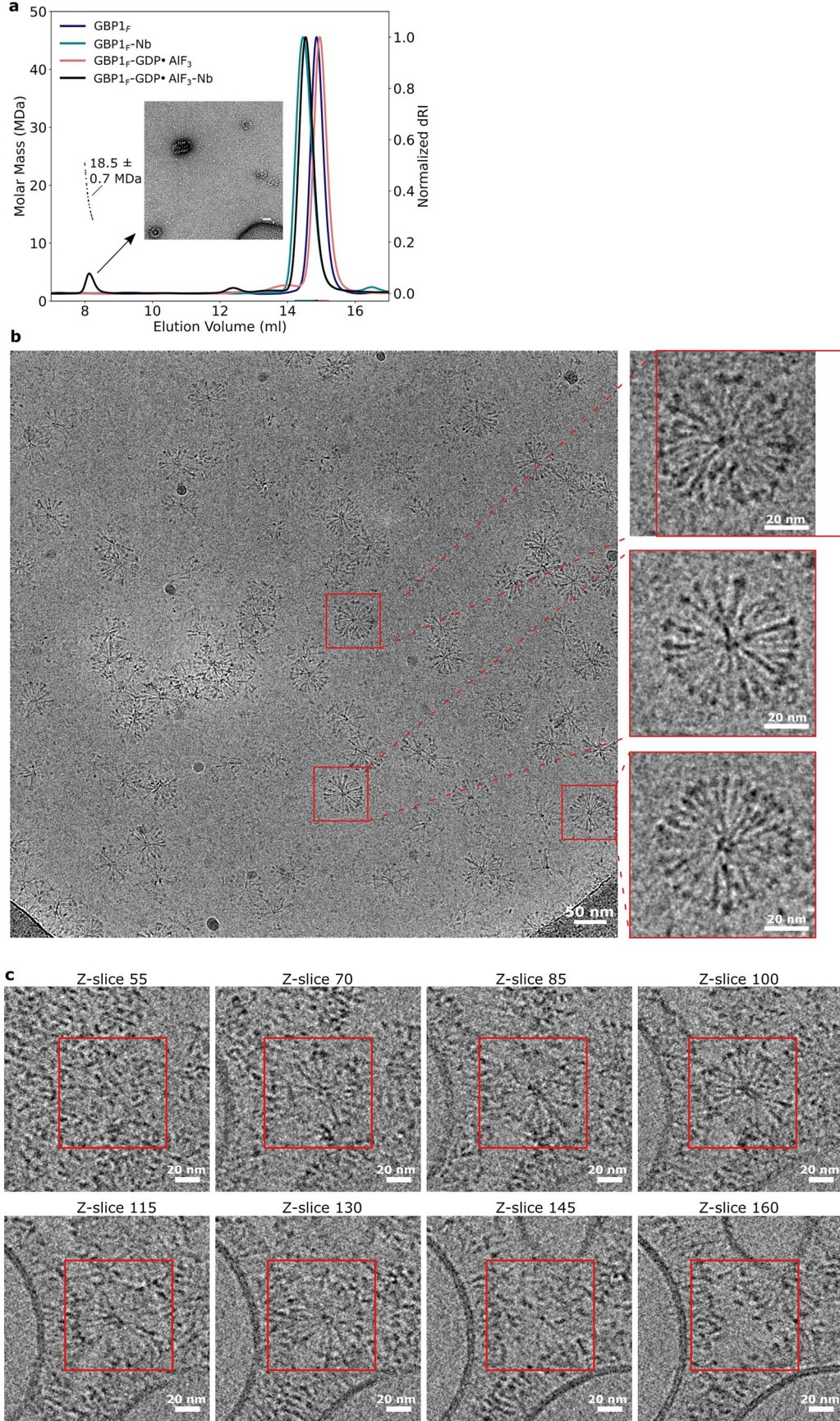

**Extended Data Fig. 3 | See next page for caption.**

**Extended Data Fig. 3 | Micellar assemblies formed by GBP1$_F$ in the presence of GDP·AlF3. (a)** SEC-MALS profile of GBP1$_F$ showing a high-molecular weight peak associated with the formation of micellar GBP1$_F$ assemblies. Inset: negative stain micrograph of peak fraction). **(b)** Representative cryo-EM micrograph of GBP1$_F$ in the presence of GDP·AlF$_3$. In the absence of lipids, GBP1$_F$ oligomerises into flower-like assemblies. **(c)** Tomographic z-stack of spherical, micellar GBP1$_F$ assemblies. Slices through the z-stack show the varying diameter of GBP1$_F$-GDP·AlF$_3$ assemblies consistent with a spherical geometry. Associated movie: Supplementary Movie 3.

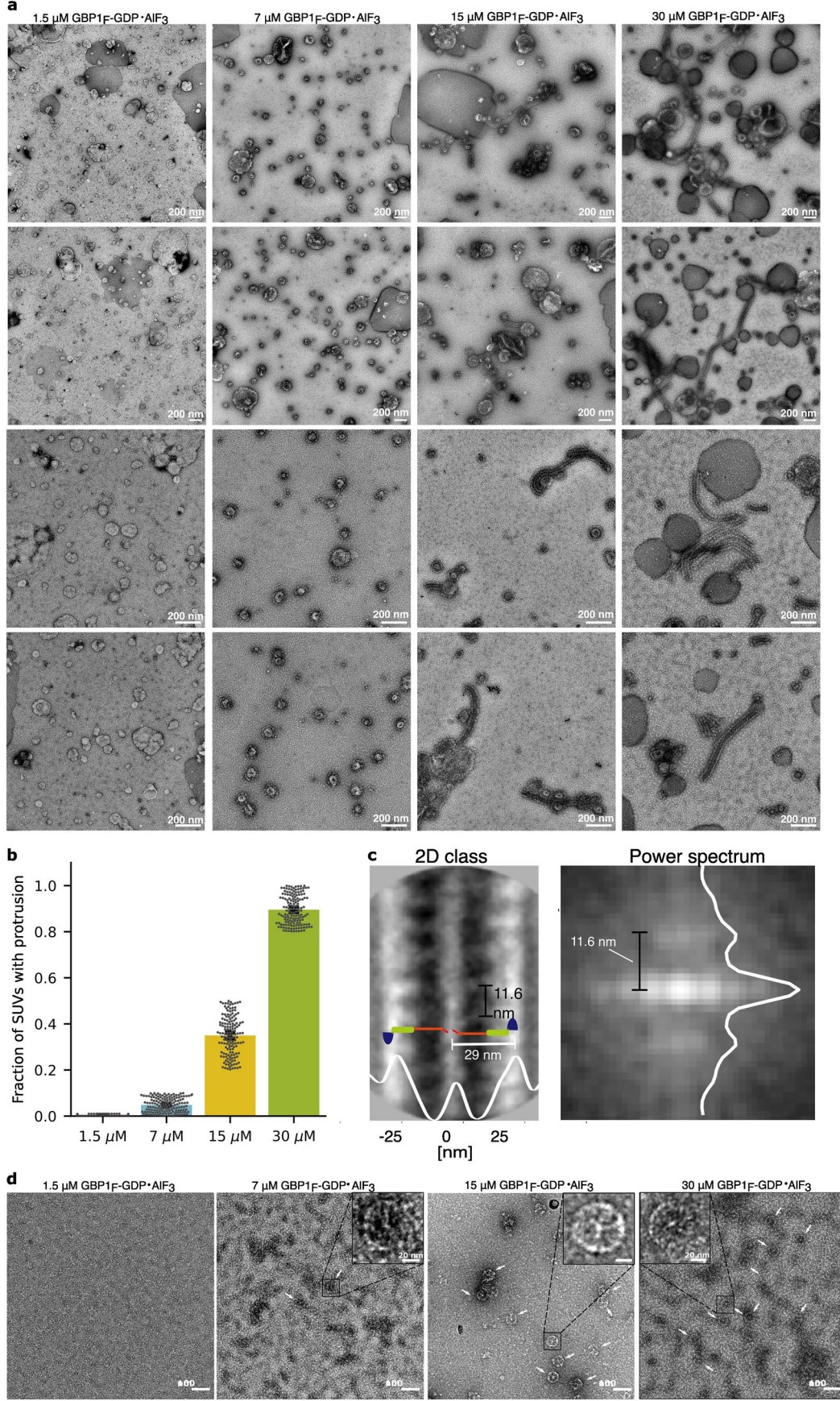

**Extended Data Fig. 4 | See next page for caption.**

**Extended Data Fig. 4 | Threshold-dependent formation of filamentous GBP1$_F$-GDP·AlF3 protrusions.** (**a**) GBP1 coatomer formation is concentration dependent. A visible coat starts to form at GBP1$_F$ concentrations of 7 µM. At a concentration of 15 µM and higher, GBP1-coated tubular protrusions become evident and are the dominant structures at concentrations exceeding 30 µM. (**b**) Quantification of GBP1-coated protrusions forming on coated SUVs. For each condition, 100 SUVs were randomly selected, classified based on the presence or absence of GBP1-coated protrusions and the fraction of protrusion-forming SUVs was calculated. Mean fractions are displayed; error bars represent a 95% confidence interval from non-parametric bootstrapping. (**c**) 2D class average of negatively stained GBP1$_F$-GDP·AlF3 tubular protrusion (left panel). The computed power spectrum shows a principal layer line at 0.086 Å-1, corresponding to a periodicity of 11.6 nm along the filament axis (right panel). (**d**) Formation of micellar assemblies of GBP1$_F$-GDP·AlF3 (white arrows) starts beyond a threshold concentration of around 7 µM.

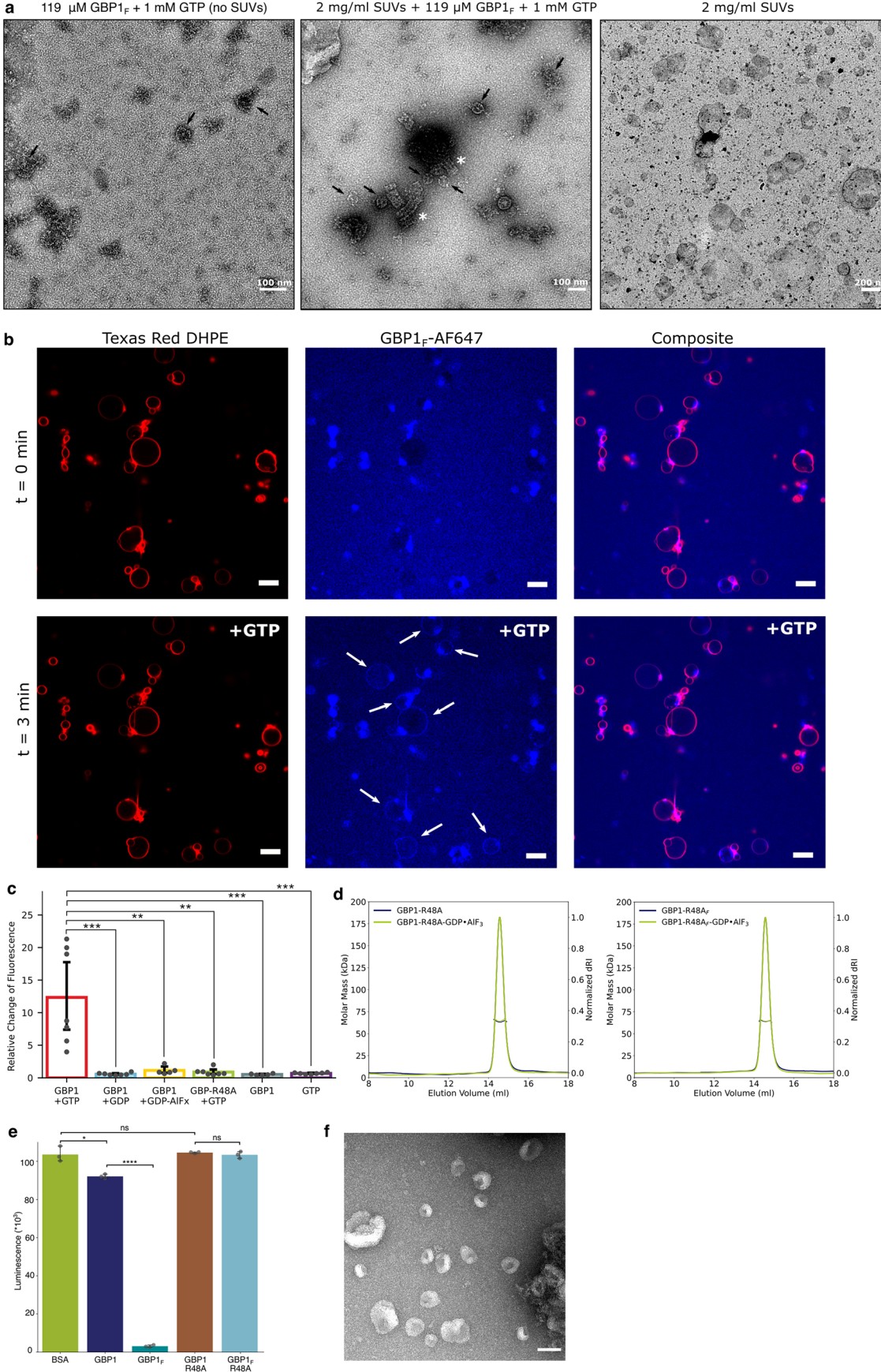

**Extended Data Fig. 5 | See next page for caption.**

**Extended Data Fig. 5 | GTP-induced assembly of GBP1$_F$ micelles and membrane fragmentation.** (**a**) Micelle formation of GBP1$_F$ (black arrows) was observed after addition of 1 mM GTP to a highly concentrated GBP1$_F$ solution (119 μM) in the absence of lipids (left panel). Upon addition of 2 mg/ml SUVs, GBP1$_F$ remodelled SUVs into spherical micelles (middle panel; arrows) and short filaments (white asterisks). A sample containing 2 mg/ml of SUVs without GBP1$_F$ is shown for comparison (right panel). (**b**) Confocal fluorescence imaging of GBP1$_F$ on GUVs. GBP1$_F$-Q577C-AF647 shows weak binding to Texas Red-DHPE labelled GUVs 3 min after the addition of 1 mM GTP (white arrows). Scale bars correspond to 5 μm. (**c**) Dual trap bead-supported membrane transfer assay. Shown are mean fold-changes in inter-bead fluorescence measured in the dual trap membrane transfer assay for GBP1 in the presence/absence of different guanosine nucleotides and for the GBP1-R48A mutant deficient in GTPase activity. Error bars represent standard deviations and significance levels were determined using a one-sided non-parametric Mann-Whitney U test (*P < 0.05, **P < 0.01, ***P < 0.001. Individual P values (all relative to GBP1-GTP): GBP1-GDP P = 0.00058; GBP1-GDPAlF$_3$ P = 0.0025; GBP1-R48A P = 0.00054; GBP1 P = 0.0026; GTP P = 0.0021). (**d**) SEC-MALS profiles for non-farnesylated GBP1-R48A (left panel) and farnesylated (right panel) GBP1$_F$-R48A showing that deficiency in GTPase activity prevents dimerisation. (**e**) GTPase activity assay for the GTPase activity-deficient GBP1-R48A and GBP1$_F$-R48A. GTPase activity for wild-type GBP1 is shown for comparison. Low luminescence signal corresponds to high GTPase activity. Data are presented as mean values +/− standard deviation (n = 3). Statistical significance was determined using two-sided Welch's t-test with Bonferroni correction (*P = 0.009, ****P = 2.7*10$^{-8}$, ns = not significant). Bovine serum albumin (BSA) was used as a negative control. (**f**) Representative negatively stained micrograph of GBP1$_F$-R48A in the presence of BPLE-SUVs. No coated SUVs could be observed.

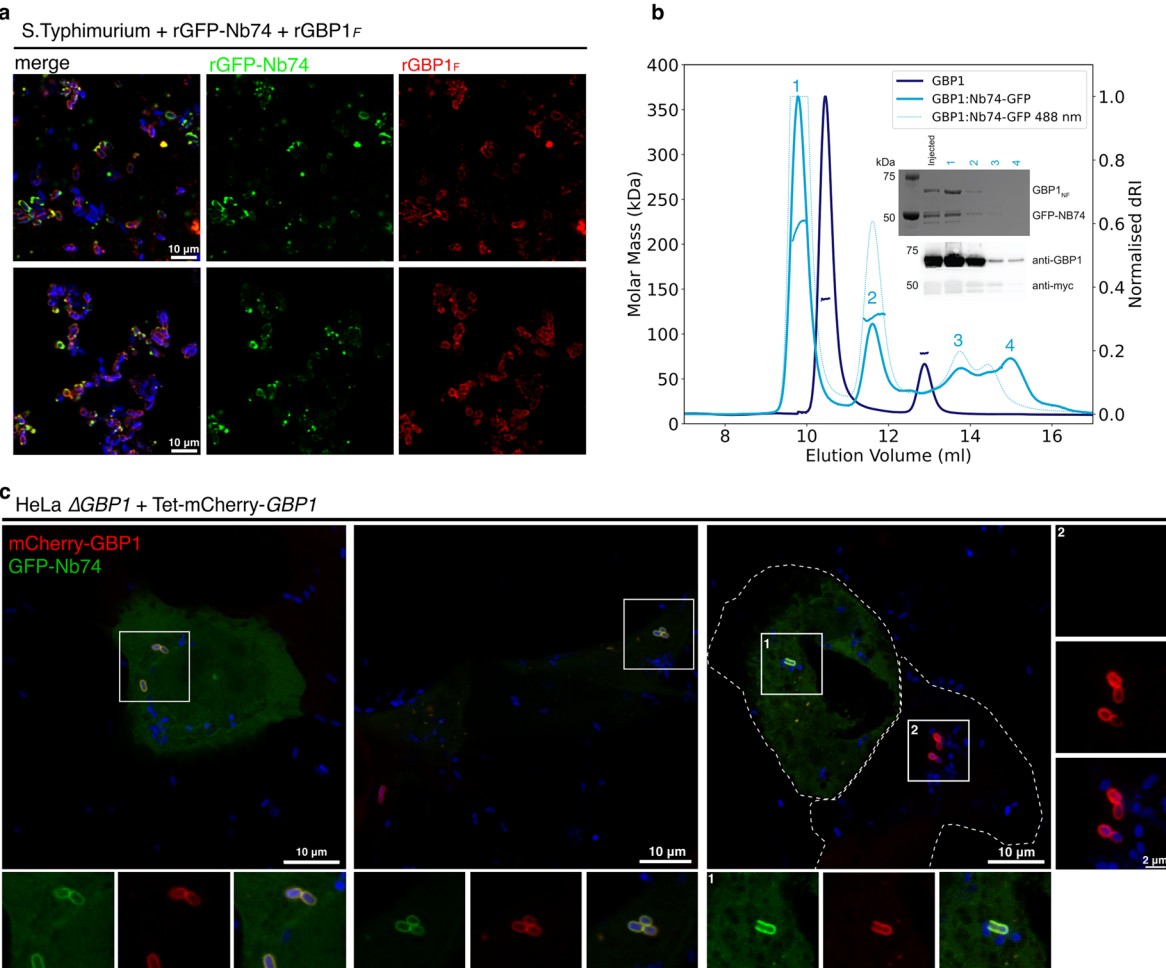

**Extended Data Fig. 6 | Confocal imaging of GFP-Nb74 on GBP1 coat formation on gram-negative bacteria in vitro and in infected cells. (a)** Confocal fluorescence images of GFP-Nb74 co-incubated with GBP1$_F$ on immobilised S. Typhimurium expressing mCerulean3. Immunostained for GBP1. **(b)** SEC-MALS profile of GFP-Nb74 in complex with GBP1$_F$-GDP·AlF3. Insets show SDS-PAGE and western blot stained for GBP1 and c-Myc for the indicated peak fractions. **(c)** Confocal fluorescence images of GFP-myc-Nb74 on GBP1-coated S. Typhimurium-mCerulean3 in infected HeLa ΔGBP1 + Tet-mCherry-GBP1 cells induced with doxycycline (dox). Split-channel images are close-ups of marked areas.

**a**

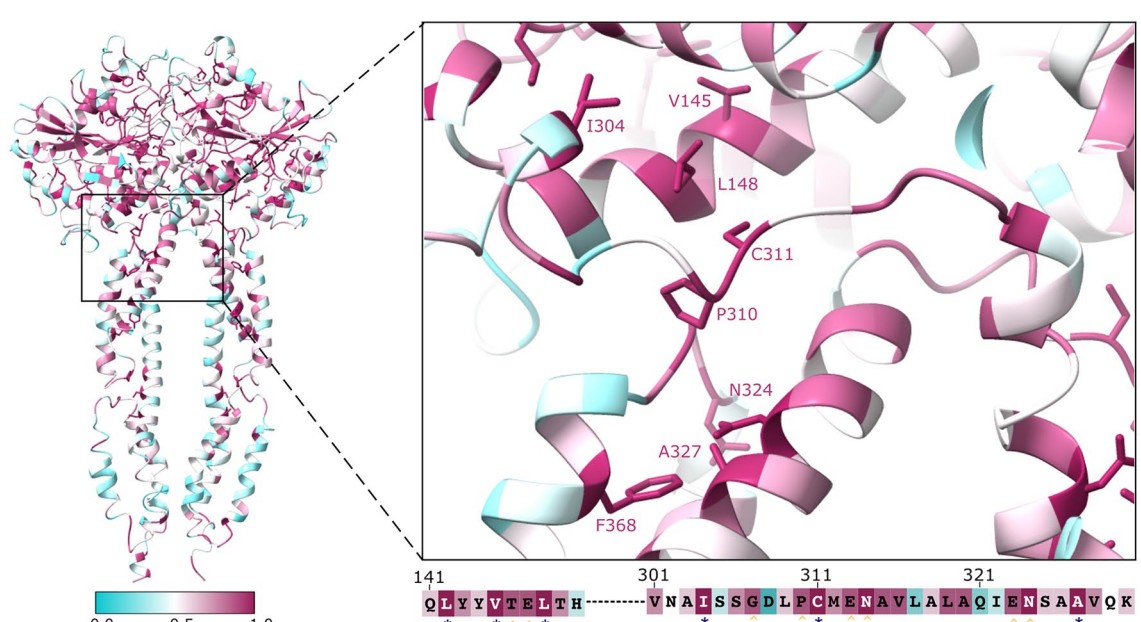

**b**

Extended Data Fig. 7 | See next page for caption.

**Extended Data Fig. 7 | Sequence conservation across human GBPs.**
(**a**) Multiple sequence alignment (MSA) of human GBP1-GBP7. Primary sequences of GBP1 - GBP7 (UniProt: P32455, P32456, Q9H0R5, Q96PP9, Q96PP8, Q6ZN66, Q8N8V2) were used as input for Clustal Omega[69]. Secondary structure elements for GBP1[35] are displayed for guidance. The colour of the alpha-helices and beta-sheets correspond to the domain architecture of GBP1 shown in Fig. 1a and used throughout the main text (blue: Large GTPase domain, green: Middle domain (MD), orange/red: GTPase effector domain). Residues are coloured by physicochemical property of the side chain (grey: hydrophobic, light blue: polar, red: negatively charged, dark blue: positively charged, yellow: aromatic, green:

special cases). The consensus sequence (100%) is shown below the alignment together with conserved physicochemical classes (l: aliphatic, a: aromatic, c: charged, h: hydrophobic, −: negative, p: polar, +: positive, s: small, u: tiny, t: turn-like). (**b**) Sequence conservation mapped onto the structure of the GBP1-GDP·AlF3-dimer. Highly conserved regions are displayed in magenta whereas less conserved areas are shown in cyan. Residues with a conservation higher than 95 % are shown in stick representation. Right panel: Close-up of the cross-over linker region. The primary sequence of highly conserved stretches is displayed below (blue asterisk: highly conserved and buried residue, yellow circumflex: highly conserved and exposed residue).

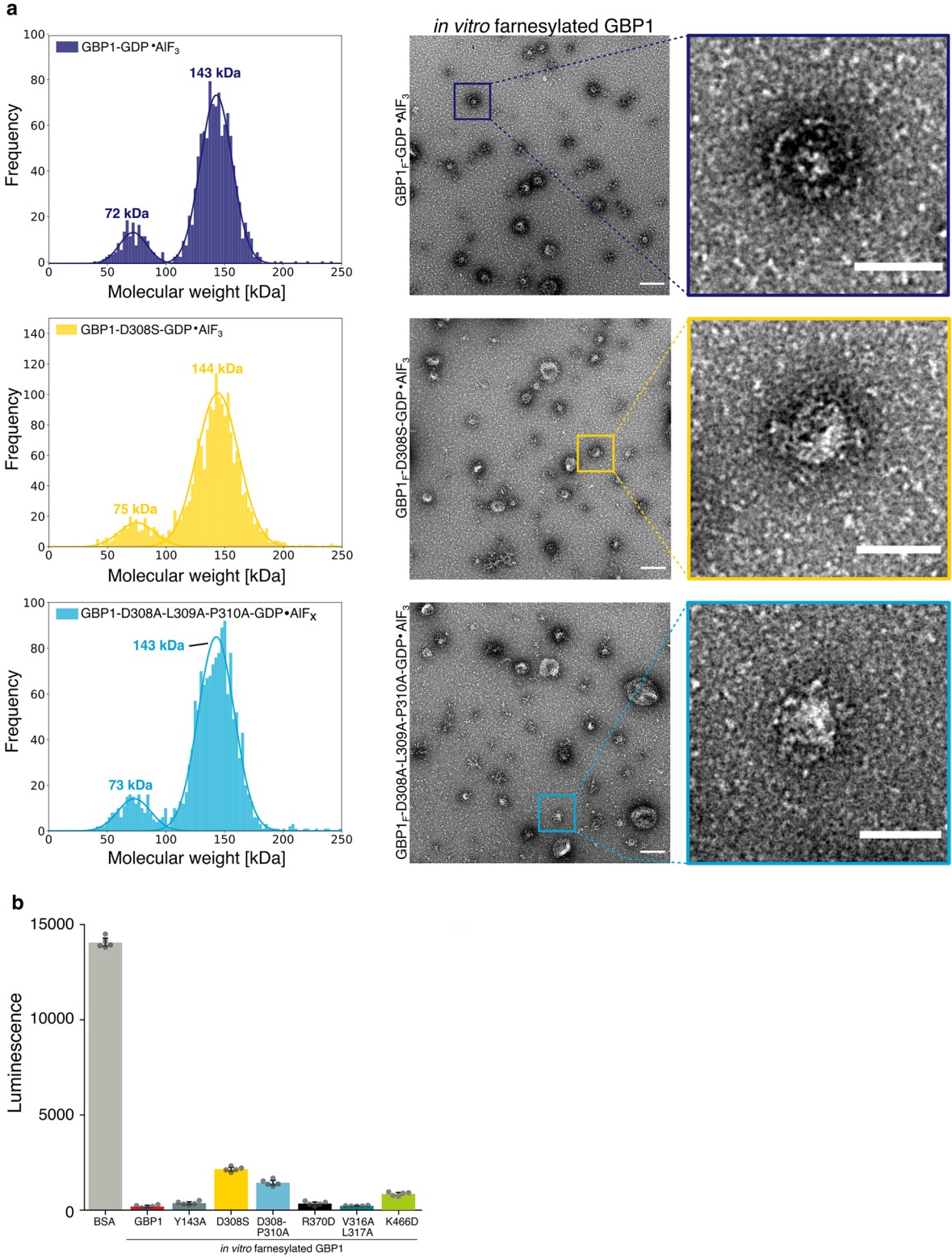

**Extended Data Fig. 8 | Effect of GBP1 variants on GTPase activity, dimerisation and membrane binding.** (**a**) Left panel: Individual mass photometry spectra revealing that variants D308S and D308A-L309A-P310A do not influence the ability of GBP1 to form dimers. Right panel: Negative stain EM of GBP1$_F$, GBP1$_F$-D308S and GBP1$_F$-D308A-L309A-P310A bound to SUVs. A reduction in membrane binding for GBP1$_F$-D308S and GBP1$_F$-D308A-L309A-P310A is observed, but the capability to bind to membranes is not entirely lost. Scale bars correspond to 200 nm in the left column and 100 nm in the right column. (**b**) GTPase activity assay of GBP1. GTPase activity of non-farnesylated GBP1 and in vitro farnesylated GBP1 WT and GBP1 variants was determined using the GTPase-Glo™ Assay (Promega). Low luminescence signal corresponds to high GTPase activity. Data are presented as mean values +/− standard deviation (n = 5). Statistical significance was determined using two-sided Welch's t-test with Bonferroni correction (*P = 0.009, **** P = 2.7*10-8, ns = not significant). Bovine serum albumin (BSA) was used as a negative control.

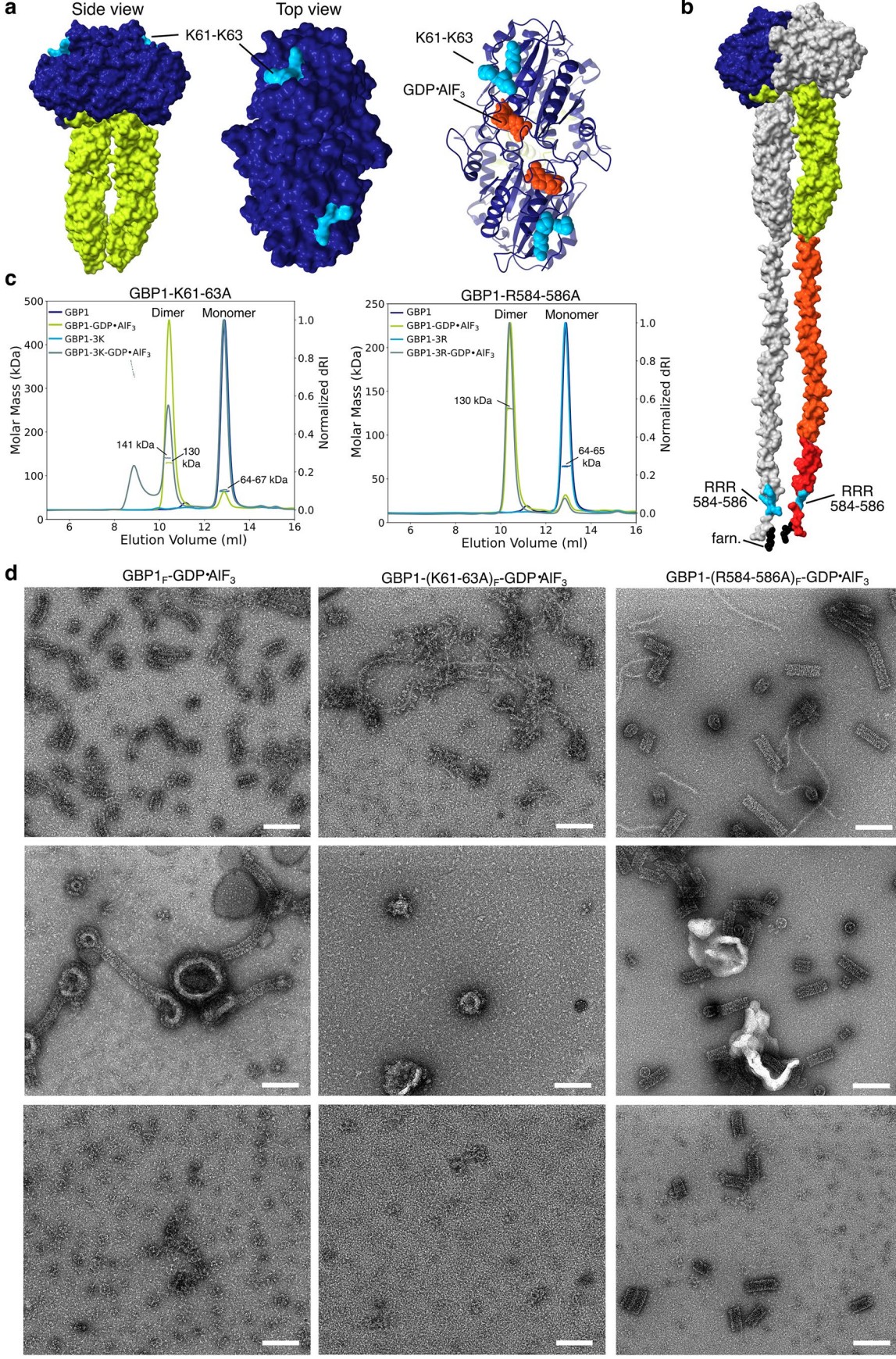

**Extended Data Fig. 9 | See next page for caption.**

**Extended Data Fig. 9 | Effect of GBP1 polybasic motif mutants of GBP1 on membrane and LPS binding.** (**a**) Surface representation of the GBP1 structure highlighting the location of the polybasic motif K61-K63. (**b**) Surface representation of the GBP1 structure with model of outstretched GED highlighting the location of the polybasic motif R584-R586. (**c**) SEC-MALS profiles of GBP1$_F$ K61-63A (left) and GBP1$_F$ R584-R586 (right) in the presence and absence of GDP·AlF3. Profiles for wild-type GBP1 are show for comparison. GBP1$_F$ K61-63A shows reduced dimerisation potential. (**d**) Representative negatively stained micrographs of GDP·AlF3-activated wildtype GBP1$_F$ (left column), GBP1F K61-63A (middle column) and GBP1$_F$ R584-R586 (right column) in the presence of Escherichia coli O111:B4 (LPS-EB; top), BPLE-SUVs (middle) and in the absence of lipids.

# Reporting Summary

## Statistics

For all statistical analyses, confirm that the following items are present in the figure legend, table legend, main text, or Methods section.

| n/a | Confirmed | |
|---|---|---|
| ☐ | ☒ | The exact sample size (*n*) for each experimental group/condition, given as a discrete number and unit of measurement |
| ☐ | ☒ | A statement on whether measurements were taken from distinct samples or whether the same sample was measured repeatedly |
| ☐ | ☒ | The statistical test(s) used AND whether they are one- or two-sided *Only common tests should be described solely by name; describe more complex techniques in the Methods section.* |
| ☒ | ☐ | A description of all covariates tested |
| ☒ | ☐ | A description of any assumptions or corrections, such as tests of normality and adjustment for multiple comparisons |
| ☐ | ☒ | A full description of the statistical parameters including central tendency (e.g. means) or other basic estimates (e.g. regression coefficient) AND variation (e.g. standard deviation) or associated estimates of uncertainty (e.g. confidence intervals) |
| ☐ | ☒ | For null hypothesis testing, the test statistic (e.g. *F*, *t*, *r*) with confidence intervals, effect sizes, degrees of freedom and *P* value noted *Give P values as exact values whenever suitable.* |
| ☒ | ☐ | For Bayesian analysis, information on the choice of priors and Markov chain Monte Carlo settings |
| ☒ | ☐ | For hierarchical and complex designs, identification of the appropriate level for tests and full reporting of outcomes |
| ☒ | ☐ | Estimates of effect sizes (e.g. Cohen's *d*, Pearson's *r*), indicating how they were calculated |

*Our web collection on statistics for biologists contains articles on many of the points above.*

## Software and code

Policy information about availability of computer code

| Data collection | Serial EM 4.0.0, EPU 2.8.1, AcquireMP software v2.3 and v2.4 |
|---|---|
| Data analysis | ImageJ 2.0.0, cryoSPARC v3.1.0 and v3.3.2, Phenix 1.19, Coot 1.9, LocScale v2.1.2, Refmac5, Servalcat v0.3.0, MotionCor2, IMOD 4.9.2, EMAN 2.3.1, TOPAZ v0.23, ConSurf webserver, MView webserver, HMMer v3.3.2, Image Lab 6.1.0.07, ASTRA v7.3.1, AcquireMP software v2.3 and v2.4. LocScale v2.1.2 is available on https://gitlab.tudelft.nl/aj-lab/locscale. |

For manuscripts utilizing custom algorithms or software that are central to the research but not yet described in published literature, software must be made available to editors and reviewers. We strongly encourage code deposition in a community repository (e.g. GitHub). See the Nature Portfolio guidelines for submitting code & software for further information.

## Data

Policy information about availability of data

All manuscripts must include a data availability statement. This statement should provide the following information, where applicable:

- Accession codes, unique identifiers, or web links for publicly available datasets
- A description of any restrictions on data availability
- For clinical datasets or third party data, please ensure that the statement adheres to our policy

The refined atomic model of the pseudo-symmetric GBP1 dimer has been deposited in the Protein Data Bank under accession code 8CQB. The primary cryo-EM density and the LocScale map of the pseudo-symmetric GBP1 dimer are available in the E1dg3lectron Microscopy Data Bank (EMDB) under accession code

EMD-16794. Tomogram reconstructions have been deposited at the EMDB under accession codes EMD-16813, EMD-16814 and EMD-16815. Raw micrographs have been deposited in the Electron Microscopy Public Image Archive (EMPIAR) with accession code EMPIAR-11459. Raw tomographic tilt series are available on Zenodo under https://doi.org/10.5281/zenodo.7740464. Atomic model coordinates for PDB IDs 2b92, 1dg3 and 7e5a were retrieved from the Protein Data Bank.

## Human research participants

Policy information about studies involving human research participants and Sex and Gender in Research.

| Reporting on sex and gender | n/a |
| Population characteristics | n/a |
| Recruitment | n/a |
| Ethics oversight | n/a |

Note that full information on the approval of the study protocol must also be provided in the manuscript.

# Field-specific reporting

Please select the one below that is the best fit for your research. If you are not sure, read the appropriate sections before making your selection.

☒ Life sciences  ☐ Behavioural & social sciences  ☐ Ecological, evolutionary & environmental sciences

For a reference copy of the document with all sections, see nature.com/documents/nr-reporting-summary-flat.pdf

# Life sciences study design

All studies must disclose on these points even when the disclosure is negative.

| Sample size | No statistical method was used to determine sample size. A sample size of n=3 or larger was used for all experiments according to standard practice for data validation and reproducibility |
| Data exclusions | During the analysis of EM images, micrographs with thick ice or excessive particle motion were excluded. Two-dimension (2D) and three-dimension (3D) classification of the particles was performed to exclude non-specimen related particles. For all other experiments no data was excluded during analysis. |
| Replication | All independent biological replications were successful and included. Exact numbers are included in figure legends. |
| Randomization | No experimental groups were present in this study. There was no allocation. Randomization is not relevant in this study. |
| Blinding | No blinding was required since randomized group allocation was not performed in this study. |

# Reporting for specific materials, systems and methods

We require information from authors about some types of materials, experimental systems and methods used in many studies. Here, indicate whether each material, system or method listed is relevant to your study. If you are not sure if a list item applies to your research, read the appropriate section before selecting a response.

## Materials & experimental systems

| n/a | Involved in the study |
| ☐ | ☒ Antibodies |
| ☐ | ☒ Eukaryotic cell lines |
| ☒ | ☐ Palaeontology and archaeology |
| ☒ | ☐ Animals and other organisms |
| ☒ | ☐ Clinical data |
| ☒ | ☐ Dual use research of concern |

## Methods

| n/a | Involved in the study |
| ☒ | ☐ ChIP-seq |
| ☒ | ☐ Flow cytometry |
| ☒ | ☐ MRI-based neuroimaging |

## Antibodies

| Antibodies used | rat monoclonal anti-hGBP1 (Santa Cruz Biotechnology; sc-53857); mouse monoclonal anti-c-Myc (MA1-980, Invitrogen); mouse |

| Antibodies used | monoclonal anti-beta actin (MA1-140, Invitrogen; Goat anti-rat IgG HRP (112-035-003, Jackson ImmunoResearch);  Horse anti-mouse-IgG HRP (7076, Cell Signaling Technologies) |
|---|---|
| Validation | anti-hGBP1: PMID: 18260761, PMID: 23405236<br>anti-c-Myc: This Antibody was verified by Relative expression to ensure that the antibody binds to the antigen stated. |

# Eukaryotic cell lines

Policy information about cell lines and Sex and Gender in Research

| Cell line source(s) | HeLa: DSMZ-German Collection of Microorganisms and Cell Cultures GmbH (ACC 57)<br>HeLa GBP1 KO: CRISPR/Cas9 engineered cell line from the original HeLa source<br>Hela GBP1 KO Tet-mCherry-GBP1: derived from HeLa GBP1 KO by lentiviral transduction<br>Hela GBP1 KO Tet-mCherry-GBP1-D308A/L309A/P310A: derived from HeLa GBP1 KO by lentiviral transduction<br>Hela GBP1 KO Tet-mCherry-GBP1-D308S: derived from HeLa GBP1 KO by lentiviral transduction<br>Hela GBP1 KO Tet-mCherry-GBP1-Y143A: derived from HeLa GBP1 KO by lentiviral transduction<br>Hela GBP1 KO Tet-mCherry-GBP1-K466D: derived from HeLa GBP1 KO by lentiviral transduction |
|---|---|
| Authentication | Cell lines were not authenticated. |
| Mycoplasma contamination | All cell lines tested negative for cytoplasma |
| Commonly misidentified lines<br>(See ICLAC register) | *Name any commonly misidentified cell lines used in the study and provide a rationale for their use.* |

