## [Peer Review File · Nature Structural & Molecular Biology]

Peer Review Information

Manuscript Title: Structural basis of antimicrobial membrane coat assembly by human GBP1

Corresponding author name(s): Arjen Jakobi

Reviewer Comments & Decisions:

Decision Letter, initial version:

Message: 15th May 2023

Dear Professor Jakobi,

Thank you again for submitting your manuscript "Structural basis of membrane targeting and coat assembly by human GBP1". We now have comments (below) from the 3 reviewers who evaluated your paper. In light of those reports, we remain interested in your study and would like to see your response to the comments of the referees, in the form of a revised manuscript.

You will see that while the reviewers appreciate the results, they do raise several concerns which should be addressed in a revision. Specifically, we agree with the reviewers that expanding the manuscript to provide evidence for the proposed mechanism of action of GBP1 in cellulose would significantly strengthen the manuscript. We would encourage pursuing the aspect of effector activation as well, in line with reviewer's #3 comments. The use of mutants to further the investigation of LPS binding should also be considered, as suggested by reviewer #1. We ask that you add the requested controls to the biophysical experiments and expand statistical analysis where suggested. While the investigation of flippase activity would be interesting as well, we do not consider it essential in the context of the current work.

Additionally, editorially we have some reservations regarding the use of the term 'coatome' when referring to GBP1 coat, as this might cause confusion with COPII heptamer. We would

ask that you consider an alternative nomenclature.

Please be sure to address/respond to all concerns of the referees in full in a point-by-point response and highlight all changes in the revised manuscript text file.

We appreciate the requested revisions are extensive. We thus expect to see your revised manuscript within 6 months. If you cannot send it within this time, please let us know. We will be happy to consider your revision as long as nothing similar has been accepted for publication at NSMB or published elsewhere. Should your manuscript be substantially delayed without notifying us in advance and your article is eventually published, the received date would be that of the revised, not the original, version.

Reporting Summary:

When submitting the revised version of your manuscript, please pay close attention to our [href="https://www.nature.com/nature-portfolio/editorial-policies/image-integrity">Digital Image Integrity Guidelines](https://www.nature.com/nature-portfolio/editorial-policies/image-integrity). and to the following points below:

Please note that all key data shown in the main figures as cropped gels or blots should be presented in uncropped form, with molecular weight markers. These data can be aggregated into a single supplementary figure. While these data can be displayed in a relatively informal style, they must refer back to the relevant figures. These data should be submitted with the last revision, prior to acceptance, but you may want to start putting it together at this point.

SOURCE DATA: we request that authors provide, in tabular form, all the data underlying the graphical representations used in figures. This is to further increase transparency in data reporting, as detailed in this editorial (<http://www.nature.com/nsmb/journal/v22/n10/full/nsmb.3110.html>). Spreadsheets can be submitted in excel format. Only one (1) file per figure is permitted; thus, for multi-paneled figures, the source data for each panel should be clearly labeled in the Excel file; alternately the data can be provided as multiple, clearly labeled sheets in an Excel file. When submitting files, the title field should indicate which figure the source data pertains to. We request our authors to provide source data at the revision stage, so that they are part of the peer-review process. Please also include the uncropped blots and gels in the Source data file.

We require deposition of coordinates (and, in the case of crystal structures, structure factors) into the Protein Data Bank with the designation of immediate release upon publication (HPUB). Electron microscopy-derived density maps and coordinate data must be deposited in EMDB and released upon publication. Deposition and immediate release of NMR chemical shift assignments are highly encouraged. Deposition of deep sequencing and microarray data is mandatory, and the datasets must be released prior to or upon publication. To avoid delays in publication, dataset accession numbers must be supplied with the final accepted manuscript and appropriate release dates must be indicated at the galley

proof stage. Please find the complete NRG policies on data availability at <http://www.nature.com/authors/policies/availability.html>.

Nature Structural & Molecular Biology is committed to improving transparency in authorship. As part of our efforts in this direction, we are now requesting that all authors identified as 'corresponding author' on published papers create and link their Open Researcher and Contributor Identifier (ORCID) with their account on the Manuscript Tracking System (MTS), prior to acceptance. This applies to primary research papers only. ORCID helps the scientific community achieve unambiguous attribution of all scholarly contributions. You can create and link your ORCID from the home page of the MTS by clicking on 'Modify my Springer Nature account'. For more information please visit please visit www.springernature.com/orcid.

[Redacted]

Sincerely,
Kat

Katarzyna Ciazynska
(she/her)
Associate Editor
Nature Structural & Molecular Biology
<https://orcid.org/0000-0002-9899-2428>

Referee expertise:

Referee #1: immunity, signalling

Referee #2: cryo-ET, viral infection

Referee #3: GBP1, inflammation

Reviewers' Comments:

Reviewer #1:

Remarks to the Author:

Human and mice GBPs have been shown to be an essential part of cell-autonomous immunity by restricting parasites and bacterial as well as viral pathogen. As part of anti-bacterial immunity they associate with intracellular bacteria (forming a coatomer) and promote the activation of inflammasomes, presumably by disrupting their outer membrane. Here, Kuhm and colleagues investigated the structural basis of membrane targeting and coatomer assembly of hGBP1. The authors show use cryo-EM to solve the structure of the GDP-AIF3 stabilized non-farnesylated hGBP1 dimer at 3.0 Å resolution using a nano-body that stabilizes the open-conformation. By contrast, they find that the farnesylated protein does not form dimers but micelles or coated liposomes as reported previously by Shydlovsky et al.. Interestingly, they also observed tubular structures on liposomes, but it is unclear if these contained membrane tubes in their interior. They further use Cryo-EM tomography to analyze the coatomers showing that they are mostly formed of GBP1 dimers. In line destabilizing the GBP1 dimers resulted in a reduction of GBP coat formation. Finally the authors show that besides liposomes, GBP1 can also associate with LPS micelles in a similar fashion by forming dimers that insert via their farnesylated residues into the LPS micelle.

This is a very nice study that is both timely and relevant to the community. The experiments are carefully carried out and convincing. It is well-described and understandable to biologist with no background in structural biology. I have few comments, mainly directed at better linking the structural data with the cell-based studies performed by other researchers.

Main comments:

Other labs have reported several mutants of hGBP1 that are affected in LPS binding and also coat assembly in cells, such as KKK61-63AAA (PMID: 29459437, PMID: 37012222) or R584-586A (PMID: 32510692). What is the impact of these mutations of the association with liposomes or LPS micelles? Can the authors show reduced binding? Moreover, in the dimer

structure where are these putative LPS binding motifs located and can the dimer structure explain the observed phenotypes. Reconciling the data obtained in cellulose with their structural data and liposome binding/LPS assays would strengthen the study and make it more interesting to researchers with a background in cell biology/infectious diseases.

The authors should include a closer comparison of their structure with the published structure of the hGBP2 and hGBP5 dimer. Are the interfaces similar?

The authors show that NB74 seems to disrupt the GBP layer on liposomes, but it does not seem to disrupt the hGBP1 micelles. Why is this the case? Shouldn't it disrupt the micelles as well?

Furthermore, would this nanobody interfere with GBP coat assembly in cells? and subsequent caspase-4 activation?

Concerning the protrusions shown in Fig. 4f: This part of the paper appears weaker to me. The evidence is insufficient to definitely conclude that GBP1 induce the formation of protrusions from liposomes. The authors base it on the observation that these are not found if no liposomes are present in the mix. But this does not automatically mean that a lipid tube is found in the center of these protrusion. It is just possible that the liposomes are needed to stabilize or seed such tubular structures. The diameter of the protrusions is the same as the GBP1 micelle, which would argue that no lipid tube is found in the center of the protrusion (59.8 +/- 2.4 nm vs. 58.1 +/- 1.2 nm). Moreover, as the authors show in Fig. 7, an LPS micelles covered with hGBP1 still allow to distinguish the membrane in the center and is larger in diameter (~63 nm).

The authors present a set of new mutants in Fig. 6 that should still dimerize but not associate with membranes due to disrupted MD alignment. What is the phenotype of these mutants on coating of bacteria in cells. It would strengthen the paper to provide such data from in cellulose experiments in addition to structural data.

Reviewer #2:

Remarks to the Author:

This study by Kuhm and colleagues describes a thorough structural and biophysical characterization of GBP1 as a function of GTP hydrolysis and farnesylation. The work seems technically sound and novel. The GBP1 micelles are stunning structures and likewise, the tubular membrane formations are very curious. While the study provides a deep insight into

the biophysical mechanisms that determine the formation of these structures, the functional relevance to cell-autonomous immunity is not addressed. While the quality of the structural/biophysical work certainly justifies publication in NSMB in my opinion, making the functional/immunological connection with these findings would make for a substantial improvement. As the authors identified several mutations in GBP1 that are important, one can't help but wonder how these mutations would impact the immune effector functions of GBP1. This aside, the study is a great piece of work.

Other comments:

- 1) The EM micrographs in Figure 1 and 3 are very difficult to read at the printed size. Consider rescaling.
- 2) Figure 1B shows so many overlapping SEC traces that it is hard to discern which is which.
- 3) The authors describe a procedure for in vitro and in vivo farnesylation of GBP1 but no data is presented to confirm the presence and occupancy of the modification.
- 4) In my opinion the sequence of Nb74 should be published along with this study. It would make for a more transparent and reproducible study, while simultaneously improving the impact as it could be a resource to others studying GBP1.
- 5) The epitope/paratope of Nb74 should be described in detail.
- 6) The assignment of the inner densities of GBP1 micelles in Figure 3 is somewhat problematic with the limited resolution. The use of Nb74 as a marker is very neat, but the assignment of the densities beyond that remain somewhat speculative.
- 7) Figure 4F construes a picture of GBP1-concentration dependent tubular membrane formations, but perhaps some more thorough statistics/distributions of the different types of membrane formations is warranted (beyond the single images shown).
- 8) The tubular membrane formations are attributed to farnesylated GBP1 and indeed it is shown that increasing concentrations will result in more tubular membranes. This begs the question whether the effect is specific to GBP1; would any farnesylated protein in similar concentrations induce similar effects?
- 9) For the dual-trap membrane fragmentation assay, presumably the trace prior to GBP1 addition serves as a control to demonstrate a GBP1-specific effect. In this respect, it would be good to see in the figure exactly when the GBP1 is added to the flow channel. As presented, a scenario in which a lipid coated bead remains stable until some critical amount of 'damage' is accumulated, after which it fragments rather suddenly is also plausible. I would therefore advocate for a control experiment where a donor bead is incubated in flowing buffer with GTP but without GBP1 for the same amount of time as the presented experiment *with* addition of GBP1.
- 10) In Figure 5, the effects of Nb74 on the tomographic reconstructions is attributed to steric hindrance of GBP1-GBP1 contacts. However, it is not described how Nb74 affects GTP

hydrolysis, which may also alter the behaviour of GBP1 on membranes.

11) The mass photometry data presented in Figure 6; the materials/methods indicated a wide range of used concentrations. It is not clear whether the MP data presented in figure 6 is all collected at the same concentration. If the concentrations are not equal between the different mutants, it is difficult to say that the shifts in monomer-dimer equilibrium are direct effects of the mutations.

12) The statistics on cryoEM particle picking are rather curious; more than 6.000.000 particles picked from 5.000 micrographs. This seems excessive and I wonder if that many particles physically fit in the field of view. Evidently the particle picking procedure yielded a nice reconstruction, but one wonders what the point of particle picking is when so many false positives can be tolerated in 2D classification.

Reviewer #3:

Remarks to the Author:

This study by Kuhm et al presents cutting-edge structural and molecular characterisation of hGBP1, which is the forerunner of the IFN γ -inducible Guanylate Binding Protein family. GBP1 has indispensable roles in sensing microbial infection, and recent cell biological and biochemical evidence points to its role in detecting cytosolic LPS from Gram-negative bacteria. However, the structural and molecular aspects of how GBP1 does this, especially whether it has membrane-modifying activities remain unknown. Here, the authors assess GBP1 oligomerisation on host membranes and LPS vesicles predominantly using CryoEM structures and in vitro assays. The authors find that dimerization of hGBP1 when associated with GDP-AIF3 is similar to the previously described dimerization mode of hGBP5 with a crucial crossing over of the MD helices which facilitate membrane insertion. They investigate this with mutagenesis and membrane co-sedimentation assays. Cooperative lateral assembly of hGBP1 coatomers is also dependent on the MD crossover and can lead to fragmentation of host membranes. They finally characterise the interaction between hGBP1 and three distinct LPS structures from pathogens, showing a common coatomer structure and Sm R595 LPS vesicle remodelling. In my opinion the structural aspects of this study are thorough and well performed, and contribute significant new knowledge regarding the mechanical aspects of GBP1 oligomerisation (i.e. MD crossing over and presumably lateral oligomerisation of dimers, membrane tubulation). The main limitation is that the physiological implications have not been investigated and some experiments lack appropriate controls.

COMMENTS:

1. To the best of my knowledge, this is the first evidence of a GBP modifying membranes in this way, which has been previously suggested but not demonstrated formally. This is a

major strength.

2. The first structure of GBP1 is in the presence of a nanobody. What evidence do the authors have that this is also the case in cells and is physiologically relevant?

3. The optical tweezer assay in figure 4j-l needs appropriate controls to be more informative. While agreeing in principle with previous observational data showing a lack of observed SUVs in the presence of GBP1-GTP and the presence of coated SUVs in the presence of GBP1-GDP-AIF3, this assay should also be carried out with the apo-GBP1, as well as GDP- and GDP-AIF3 associated GBP1 to rule out the possibility that GBP1 alone leads to lipid transfer in this context. Likewise, this should also be carried out with a GTPase-deficient mutation such as GBP1R48A (Praefcke et al., 2004; Li et al., 2017; Xavier et al., 2020; Fisch et al., 2021) or other mutants they have generated to show conclusively that it is the active GTP hydrolysis event that leads to fragmentation.

4. The effects of mutants in Fig 6 are relatively minor, perhaps because the in vitro assay. It would be good to validate these findings in cells – for example by introducing GBP1 mutants into GBP1-/- macrophages or epithelia and assess the effects of the newly identified residues in the cross-over region or others. Do these mutants behave similarly when binding SUV host membranes to LPS-containing bacterial membranes? Do they still translocate to bacteria? Can they still facilitate Caspase 4 activation?

5. Similarly, new mutants should be investigated with LPS OMVs in addition to the brain SUVs which are generic membranes that lack a GBP1-binding ligand, i.e., LPS. The flipping out of the α 12/13 on LPS OMVs is really interesting. Have the authors attempted mutations that would abolish the flipping out of the α -12/13 regions on LPS OMVs?

MINOR POINTS:

- I find it interesting and slightly concerning that in Fig. 1b, the commonly used and more well-understood transition state-mimicking compounds GTPyS, GMPPNP and GppCp do not induce dimerization, despite being non-hydrolysable GTP analogues to which GDP-AI3 is assumed to be similar. Could this please be acknowledged in text along with a hypothesis as to what the reason for this may be? This could inform future studies as to the limitations of these ubiquitous compounds.
- The mutants used in Fig. 6 should be more thoroughly explained in the text, as while being present in the figures, Y143A and K466D are not mentioned in the text itself. Furthermore, specific variants should be referenced in text when discussing conclusions which apply to the mutant in question, rather than describing them as 'LG-MD' or 'MD-MD' variants. Likewise, the propensity of D308A/L309A/P310A mutant variants for MD crossing should be assessed using Cryo-EM as in Fig. 1h, I, instead of presuming due to the position of the mutations. This would also inform as to whether the domain rearrangement still occurs in the absence of MD crossing.
- Fig. 3g is referenced in text, however does not exist

- The figures could be better labelled – for instance Fig 2A conformers could be showed separately to understand the change in the linker. Similarly, the ‘yellow’ in panel 2D is not clear.

Author Rebuttal to Initial comments

We would like to thank all three reviewers for the positive assessment of our work, their encouraging comments, and the constructive feedback. We provide a point-by-point response to the individual comments below.

Reviewer #1:

Human and mice GBPs have been shown to be an essential part of cell-autonomous immunity by restricting parasites and bacterial as well as viral pathogen. As part of anti-bacterial immunity they associate with intracellular bacteria (forming a coatomer) and promote the activation of inflammasomes, presumably by disrupting their outer membrane. Here, Kuhm and colleagues investigated the structural basis of membrane targeting and coatomer assembly of hGBP1. The authors show use cryo-EM to solve the structure of the GDP-AIF3 stabilized non-farnesylated hGBP1 dimer at 3.0 Å resolution using a nano-body that stabilizes the open-conformation. By contrast, they find that the farnesylated protein does not form dimers but micelles or coated liposomes as reported previously by Shydlovsky et al.. Interestingly, they also observed tubular structures on liposomes, but it is unclear if these contained membrane tubes in their interior. They further use Cryo-EM tomography to analyze the coatomers showing that they are mostly formed of GBP1 dimers. In line destabilizing the GBP1 dimers resulted in a reduction of GBP coat formation. Finally the authors show that besides liposomes, GBP1 can also associate with LPS micelles in a similar fashion by forming dimers that insert via their farnesylated residues into the LPS micelle.

This is a very nice study that is both timely and relevant to the community. The experiments are carefully carried out and convincing. It is well-described and understandable to biologist with no background in structural biology. I have few comments, mainly directed at better linking the structural data with the cell-based studies performed by other researchers.

Thank you for your appreciation of our work and the useful comments.

Main comments:

1.1) Other labs have reported several mutants of hGBP1 that are affected in LPS binding and also coat assembly in cells, such as KKK61-63AAA (PMID: 29459437, PMID: 37012222) or R584-586A (PMID: 32510692). What is the impact of these mutations of the association with liposomes or LPS micelles?

- Can the authors show reduced binding?

- Moreover, in the dimer structure where are these putative LPS binding motifs located and can the dimer structure explain the observed phenotypes.

We have generated the K61-63A and R584-586A mutants and characterised their ability to bind to LPS and SUVs using TEM imaging. These data are summarized in the new **Extended Data Fig. S21**. We find that neither of the mutants binds LPS or liposomes. To investigate the origin of these observations we first tested if both mutants still dimerise, since our other data suggest that this is a prerequisite for coat formation. As shown in **Extended Data Figure S21c**, the K61-63A mutant is severely compromised in its ability to dimerise and, in accordance with our data suggesting that GBP1 dimers are the essential units for coat formation, does only weakly associate with SUVs or LPS, does not form ordered coats and also does not self-assemble into GBP1 micelles in the absence of lipids (**Extended Data Figure S21d**). We therefore conclude that rather than specifically affecting its ability to bind LPS, the failure of this mutant to be recruited to *S. Typhimurium* in infected epithelial cells (PMID: 29459437, PMID: 37012222) is a consequence of its impaired ability to form coatomer-promoting dimers. In contrast, the R584-586A mutant dimerises equally effective as wild-type GBP1 (**Extended Data Fig. S21c**), but completely fails to associate with lipid membranes or LPS, and in the absence of lipids assembles tubular structures similar to those observed for wild-type GBP1 exclusively in the presence of lipid membranes or LPS (**Extended Data Fig. 21c, S8a & Fig. 7**). We hypothesise that, for wild-type GBP1, charge neutralisation of the polybasic motif in $\alpha 13$ extension during dense packing into tubular assemblies requires the presence of negatively charged lipids, and that the intrinsic charge neutralization in the R584-586A mutant abolishes this requirement and thus constitutively promotes polymerisation into tubular assemblies regardless of the presence of lipid membranes. These observations allow reconciling the phenotypic observations in PMID: 29459437, PMID: 37012222 & PMID: 32510692 on a mechanistic level.

We have added the following paragraph to the manuscript:

*Mutations in two polybasic motifs of GBP1 differentially affect LPS binding. GBP1 variants of two distinct polybasic motifs (K61-K63A and R584-R586A) have been recently reported to affect coat formation on intracellular *S. Typhimurium*, *Franciscella novicida* and *Shigella flexneri* (16, 20, 42). To interrogate whether*

these observations are functionally linked to formation of the outstretched cross-over conformation observed for the GBP1 dimer, we first expressed recombinant GBP1_F^{K61-63A} and GBP1_F^{R584-586A} and tested their ability to form stable dimers in the presence of GDP·AIF3. The K61-K63 motif is located in the α 1- β 2 loop in close vicinity to the LG dimerisation interface [Extended Data Fig. S21a] and stabilises the conformation of the β 6- α 5 loop involved in nucleotide coordination. The GDP·AIF3 activated GBP1_F^{K61-63A} mutant showed a markedly reduced dimer fraction relative to wild-type GBP1. Disrupting this motif thus appears to impair the ability to form nucleotide-induced GBP1 dimers [Extended Data Fig. S21c]. In contrast, the R584-R586 motif is located near the end of α 13 within the extended GED directly adjacent to the farnesylation site. For this mutant the nucleotide-dependent dimerisation propensity is not affected [Extended Data Fig. S21b,c]. We next tested whether the GBP1_F^{K61-63A} and GBP1_F^{R584-586A} affect the ability of coat formation and membrane remodelling on liposomes and LPS. We incubated nucleotide-activated GBP1_F^{K61-63A} and GBP1_F^{R584-586A} with BPLE SUVs or micelles of smooth LPS from *Escherichia coli* O111:B4 (LPS-EB). Unlike wild-type GBP1_F, which efficiently coats and remodels SUVs and LPS, both GBP1_F^{K61-63A} only sparsely decorates LPS and SUVs, and does not assemble into the characteristic micellar assemblies observed for wild-type GBP1_F in the absence of lipids [Extended Data Fig. S21d]. GBP1_F^{R584-586A} entirely failed to associate with either liposomes or LPS. Instead, GBP1_F^{R584-586A} polymerises into elongated tubular assemblies even in the absence of lipids [Extended Data Fig. S21d]. In contrast, for wild-type GBP1_F such tubular structures are exclusively observed in the presence of lipids [Fig. 4, Extended Data Fig. S10], suggesting that filamentous packing of highly positively charged α -13 into dense arrays of tubular assemblies may be facilitated by integration of negatively charged lipids such as Lipid A, while for GBP1_F^{R584-586A} the charge-neutralising mutations render this requirement dispensable and promote constitutive polymerisation and disfavour association with membranes. Collectively, these data allow reconciling previous phenotypic observations for these mutants (16, 20, 42) and provide disparate mechanistic explanations for the abolished localisation to cytosolic bacteria.

Extended Data Fig. S21. | Effect of GBP1 polybasic motif mutants of GBP1 on membrane and LPS binding. (a) Surface representation of the GBP1 structure highlighting the location of the polybasic motif K61-K63 (GBP1_F^{K61-63A}). (b) Surface representation of the GBP1 structure with model of outstretched GED highlighting the location of the polybasic motif R584-R586 (GBP1_F^{R584-586A}). (c) SEC-MALS profiles of GBP1_F^{K61-63A} (left) and GBP1_F^{R584-586A} (right) in the presence and absence of GDP·AIF3. Profiles for wild-type GBP1 are shown for comparison. GBP1_F^{K61-63A} shows reduced dimerization potential. (d) Representative negatively stained micrographs of GDP·AIF3-activated wild-type GBP1_F (left column), GBP1_F^{K61-63A} (middle column) and GBP1_F^{R584-586A} (right column) in the presence of *Escherichia coli* O111:B4 (LPS-EB; top), BPLE-SUVs (middle) and in the absence of lipids.

- Reconciling the data obtained in *cellulo* with their structural data and liposome binding/LPS assays would strengthen the study and make it more interesting to researchers with a background in cell biology/infectious diseases.

To address this question, we have we have generated CRISPR/Cas9-engineered HeLa GBP1 KO cells stably expressing mCherry-GBP1 variants under a Tet-inducible promotor (HeLa Δ GBP1 + Tet-mCherry-GBP1) and have tested their effect on GBP1 coat in *S. Typhimurium* infected cells. Our results of these cellular infection assays replicate our observations of liposome/LPS assays obtained *in vitro* and support the critical importance of the cross-over dimer conformation for coat formation in a cellular infection model. Please see our response to **point 1.6** for details.

1.2) The authors should include a closer comparison of their structure with the published structure of the hGBP2 and hGBP5 dimer. Are the interfaces similar?

We have now included a comparison to the GBP5 dimer structure (PDB ID 7E5A) in **Extended Data Fig. S6**. To the best of our knowledge there is no dimer structure of GBP2 including the MD. The only published structure of GBP2 are all monomeric (PDB IDs 7m1s, 7e58, 6vkj). We find that the overall configuration of dimeric GBP1 and GBP5 dimer are similar but differ in their arrangement of the parallel protruding stalks and the way the MDs contact at the peripheral regions. We do note that the GBP5 dimer crystal structure was obtained using a construct truncated after the MD, whereas our cryo-EM structure of the GDP·AIF3-stabilised dimer was obtained using full-length GBP1. Our previous flexibility analysis of our cryo-EM data also revealed that the GED of GBP1 is disordered, and the MD showed flexibility, which together likely account for part of the differences that we observe (Fig. 2e, Extended Data Fig. S7, Extended Data Movie SM1).

Extended Data Fig. S6. | Comparative structural analysis of GBP1 and GBP5 dimers (a) Atomic coordinate model in cartoon representation of GDP·AIF3 activated GBP1 dimer and Gbp5-dimer (residues 1 – 487, PDB-ID: 7E5A). (b) Overlay of both atomic models visualized in (a). (c) Comparison of a single monomer from (b). The main differences are in the orientation of the MD relative to the LG domain, with the MD stalk of the GBP1 dimer displaying a larger twist relative to the long axis of the dimer. The overall RMSD calculated over all Ca atoms was 4.46 Å. (d) Dividing the LG and MD at either end of the cross over linker followed by separate alignment of the LG domain (residues 1-306) and the MD (residues 317-483) allows improvement of the fit, suggesting the main determinant of the differential twists is the crossover linker. (e) Close-up of the dimer interfaces at the base of the MD. Likely interacting residues from PISA analysis are shown. Please note that the resolution of GBP1 dimer in this area does not allow modelling side chain conformations. (f) Close-up of superposed cross-over linkers for GBP1 (dark blue) and GBP5 (grey). (g) Close-up of superposed the hydrophobic plug of GBP1 (dark blue and light green) and GBP5 (grey). Putative residues stabilising the crossover arrangement are shown in stick representation.

1.3) The authors show that NB74 seems to disrupt the GBP layer on liposomes, but it does not seem to disrupt the hGBP1 micelles. Why is this the case? Shouldn't it disrupt the micelles as well?

The reviewer is correct that this may appear surprising. We occasionally do observe that the Nb74 also disrupts micelles, i.e. the number of disrupted micelles in the presence of Nb74 is higher than that observed in its absence. We provide several examples below. In general, however, the occurrence of disrupted micelles is small compared to the number of disrupted GBP coats observed on liposomes. We attribute this difference to the much larger curvature on GBP1 micelles, which provides maximum spacing of a still assembled GBP1

polymer and which appears to provide just enough space for the Nb74 to still bind without disrupting long-range order.

Intact GBP1_F micelle
+ Nb74

Distorted GBP1_F micelle + Nb74

1.4) Furthermore, would this nanobody interfere with GBP coat assembly in cells? and subsequent caspase-4 activation?

To answer this question, we first tested GBP1 coat formation on immobilised bacteria by incubating *S. Typhimurium* with GBP1 in the presence of recombinantly produced GFP-Nb74 fusion protein (we experimentally verified that GFP-Nb74 still binds nucleotide-activated GBP1 dimers as expected; new **Extended Data Fig. S14**). We then imaged coated bacteria using confocal microscopy. Consistent with our cryo-EM images, we observe that Nb74 does not completely abrogate coat formation, but we observe recognisable confinement of Nb74 to delimited puncta on the coat in regions where GBP1 signal is substantially weaker than on other regions of the coat where Nb74 signal is absent, consistent with our cryo-ET observations that show distorted GBP1 coats on liposomes (new **Fig. 5d**). To test how nanobody binding affects coat formation in bacteria-infested cells, we generated CRISPR/Cas9-engineered HeLa GBP1 KO cells stably expressing mCherry-GBP1 under a Tet-inducible promoter (HeLa Δ GBP1 + Tet-mCherry-GBP1). We then infected the cells transiently expressing GFP-Nb74 under a CMV promoter with *S. Typhimurium* and imaged GBP1 coat formation at 2 hours post infection. Like the observations on bacteria alone, we observe that Nb74 often accumulates within distinct patches on the coat (new **Fig. 5e**). In accordance with our other observations, we assume that locally there appears to be some tolerance towards accommodating nanobody in the coat, but that it is not integrated uniformly. However, we observe that local confinement of Nb74 is generally less prominent in cells compared to the *in vitro* bacterial assays and cryo-ET experiments on coated liposomes (new **Extended Data Fig. 14**). This suggests that, in comparison to the *in vitro* assays, the GBP1 coat on cytosolic bacteria is either less densely packed under the conditions tested (and hence more permissive to uniform integration of GFP-Nb74), that the GFP-Nb74 expression level is substantially below stoichiometric molar ratios, or that the resolution of the confocal imaging is insufficient to robustly image local differences in coat density in cellular assays.

We believe investigation of the effect of Nb74-distorted coats on caspase-4 activation goes beyond the scope of this study. We have meanwhile identified other nanobodies that completely abrogate coat formation and will report on these and their effect on GBP1-mediated activation of caspase-4 in detail in another study.

We have added the following paragraph to the manuscript:

*To investigate whether Nb74 also affects GBP1 coat formation on gram-negative bacterial pathogens, we imaged mCerulean3-expressing Salmonella Typhimurium after incubation with GBP1F and recombinant GFP-Nb74. Consistent with our cryo-EM data, we observe that Nb74 does not completely abrogate coat formation but appears to be preferentially confined to sharply delimited puncta on the coated bacteria. At and surrounding these puncta, GBP1 coat density is noticeably weakened, consistent with our cryo-ET observations showing partially disrupted coats on SUVs [Fig. 5d, Extended Data Fig. S14a,b]. We next asked how nanobody binding affects GBP1 coat formation in bacteria-infected cells and generated CRISPR/Cas9-engineered HeLa GBP1 knockout cells stably expressing mCherry-GBP1 under a Tet-inducible promoter (HeLa GBP1 + Tet-mCherry GBP1). We then infected cells transiently expressing GFP-Nb74 with S. Typhimurium and imaged GBP1 coat formation at 2 hours post infection using confocal microscopy. Like the *in vitro* observations on bacteria alone, we observe that Nb74 stains cytosolic bacteria non-uniformly and preferentially accumulates within distinct patches on the GBP1 coat [Fig. 5e, Extended Data Fig. S14c]. However, we observe that local confinement of Nb74 is generally less pronounced in cells compared to the *in vitro* bacterial assays and cryo-ET*

experiments on coated liposomes [Extendend Data Fig. S14c], suggesting that GBP1 coats on encapsulated bacteria in infected cells are more permissive to nanobody integration under the conditions tested.

Fig. 5. | Electron cryo-tomography and confocal imaging of the GBP1 coat (a) Electron cryo-tomogram of GBP1-coated liposomes. Projected z-stack of reconstructed tomogram and close-up of an individual z-slice showing discernible repetitive subunits. (b) Segmented tomogram from (a); GBP1: pink, membranes: green. (c) Cryo-EM micrograph of GBP1F-coated liposomes in the presence of Nb74. The close-up shows a comparison to GBP1F-coated liposomes in the absence of Nb74 (lower panel). The presence of Nb74 results in only partially coated liposomes with a higher degree of structural disorder. (d) Confocal microscopy images of immobilised *S. Typhimurium* incubated with GBP1F in the presence of recombinant GFP-Nb74 at 2 hours post infection with mCherry-GBP1. (e) Confocal microscopy of HeLa Δ GBP1 + Tet-mCherry-GBP1 cells expressing GFP-Nb74 at 2 hours post infection with mCherry-GBP1. Both immobilised bacteria and cytosolic bacteria in infected cells show a non-uniform distribution of GFP Nb74 across the GBP1 coat.

Extended Data Fig. S14. | Confocal imaging of GFP-Nb74 on GBP1 coat formation on gram-negative bacteria in vitro and in infected cells. (a) Confocal fluorescence images of GFP-Nb74 on GBP1 coats of immobilised *S. Typhimurium* expressing mCherry-GBP1. Images are immunostained for GBP1. (b) SEC-MALS profile of GFP-myc-Nb74 in complex with GBP1F-GDP-AIF3. Insets show SDS-PAGE and western blot stained for GBP1 and myc for the indicated peak fractions. (c) Confocal fluorescence images of GFP-Nb74 on GBP1-coated mCherry-GBP1-*S. Typhimurium* in infected HeLa Δ GBP1 + Tet-mCherry-GBP1 cells induced with doxycycline (dox). Split-channel images are close-ups of marked areas.

1.5) Concerning the protrusions shown in Fig. 4f: As the authors note themselves, it is also possible that the liposomes are needed to stabilize or seed such tubular structures. The diameter of the protrusions is similar to the GBP1 micelle, which would argue that no lipid tube is found in the center of the protrusion (59.8 ± 2.4

nm vs. 58.1 +/- 1.2 nm). Moreover, as the authors show in Fig. 7, an LPS micelles covered with hGBP1 still allow to distinguish the membrane in the center and is larger in diameter (~63 nm).

We agree with the reviewer that it is currently unclear whether the liposome-protrusions contain lipid tubes, and our present thinking is that rather than constricted lipid tubes, the protrusion may extract individual lipids rather than completely constricting the membrane. Support for this interpretation comes from our observations with the R584-586A mutant (see **point 1.1**), which forms tubes even in the absence of lipids (something we never observe for wild-type GBP1). It is possible that the C-terminal arginine patch, located in direct vicinity of the membrane-anchoring farnesyl motif, binds polar lipid headgroups and that tight packing of the outstretched stalks is precluded if no neutralising lipid charges are present. Forced neutralisation of the charged patch in the R584-586A mutant may avoid the requirement for additional lipid countercharges for assembly of the α 13 helices. In contrast, as the reviewer correctly notes, our data on LPS clearly show a lipid tube in remodelled LPS micelles with a GBP1 coat. The precise molecular origin for these observations, and how they are to be reconciled with the membrane fragmentation capability of actively hydrolysing GBP1 are important points for future mechanistic studies.

1.6) The authors present a set of new mutants in Fig. 6 that should still dimerize but not associate with membranes due to disrupted MD alignment. What is the phenotype of these mutants on coating of bacteria in cells. It would strengthen the paper to provide such data from *in cellulo* experiments in addition to structural data.

To address this question, we have we have generated CRISPR/Cas9-engineered HeLa GBP1 KO cells stably expressing mCherry-GBP1 variants under a Tet-inducible promoter (HeLa Δ GBP1 + Tet-mCherry-GBP1) and have tested their effect on GBP1 coat in *S. Typhimurium* infected cells at 2 hpi. We quantified integrated fluorescence intensities of GBP1 coats on cytosolic *S. Typhimurium* for the different GBP1 variants (**new Fig. 8**) and find that three of the variants affecting MD alignment and membrane recruitment in the *in vitro* assays also show similar effects in cells. Like for the *in vitro* assays the strongest variant is the D398-P310A, which reduces average coat intensity by 56% ($P=0.0002$). Two other variants (Y143A, affecting the hydrophobic plug in the crossover dimer, and K466D, putatively affecting the electrostatic zipper of the aligned MDs) also showed significantly reduced coat intensities of ~30%. One single residue variant (D308S) did not show significant effect (**Fig.8 b,c**). We also verified that the observed differences are not due to differences in expression levels; we find that *dox*-induced expression levels of wild-type and variant mCherry-GBP1 are equivalent within error (**Fig.8c, Extended Data Fig. S23**). We note that in our *in vitro* assays we strongly promote coat formation using AIF3-stabilised GBP1, whereas in cells GBP1 will exist in a mixture of nucleotide-bound states under constant turnover. We consider the magnitude of effects observed *in cellulo* therefore yet more encouraging in support of a critical role of the cross-over conformation of dimeric GBP1 for priming GBP1 coat formation on bacterial membranes.

We have added the following paragraph to the manuscript:

*Having established the importance of the outstretched crossover arrangement of GBP1 for binding to LPS membranes in vitro using GDP-AIF3-stabilised GBP1 dimers, we next asked whether it is required for antimicrobial coat formation in infected cells using the previously characterised GBP1 variants (see Fig. 6). We used CRISPR/Cas9-engineered HeLa GBP1 knockout cells stably expressing mCherry-GBP1 variants under a Tet-inducible promoter (HeLa Δ GBP1 + Tet-mCherry- GBP1) cells for infection with mCerulean3-expressing *S. Typhimurium* and quantified coat density on cytosolic bacteria at two hours post infection (Fig. 8a). Consistent with the in vitro liposome co-sedimentation assays, we observed the most significant effect for the cross-over mutant D308/L309/P310A, which resulted in a 56% reduction in coat density relative to wild-type GBP1. Other mutants (K466D, Y143A) showed weaker effects with 35 and 37% reduction in coat density, respectively (Fig. 8b). The D308S mutant alone showed no significant effect. To verify that observed differences in coat formation are not in part the result of differential dox-induced expression levels, we quantified relative expression levels of wild-type and variant GBP1 and found no significant differences (Fig. 8c, Extended Data Fig. S23). Together, the data from in vitro and in cellulo experiments suggest that the outstretched MD crossover conformation of nucleotide-activated GBP1 dimers is important for promoting efficient coat formation on target membranes.*

Fig. 8. Effect of MD cross-over destabilisation on antibacterial GBP1 coat formation in *S. Typhimurium* infected epithelial cells. HeLa (Δ GBP1 Tet-mCherry-GBP1) expressing wild-type or variant mCherry-GBP1 were infected with mCherry3-S. Typhimurium for 2h. (a) Representative of confocal images of GBP1-coated bacteria. Close-up of marked areas are shown in the bottom row. Blue: mCherry3-S. Typhimurium; Red: mCherry-GBP1 variants (b) Mean and standard deviation of the normalised corrected total coat fluorescence (CTCF). Statistical significance of differences relative to wild-type GBP1 was determined using Welch's t-test with Bonferroni correction (GBP1 D308A/L309A/P310A $P=0.000012$ (****); GBP1 D308S $P=0.11$ (non-significant, ns), GBP1 K466D $P=0.043$ (*), GBP1 Y143A $P=0.022$ (**). (c) Immunoblot validation of the CRISPR-engineered HeLa (Δ GBP1) and HeLa (Δ GBP1 Tet-mCherry-GBP1) genotypes. Dox-induced expression of GBP1 variants shows comparable expression levels for all variants. β -actin was used as a loading control.

Reviewer #2:

This study by Kuhm and colleagues describes a thorough structural and biophysical characterization of GBP1 as a function of GTP hydrolysis and farnesylation. The work seems technically sound and novel. The GBP1 micelles are stunning structures and likewise, the tubular membrane formations are very curious. While the study provides a deep insight into the biophysical mechanisms that determine the formation of these structures, the functional relevance to cell-autonomous immunity is not addressed. While the quality of the structural/biophysical work certainly justifies publication in NSMB in my opinion, making the functional/immunological connection with these findings would make for a substantial improvement. As the authors identified several mutations in GBP1 that are important, one can't help but wonder how these mutations would impact the immune effector functions of GBP1. This aside, the study is a great piece of work.

Thank you for your appreciation of our work and the useful feedback!

Other comments:

1) The EM micrographs in Figure 1 and 3 are very difficult to read at the printed size. Consider rescaling.

We have reformatted Figures 1 and 3 for better readability/visibility.

2) Figure 1B shows so many overlapping SEC traces that it is hard to discern which is which.

To improve clarity, we have added an inset in Fig. 1b showing the individual peaks with a fixed offset. Please note that the individual traces are plotted separately in Extended Data Figure 1.

3) The authors describe a procedure for *in vitro* and *in vivo* farnesylation of GBP1, but no data is presented to confirm the presence and occupancy of the modification.

Our protocol is a modified adaption for *in vitro* and *in vivo* farnesylation of GBP1 as described in PMID: 20348589 where it was shown using mass spectrometry that the protein is farnesylated. We purify farnesylated GBP1 using hydrophobic interaction chromatography (see profiles below), which we use to separate modified and unmodified fractions. We therefore are certain that the fractions we work with are quantitatively farnesylated.

4) In my opinion the sequence of Nb74 should be published along with this study. It would make for a more transparent and reproducible study, while simultaneously improving the impact as it could be a resource to others studying GBP1.

We agree and we will make the bacterial and mammalian expression plasmids for (EGFP)-Nb74 plasmids and sequence available on Addgene upon publication of the study.

5) The epitope/paratope of Nb74 should be described in detail.

Since the resolution of the nanobody in the cryo-EM structure is very low and precludes detailed interpretation (see Extended Data Fig. S4 for a local resolution plot), we have so far refrained from modelling the nanobody and from providing a detailed description of the paratope. We include schematic illustration showing the interaction interface for the reviewer's information, but we would prefer to not include it in the manuscript as in our opinion the quality of the map in this region prohibits confident modelling.

6) The assignment of the inner densities of GBP1 micelles in Figure 3 is somewhat problematic with the limited resolution. The use of Nb74 as a marker is very neat, but the assignment of the densities beyond that remain somewhat speculative.

We agree, but we believe to have phrased our interpretation carefully. The class averages of the micelles, together with the 2D class averages of the GBP1 coat on SUVs and LPS and of the GBP1-coated tubular protrusions all support a model in which an outstretched GED extends beyond the MDs to reach out towards the membrane, or to interact at the center of the GBP1 micelles. How the α 13 helices assemble in micelles, on the membrane and in tubular protrusions is indeed and how they stabilize the GBP1 assemblies are indeed important questions for future studies.

7) Figure 4F construes a picture of GBP1-concentration dependent tubular membrane formations, but perhaps some more thorough statistics/distributions of the different types of membrane formations is warranted (beyond the single images shown).

We have added a series of additional images for each concentration in **Extended Data Fig. S9** and have quantified the fraction of SUVs with tubular protrusions as a function of GBP1 concentration (Extended Data Fig. S9b). This analysis shows that tubular protrusions start to form occasionally at 7 μ M, occur frequently at 15 μ M and form the dominant phenotype at 30 μ M GBP1.

Extended Data Fig. S9. I GBP1_F-GDP·AIF₃ tubular protrusions. (a) GBP1 coatomer formation is concentration dependent. A visible coat starts to form at GBP1_F concentrations of 7 μM . At a concentration of 15 μM and higher, GBP1-coated tubular protrusions become visible and are the dominant structures at concentrations exceeding 30 μM . (b) Quantification of GBP1-coated protrusions forming on coated SUVs. For each condition, 100 SUVs were randomly selected, classified based on the presence or absence of GBP1-coated protrusions and the fraction of protrusion-forming SUVs was calculated. Error bars represent a 95% confidence interval from non-parametric bootstrapping. (c) 2D class average of negatively stained GBP1_F-GDP·AIF₃ tubular protrusion (left panel). The computed power spectrum shows a principal layer line at 0.086 \AA^{-1} , corresponding to a periodicity of 11.6 nm along the filament axis (right panel)

8) The tubular membrane formations are attributed to farnesylated GBP1 and indeed it is shown that increasing concentrations will result in more tubular membranes. This begs the question whether the effect is specific to GBP1; would any farnesylated protein in similar concentrations induce similar effects?

This is an interesting question that we currently have no fully conclusive answer to. The GBP1 coat on protrusions shows at least short-range order, and the class averages of GBP1 on SUVs and LPS show that lateral interactions of LG and MD are likely stabilising the polymer, suggesting that farnesylation alone is not enough. To answer the reviewer's question whether the effect is GBP1-specific, we constructed a GBP2 chimera harbouring the GBP1-CTIS motif for farnesylation. Unlike GBP, which readily forms micelles in the absence of lipids and decorates SUVs, including the formation of protrusions, farnesylated GBP2-CTIS does not self-assemble into ordered micelles and does not associate with SUV membranes at the same concentrations. This suggests that structural elements other than farnesylation alone are required for the observed effect.

9) For the dual-trap membrane fragmentation assay, presumably the trace prior to GBP1 addition serves as a control to demonstrate a GBP1-specific effect. In this respect, it would be good to see in the figure exactly when the GBP1 is added to the flow channel. As presented, a scenario in which a lipid coated bead remains stable until some critical amount of 'damage' is accumulated, after which it fragments rather suddenly is also plausible. I would therefore advocate for a control experiment where a donor bead is incubated in flowing buffer with GTP but without GBP1 for the same amount of time as the presented experiment *with* addition of GBP1.

We agree with the reviewer that the assay as presented was missing relevant controls. Following the reviewer's suggestions, we have expanded the dual trap membrane fragmentation assay with a series of new experiments using controls including other guanosine nucleotides and a GTPase activity deficient mutant of GBP1 (R48A). Please see **Extended Data Fig. S13** and our detailed response to **Point 3** of **reviewer 3**.

10) In Figure 5, the effects of Nb74 on the tomographic reconstructions is attributed to steric hindrance of GBP1-GBP1 contacts. However, it is not described how Nb74 affects GTP hydrolysis, which may also alter the behaviour of GBP1 on membranes.

To test whether Nb74 affects GTP hydrolysis of GBP1, we performed a GTPase assay in the presence of Nb74. The hydrolysis of farnesylated GBP1 (GBP1_F) was not affected by the addition of Nb74 in a 1:1 molar ratio (the same ratio used when determining for cryo-EM SPA and cryo-ET imaging of the GBP1 coat).

11) The mass photometry data presented in Figure 6; the materials/methods indicated a wide range of used concentrations. It is not clear whether the MP data presented in figure 6 is all collected at the same concentration. If the concentrations are not equal between the different mutants, it is difficult to say that the shifts in monomer-dimer equilibrium are direct effects of the mutations.

Thank you for noticing this. We do agree and apologise for this oversight. To exclude that shifts in monomer-dimer equilibrium are attributable to different protein concentrations used, we repeated the experiments with equivalent concentration (15 nM) for all GBP1 variants. In accordance with our previous experiments these data show that none of the variants has a significant effect on GBP1 dimerisation via the LG domain. We have replaced all relevant panels in **Fig. 6** and **Extended Data Fig. S18** with the new data. Below we show a comparison for the reviewer's convenience.

Reviewer #3:

Remarks to the Author:

This study by Kuhm et al presents cutting-edge structural and molecular characterisation of hGBP1, which is the forerunner of the IFN γ -inducible Guanylate Binding Protein family. GBP1 has indispensable roles in sensing microbial infection, and recent cell biological and biochemical evidence points to its role in detecting cytosolic LPS from Gram-negative bacteria. However, the structural and molecular aspects of how GBP1 does this, especially whether it has membrane-modifying activities remain unknown. Here, the authors assess GBP1 oligomerisation on host membranes and LPS vesicles predominantly using CryoEM structures and in vitro assays. The authors find that dimerization of hGBP1 when associated with GDP-AIF3 is similar to the previously described dimerization mode of hGBP5 with a crucial crossing over of the MD helices which facilitate membrane insertion. They investigate this with mutagenesis and membrane co-sedimentation assays. Cooperative lateral assembly of hGBP1 coatomers is also dependent on the MD crossover and can lead to fragmentation of host membranes. They finally characterise the interaction between hGBP1 and three distinct LPS structures from pathogens, showing a common coatomer structure and Sm R595 LPS vesicle remodelling. In my opinion the structural aspects of this study are thorough and well performed, and contribute significant new knowledge regarding the mechanical aspects of GBP1 oligomerisation (i.e. MD crossing over and presumably lateral oligomerisation of dimers, membrane tubulation). The main limitation is that the physiological implications have not been investigated and some experiments lack appropriate controls.

Thank you for your appreciation of our work and the constructive comments.

COMMENTS:

1. *To the best of my knowledge, this is the first evidence of a GBP modifying membranes in this way, which has been previously suggested but not demonstrated formally. This is a major strength.*

Thank you! We agree this finding is novel and exciting, and we now provide additional support that further strengthens our previous observations that this remodelling activity is dependent both on the cross-over conformation and on the GTPase activity of GBP1 (see reply to **points 3 and 4**).

2. *The first structure of GBP1 is in the presence of a nanobody. What evidence do the authors have that this is also the case in cells and is physiologically relevant?*

Indeed, the cryo-EM structure of dimeric nucleotide-activated GBP1 was obtained in complex with a nanobody (Nb74; Figure 1). All other cryo-EM/ET data on GBP1 assemblies throughout the manuscript (**Fig. 3**, **Fig. 5**, **Fig. 7**) have been performed without nanobody; but high-resolution 2D classes or tomographic reprojections show that the relevant unit in these assemblies is consistent with the outstretched cross-over conformation observed in our nanobody-stabilised GBP1 structure from single-particle analysis. In addition, validation of the relevance of this conformation through disruptive mutations and their effect on membrane-binding potential in vitro (**Fig. 5**) and in cells (**Fig. 8**) provide solid support that this conformation is important for GBP1 coat formation under physiologically relevant conditions.

3. *The optical tweezer assay in figure 4j-l needs appropriate controls to be more informative. While agreeing in principle with previous observational data showing a lack of observed SUVs in the presence of GBP1-GTP and the presence of coated SUVs in the presence of GBP1-GDP-AIF3, this assay should also be carried out with the apo-GBP1, as well as GDP- and GDP-AIF3 associated GBP1 to rule out the possibility that GBP1 alone leads to lipid transfer in this context. Likewise, this should also be carried out with a GTPase-deficient mutation such as GBP1R48A (Praefcke et al., 2004; Li et al., 2017; Xavier et al., 2020; Fisch et al., 2021) or other mutants they have generated to show conclusively that it is the active GTP hydrolysis event that leads to fragmentation.*

We agree and have performed a series of experiments to show that the observed membrane fragmentation is indeed dependent on GBP1 and the GTPase activity of GBP1. We have repeated dual trap optical tweezers experiments either with GBP1 + GTP, or with GBP1 in the presence of nucleotides (GDP, GDP-AIFx), as well as with a GTPase-dead mutant (R48) as suggested by the reviewer (we have also verified that this mutant indeed shows no GTPase activity and does not associate with membranes; **Extended Data Fig. S13c**). In addition, we have performed control experiments using apo-GBP1, and with GTP in the absence of GBP1. To quantify membrane release from the donor bead, we monitored the time-averaged fold-change of fluorescence relative to baseline after incubation onset. The experiments demonstrate that significant release of membrane is only observed for GBP1 in the presence of GTP, and not in the presence of other nucleotides or if a GTPase-deficient mutant (R48A) is used. Likewise, no membrane release is observed for GBP1 in the absence of

nucleotides, or for GTP in the absence of GBP1 (Extended Data Figure 13a). Together, these data support our previous conclusion, and our complementary EM observations, that GBP1 has nucleotide-dependent membrane remodeling activity that can promote membrane fragmentation during GTP hydrolysis.

Extended Data Fig. S13. | Dual trap bead-supported membrane transfer assay. Mean fold-changes in interbead fluorescence measured in the the dual trap membrane transfer assay for GBP1 in the presence/absence of different guanosine nucleotides and for the GBP1-R48A mutant deficient in GTPase activity. Error bars represent standard deviations and significance levels were determined using a non-parametric Mann-Whitney U test. (* $P < 0.05$, ** $P < 0.01$, *** $P < 0.001$). (b) SEC-MALS profiles for non-farnesylated GBP1-R48A (left panel) and farnesylated (right panel) GBP1_F-R48A showing that deficiency in GTPase activity prevents dimerisation. (c) GTPase activity assay for the GTPase activity-deficient GBP1-R48A and GBP1_F-R48A. GTPase activity for wild-type GBP1 is shown for comparison. Low luminescence signal corresponds to high GTPase activity. Bovine serum albumin (BSA) was used as a negative control (n=3). (d) Representative negatively stained micrograph of GBP1_F-R48A in the presence of BPLE-SUVs. No coated SUVs could be observed.

4. The effects of mutants in Fig 6 are relatively minor, perhaps because the *in vitro* assay. It would be good to validate these findings in cells – for example by introducing GBP1 mutants into GBP1^{-/-} macrophages or epithelia and assess the effects of the newly identified residues in the cross-over region or others. Do these mutants behave similarly when binding SUV host membranes to LPS-containing bacterial membranes? Do they still translocate to bacteria? Can they still facilitate Caspase 4 activation?

To address this question, we have we have generated CRISPR/Cas9-engineered HeLa GBP1 KO cells stably expressing mCherry-GBP1 variants under a Tet-inducible promoter (HeLa Δ GBP1 + Tet-mCherry-GBP1) and have tested how they affect efficiency of GBP1 coat in *S. Typhimurium* infected cells. We indeed observe that the mutants behave similar to the observations made *in vitro* on SUVs and LPS micelles. Please see **Figure 8, Extended Data Fig. S23** and our detailed response to **point 1.6 of reviewer 1**.

We believe investigation of the effect of Nb74-distorted coats on downstream caspase-4 activation goes beyond the scope of this study, but we agree this is a very interesting question that we will be investigating in forthcoming work.

The flipping out of the α 12/13 on LPS OMVs is really interesting. Have the authors attempted mutations that would abolish the flipping out of the α -12/13 regions on LPS OMVs?

We agree that this an interesting mechanistic question and we are working on time-resolved cryo-EM imaging of the conformational changes that are occurring during nucleotide activation. However, we think a detailed mechanistic study of this process goes beyond the scope of the current study. Of note, chemically crosslinking the α 12 region of the GED to the MD through engineered disulfide bridges abolished the MD crossing of the dimer membrane recruitment, suggesting that both structural changes must occur in a concerted manner.

MINOR POINTS:

• I find it interesting and slightly concerning that in Fig. 1b, the commonly used and more well-understood transition state-mimicking compounds GTPyS, GMPPNP and GppCp do not induce dimerization, despite being non-hydrolysable GTP analogues to which GDP-AIF₃ is assumed to be similar. Could this please be acknowledged in text along with a hypothesis as to what the reason for this may be? This could inform future studies as to the limitations of these ubiquitous compounds.

We entirely agree with the reviewer that this is interesting and also share the view that this is slightly concerning. We can note that we observe similar effects also for other members of the GBP1 family, but we currently have no conclusive explanation for the origin of this effect. More generally, for non-hydrolysable nucleoside analogs it is known that the position of the modification within the triphosphate exerts influence on its binding affinity to a particular enzyme and its hydrolysis rate. Related factors may be responsible here. It is also instructive to recall that GDP-AIF₃ is considered a transition state analog of the GTP hydrolysis reaction, whereas other non-hydrolysable GTP analogs mimic GTP binding but do not stabilise the transition state of the enzymatic reaction. It is possible that for GBP1 (and other GBPs) GTP binding is not sufficient to induce the structural changes required to form the outstretched dimeric arrangement, and that the GTP hydrolysis step is required to trigger these transitions. Some support for this hypothesis may come from our observation that in the presence of GTP we can find assembled GBP structures that contain GBP1 in an outstretched conformation but, because we are not inhibiting the GTPase hydrolysis cycle in this case, their occurrence is much reduced compared to conditions with GDP-AIF₃, in which all molecules are arrested in the activated state. We concur with the reviewer that this is an important mechanistic question with potential implications beyond GBPs alone and we are keen to pursue future studies along these lines. Until we have a solid mechanistic explanation, we prefer not to comment or speculate as this may cause more confusion to the field than it may help.

• The mutants used in Fig. 6 should be more thoroughly explained in the text, as while being present in the figures, Y143A and K466D are not mentioned in the text itself. Furthermore, specific variants should be referenced in text when discussing conclusions which apply to the mutant in question, rather than describing them as 'LG-MD' or 'MD-MD' variants. Likewise, the propensity of D308A/L309A/P310A mutant variants for MD crossing should be assessed using Cryo-EM as in Fig. 1h, I, instead of presuming due to the position of the mutations. This would also inform as to whether the domain rearrangement still occurs in the absence of MD crossing.

We now explain different variants in more detail in the text and reference to them explicitly throughout the text. To address the reviewer's question regarding the propensity of the mutants for MD crossing, we have collected cryo-EM data of the GBP1 D308-P310A variant, which showed the strongest effect both *in vitro* and in the cellular assays prepared for this revision. Since these variants appear to affect the likelihood of forming the cross-over conformation observed for wild-type GBP1, attempting to resolve them by cryo-EM is exceedingly difficult. To compare the likelihood of forming the cross-over conformation between the mutant and WT GBP1,

we analysed the class populations after unbiased 2D classification of particle sets drawn from both datasets. A good proxy for this analysis is the comparison of class populations for a projection view along the long axis of a potential outstretched GBP1 dimer, which allows discerning the existence of parallel aligned MDs by evaluating the presence or absence of the bright signal associated with a projection along the alpha-helical stalks. A simulation (a; below) illustrates this effect. Comparing the abundance of this view in unbiased 2D classifications shows that this class is the most prominent class for the WT GBP1 dataset but is absent in most 2D classes of equivalent projection views in the mutant dataset, suggesting the cross-over conformation with parallel stalks is disfavored for the mutant relative to WT GBP1. We note that such analysis is not without potential flaws due to a large number of factors potentially affecting particle orientation in cryo-EM imaging, but along with other data in the manuscript (Figure 4,5,6,7,8) is supportive of our model that destabilisation of the cross-over conformation by mutation or steric hindrance lies at the basis of the observed effects of the cross-over conformation for membrane recruitment and remodeling.

(a) Simulated projections

(b) Experimental 2D class averages from unbiased classification of wild-type GBP1 and D308-P310A variant

• Fig. 3g is referenced in text, however does not exist

Thank you. We have corrected the referencing.

• The figures could be better labelled – for instance Fig 2A conformers could be showed separately to understand the change in the linker. Similarly, the ‘yellow’ in panel 2D is not clear.

We have attempted to improve the presentation of these figures.

Decision Letter, first revision:

Message: Our ref: NSMB-A47474A

16th Apr 2024

Dear Dr. Jakobi,

Thank you for submitting your revised manuscript "Structural basis of membrane targeting and coatomer assembly by human GBP1" (NSMB-A47474A). It has now been seen by the original referees and their comments are below. The reviewers find that the paper has improved in revision, and therefore we'll be happy in principle to publish it in Nature Structural & Molecular Biology, pending minor revisions to satisfy the referees' final requests and to comply with our editorial and formatting guidelines.

To facilitate our work at this stage, it is important that we have a copy of the main text as a word file. If you could please send along a word version of this file as soon as possible, we would greatly appreciate it; please make sure to copy the NSMB account (cc'ed above).

Sincerely,
Kat

Katarzyna Ciazynska, PhD
(she/her)
Associate Editor
Nature Structural & Molecular Biology
<https://orcid.org/0000-0002-9899-2428>

Reviewer #1 (Remarks to the Author):

The authors have performed an. number of additional controls and experiments that address all the points I have raised. I have no further comments and congratulate the authors on this nice study.

Reviewer #2 (Remarks to the Author):

The authors have addressed my main comments. I would like to congratulate them on an excellent piece of work. I maintain that the study could benefit from establishing a better connection of their findings with the immune effector functions of GBP1, but the structural biology/biophysics experiments are sound, novel and exciting. I would like to advocate for early deposition of the nanobody construct in Addgene so that the information can be directly linked in the published paper (or otherwise for the nanobody sequence to be included in the supplementary material).

Reviewer #3 (Remarks to the Author):

In their revised version, the authors have addressed almost all queries raised in the previous submission. This is still a good piece of work that illuminates structural underpinning of GBP1 actions. The authors should address the following points:

1. It is a bit disappointing that they only measured bacterial trafficking and not caspase-4 activation in pyroptosis assays, which are relatively easy. Therefore, they can only comment on trafficking to the bacterium but not the downstream steps.
2. They measured total coat fluorescence in Fig 8. What was the relative percentage of bacterial coating by different GBP1 variants? Were similar number of cytosolic bacteria labelled? Some panels have more bacteria with GBP1 than others, but this could simply be the image chosen for figures.
3. The discussion around Zhu et al, which recently appeared in Science, should be updated. They also observed dimers of open conformation of GBP1, which is consistent with this work.

Author Rebuttal, first revision:**Reviewer response**

We thank all three reviewers for their positive comments and endorsement of our work.

Reviewer 1:

The authors have performed an. number of additional controls and experiments that address all the points I have raised. I have no further comments and congratulate the authors on this nice study.

Thank you for the nice comments.

Reviewer 2:

The authors have addressed my main comments. I would like to congratulate them on an excellent piece of work. I maintain that the study could benefit from establishing a better connection of their findings with the immune effector functions of GBP1, but the structural biology/biophysics experiments are sound, novel and exciting. I would like to advocate for early deposition of the nanobody construct in Addgene so that the information can be directly linked in the published paper (or otherwise for the nanobody sequence to be included in the supplementary material).

Thank you for the nice comments. We agree that future work building on the mechanistic insight obtained from the structural, biophysical, biochemical and cellular experiments reported here will be required to better connect these data to the immune effector function of GBP1 in different host-pathogen systems. Nevertheless, we think that on their own our data substantially advance the general understanding of GBP1 coat formation on intracellular bacteria. Connecting these data to the cellular mechanism of immune effector activity of GBP1 is work we will pursue in future studies.

We have made the primary sequence of the nanobody available in the supplementary information.

Reviewer 3:

In their revised version, the authors have addressed almost all queries raised in the previous submission. This is a good piece of work that illuminates structural underpinning of GBP1 actions. The authors should address the following points:

1. It is a bit disappointing that they only measured bacterial trafficking and not caspase-4 activation in pyroptosis assays, which are relatively easy. Therefore, they can only comment on trafficking to the bacterium but not the downstream steps.

We agree that investigating the effect on caspase-4 activation is interesting, but we maintain that this is not relevant for the conclusions drawn from the data in this article and should be subject to future work. The present article focuses on the mechanistic underpinnings of the coat formation process itself, and the membrane remodelling capabilities of GBP1 resulting from this.

2. They measured total coat fluorescence in Fig 8. What was the relative percentage of bacterial coating by different GBP1 variants? Were similar number of cytosolic bacteria labelled? Some panels have more bacteria with GBP1 than others, but this could simply be the image chosen for figures.

Since we are investigation salmonella-infected cells it is difficult to compare the relative number of GBP1-coated bacteria between different infected cells, and between mutants, because salmonella exist in both vacuole-bound and cytosolic form. Only cytosolic bacteria will be coated (see e.g. PMID: 32783936), whose number can differ between cells and thus quantification of number of coated cells is not conclusive. Therefore, we have limited our analysis to quantification of relative coat fluorescence intensities of coated cells, which are directly comparable between the different variants.

3. The discussion around Zhu et al, which recently appeared in Science, should be updated. They also observed dimers of open conformation of GBP1, which is consistent with this work.

We have updated the discussion and reference. We note, however, that the original preprint from Zhou et al. did not describe an open conformation of GBP1 dimers and this conclusion was revised in the journal-published version after appearance of the preprint of our present study in which evidence of the dimer was reported.

Final Decision Letter:

Message: 5th Sep 2024

Dear Dr. Jakobi,

We are now happy to accept your revised paper "Structural basis of antimicrobial membrane coat assembly by human GBP1" for publication as an Article in Nature Structural & Molecular Biology.

Note the policy of the journal on data deposition:

<http://www.nature.com/authors/policies/availability.html>.

Your paper will be published online soon after we receive proof corrections and will appear in print in the next available issue. You can find out your date of online publication by contacting the production team shortly after sending your proof corrections.

You may wish to make your media relations office aware of your accepted publication, in case they consider it appropriate to organize some internal or external publicity. Once your paper has been scheduled you will receive an email confirming the publication details. This is normally 3-4 working days in advance of publication. If you need additional notice of the date and time of publication, please let the production team know when you receive the proof of your article to ensure there is sufficient time to coordinate. Further information on our embargo policies can be found here:

<https://www.nature.com/authors/policies/embargo.html>

An online order form for reprints of your paper is available

at <https://www.nature.com/reprints/author-reprints.html>. Please let your coauthors and your institutions' public affairs office know that they are also welcome to order reprints by this method.

Please note that *Nature Structural & Molecular Biology* is a Transformative Journal (TJ). Authors may publish their research with us through the traditional subscription access route or make their paper immediately open access through payment of an article-processing charge (APC). Authors will not be required to make a final decision about access to their article until it has been accepted. Find out more about Transformative Journals

Sincerely,

Katarzyna Ciazynska, PhD
(she/her)
Associate Editor
Nature Structural & Molecular Biology
<https://orcid.org/0000-0002-9899-2428>